

# Constraining pesticide degradation in conceptual distributed catchment models with compound-specific isotope analysis (CSIA)

Sylvain Payraudeau[1], Pablo Alvarez-Zaldivar[1], Paul van Dijk[2], Gwenaël Imfeld[1]

[1]Institut Terre et Environnement de Strasbourg, Université de Strasbourg, CNRS/ENGEES, ITES UMR 7063, Strasbourg, 67084, France
[2]Chambre Régionale d'Agriculture Grand Est, Espace Européen de l'Entreprise, 2 rue de Rome CS 30022 Schiltigheim, 67013 France

*Correspondence to*: Sylvain Payraudeau (sylvain.payraudeau@engees.unistra.fr)

**Abstract.** The prediction of pesticide dissipation on the catchment scale through hydrological models often encounters challenges due to the limited availability of field data capable of distinguishing between degradative and non-degradative processes. This limitation complicates the calibration of pesticide dissipation and frequently results in equifinality, impeding the reliable forecast of pesticide persistence in soil and its transportation from agricultural plots to the catchment outlet. This study examines the benefits of integrating pesticide Compound-Specific Isotope Analysis (CSIA) data to improve the predictive accuracy of models assessing pesticide persistence in soil and off-site transport at the catchment scale. The research was conducted in a 47-ha crop catchment, focusing on the widely used pre-emergence herbicide S-metolachlor. A novel conceptual model, named PIBEACH, was developed to predict daily pesticide dissipation in soil and its transport to rivers, incorporating changes of the carbon isotopic signatures ($\delta^{13}$C) of the targeted pesticide during degradation. Parameter and model uncertainties were estimated using the Generalized Likelihood Uncertainty Estimation (GLUE) method. The inclusion of field data on S-metolachlor concentrations in the topsoil and their associated $\delta^{13}$C values in the model resulted in a more than two-fold reduction in uncertainties related to S-metolachlor degradation half-life and six metrics of pesticide persistence and off-site transport. Moreover, the study indicates that a moderate yet targeted sampling effort can effectively identify hot-spots and hot-moments of pesticide degradation in agricultural soil when isotope fractionation is integrated into the model. In summary, the incorporation of CSIA data into conceptual distributed hydrological models holds the potential to alleviate parameter equifinality, therewith significantly improving our ability to predict the dynamics of pesticide degradation on the catchment scale.

## 1 Introduction

Ongoing intensive use of pesticides can lead to accumulation of pesticides and their transformation products in agricultural soils, and pesticide off-site transport for decades. Pesticides on agricultural soil can be mobilized and transported off-site towards aquatic ecosystems during rainfall-runoff events, thereby threatening drinking water supply as well as ecosystem and human health (Vorosmarty et al., 2010; Fenner et al., 2013; Stone et al., 2014; Weisner et al., 2021). While pressure on aquatic



ecosystems is increasing, the contribution of pesticide dissipation processes in soil and pesticide off-site transport on the catchment scale remains difficult to evaluate and predict accurately. In this context, reliable and validated hydrological models to predict pesticide dissipation and off-site transport hold potential to address fundamental questions, such as the relationship between hydro-climatic factors and pesticide dissipation processes and the transfer risks of pesticides on the catchment scale.

The contribution of degradative, including biotic or abiotic degradation, and non-degradative pesticide dissipation processes, such as sorption, leaching, volatilization, off-site transport, may be evaluated for this purpose in models including both pesticide dissipation and hydrological functioning (Larsbo and Jarvis, 2005; Steffens et al., 2015; Gassmann, 2021). The accurate and sufficiently precise prediction of pesticide dissipation and off-site transport on the agricultural catchment scale raises, however, two fundamental limits. First, the complexity of reactive transport models has increased over the last decade

while field datasets available to calibrate and validate such models remain scarce (Medici et al., 2012; Ammann et al., 2020). In addition, existing models frequently fail to accurately quantify the contribution of concomitant pesticide dissipation processes for accurate estimation of off-site pesticide transport (Gassmann et al., 2021).

Regarding the first issue, model prediction of pesticide dissipation and off-site transport towards aquatic ecosystems generally

suffers from field data limitations to constrain model parameters. Conceptual and physically-based hydrological models account for many dissipation and transport phenomena, each of which generally accounting for several physico-chemical processes and described by several parameters obtained in reference laboratory experiments (Gatel et al., 2020; Gassmann, 2021). Linking pesticide dissipation parameters obtained under laboratory conditions with field processes is however difficult, because the extent and the range of values for a given parameter may differ under laboratory and field conditions (Malone et

al., 2004; Köhne et al., 2009). As a result, parameter calibration is often required to fit observations. For instance, pesticide half-life ($DT_{50}$) in soil, i.e., the time required for 50 % dissipation of the parent compound in soil, is typically derived under laboratory conditions (Lewis et al., 2016). However, when extrapolated to field conditions, other process-controlling parameters, including soil moisture, temperature, $K_{OC}$, organic carbon content and porosity distribution, influence pesticide concentrations (Dubus et al., 2003). As a result, $DT_{50}$ values extracted from literature typically span up to one order of

magnitude (Lewis et al., 2016; Wang et al., 2018). Pesticide half-life is thus generally considered as a lumped calibration parameter in reactive transport models (Dubus et al., 2002; Gassmann, 2021), merging both degradative and non-degradative pesticide dissipation processes across different soil components and redox conditions (Honti and Fenner, 2015).

Pesticide concentrations in topsoil and at the catchment outlet is currently the most accessible information for modelling

pesticide reactive transport at the catchment scale (Wendell et al., 2024). In the best of cases, calibration is carried out based on concentrations of both parent and main transformation products in soil, runoff or aquifer (Gassmann, 2021), which improved model prediction (Sidoli et al., 2016; Lefrancq et al., 2018). Transformation products, however, are numerous, several of them are unknown and possibly further degraded, which may lead to incomplete or uncertain mass balance accounts. As a result, pesticide and transformation product concentration data often barely help to disentangle degradative from non-degradative





pesticide dissipation processes (Fenner et al., 2013) in catchments. Existing models thus remain limited to distinguish competing dissipation pathways and to quantify their contribution.

Evaluating the contribution of dissipation processes of pesticides is nevertheless essential in field studies because degradation is the only process that contributes to decrease parent pesticide load in environmental compartments. While pesticide
degradation prevents their long-term accumulation in environmental compartments, it may also generate unknow and potentially toxic transformation products that are further transported in surface and groundwater. In this context, pesticide compound specific isotope analysis (CSIA) offers an alternative approach because information on parent pesticide degradation is derived independently from concentrations and produced transformation products, and relies on stable isotope ratios (Elsner and Imfeld, 2016; Hofstetter et al., 2024). During chemical and biological reactions of pesticide degradation, molecules with
lighter isotopes (e.g., $^{12}C$) are generally transformed at slightly higher rates relative to their heavier counterparts (e.g., $^{13}C$). This results in a kinetic isotope effect leaving a chemical imprint in the form of characteristic changes in isotope ratios of reacting molecules (Elsner, 2010). In contrast, non-degradative pesticide dissipation processes, such as sorption generally do not result in significant isotope fractionation (Schmidt et al., 2004; Alvarez-Zaldivar et al., 2018; Droz et al., 2021). Hence, incorporation of pesticide CSIA data in hydrological models bears the potential to reduce the uncertainty associated with
pesticide dissipation processes by distinguishing pesticide degradative and non-degradative processes, and transformation pathways. Such uncertainty typically results from compensating effects across competing dissipation processes, as shown previously for legacy contaminants in aquifers (Hunkeler et al., 2008; Blázquez-Pallí et al., 2019; Thouement et al., 2019; Antelmi et al., 2021; Prieto-Espinoza et al., 2021) or in wetlands (Alvarez-Zaldivar et al., 2016).

More recently, incorporation of pesticide CSIA data in a parsimonious lumped model based on transport formulation by travel-time distributions has shown that CSIA data can constrain lumped model (Lutz et al., 2017) to interpret pesticide transport on the catchment scale. Lumped models, however, generally describe the overall hydrological behavior without spatial information, such as soil water content or temperature, across the catchment (Fatichi et al., 2016). This particularly limits the identification of contaminant sources and degradation hot spots across the catchment (Grundmann et al., 2007). Distributed
conceptual and physically-based models, in contrast, explicitly represent spatial hydro-climatic dynamics regulating hydrological processes (e.g., runoff, infiltration). However, incorporation of pesticide CSIA data during the calibration and validation of distributed hydrological models has never been implemented, mainly because a detailed field dataset including pesticide CSIA data was lacking. The environmental application of pesticide CSIA is also limited by the general lack of knowledge regarding the initial isotopic composition of active substances in commercially available formulations used by
farmers. This limitation has recently been addressed through ISOTOPEST, which compiles and distributes isotopic signatures of commercial formulations to expand the application of CSIA for pesticides (Masbou et al., 2024).



The objective of this study was to examine the advantages of incorporating both topsoil pesticide concentrations and CSIA data for model calibration to evaluate pesticide dissipation processes and off-site pesticide transport at the catchment scale.

The study relied on a unique field data set, which includes weekly soil and runoff water concentrations and carbon isotope signatures ($\delta^{13}$C) of S-metolachlor, a widely used and well characterized pre-emergent herbicide (Alvarez-Zaldivar et al., 2018). This dataset allowed us to interpret changes of stable isotope ratios and effectively constrain the degradation of S-metolachlor in a well-characterized 47-hectare agricultural headwater catchment located in Alsace, France. Additionally, our study investigated the influence of hydro-climatic dynamics on pesticide degradation within distributed hydrological models

and their associated model structures, drawing upon previous work by Boesten and Vanderlinden (1991), Schroll et al. (2006) and Gassmann (2021). We adapted the Bridge Event Continuous Hydrological (BEACH) model, a distributed conceptual model that simulates variations in soil water content conditions, based on prior soil characterizations of the study catchment (Lefrancq et al., 2017; Alvarez-Zaldivar et al., 2018; Lefrancq et al., 2018). This adaptation resulted in the creation of the Pesticide isotope BEACH model (PiBEACH), which includes the reactive transport of pesticides and considers pesticide

carbon stable isotope fractionation, following the approach proposed by Lutz et al. (2017). During the calibration process, we applied the Generalized Likelihood Uncertainty Estimation (GLUE) technique using Monte-Carlo methodology to PiBEACH. This allowed to examine the improvement in both prediction accuracy and model parameter identification when using CSIA data.

## 2 Material and Methods

### 2.1 Field site


The dataset, including concentrations and carbon isotope signatures ($\delta^{13}$C) of S-metolachlor in soil, was collected from March 19th (day 0) to July 12th (day 115), 2016 from the 47-ha Alteckendorf headwater catchment (Bas-Rhin, France; 48 47 19.56 N; 7 35 2.27 E) (Alvarez-Zaldivar et al., 2018; Lefrancq et al., 2018) (Fig. 1).



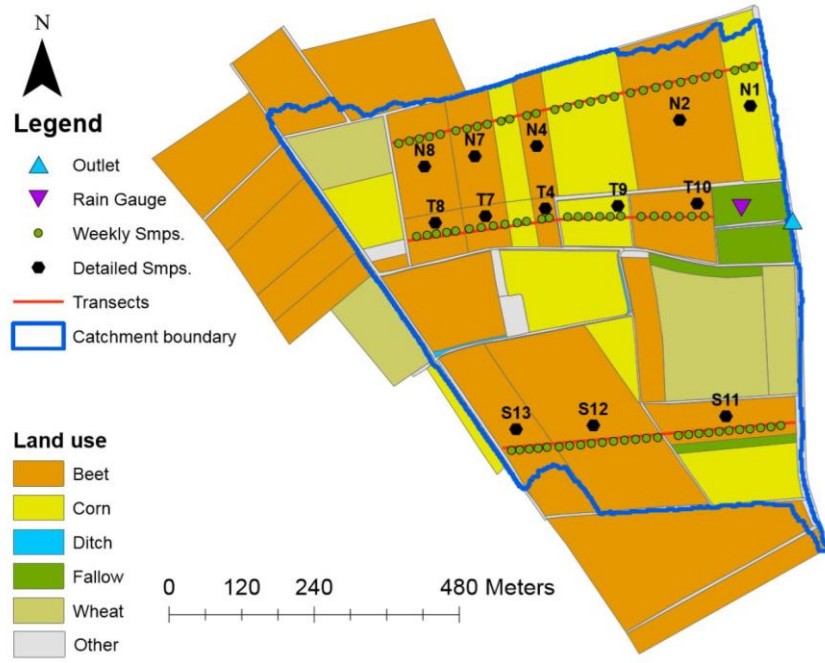


**Figure 1:** The headwater catchment of Alteckendorf (Bas-Rhin, France) with the experimental setup, including transects and plot sampling, and the landuse for 2016. "Other" contains roads, grass strips and orchards.

Climate and hydrological characteristics are provided in Table S1. Arable land dominates the catchment, with corn (18 %),

sugar beet (70 %) and wheat (3 %) being the principal crops in 2016. The catchment has a tile drainage system of unknown spatial extent at 0.8 m depth and water flows in ditches to a 50 cm diameter pipe at the outlet (Fig. 1). The soil type is a Haplic Cambisol Calcaric Siltic and Cambisol Eutric Siltic on hillsides (north and south) and Cambisol Colluvic Eutric Siltic in the central valley. Soil characteristics from 48 topsoil samples (0-20 cm) and six 2 m depth profiles indicated low spatial variability across the catchment (Lefrancq et al., 2018), with low variation of grain size distribution (clay: 30.8 ± 3.9 %, silt: 61.0 ± 4.5

%, sand: 8.5 ± 4.2 %) and soil composition ($CaCO_3$: 1.1 ± 1.6 %; organic matter = 2.2 ± 0.3 %, total soluble phosphorus: 0.11 ± 0.04 g kg$^{-1}$, and CEC: 15.5 ± 1.3 cmol kg$^{-1}$). The soil pH was 6.7 ± 0.8 (n = 30). A compacted plough layer was observed between 20 and 30 cm depth. In 2016, S-metolachlor was spread on corn and beet plots over three applications (March 20-25th, April 13-14th and May 25-31st). Corn and beet plots accounted for 88 % of the catchment area. Application dates, doses and formulation of S-metolachlor applied were collected for each plot by farmer surveys (Table S2).




## 2.2 Soil and water sampling and outlet discharge measurements

Topsoils (0-1 cm) were sampled from individual plots and upstream-downstream transects (Fig. 1 and S1) (Alvarez-Zaldivar et al., 2018). Soil samples from13 marked plots (before S-metolachlor applications and on days 1, 50 and 100 after applications) and from north, valley and south transects across the catchment (weekly from March 19th, i.e., day 0, to July 12th, i.e., day 115) 140 were collected for quantification and CSIA ($\delta^{13}$C) of S-metolachlor in soil.

Outlet runoff discharge was measured using a Doppler flowmeter (2150 Isco). Automatic, refrigerated continuous flow proportional sampling (Isco Avalanche with twelve bottles of 330 mL) was conducted at fixed weekly discharge volumes ranging from 50 to 150 m$^3$ from April to June to capture increasing minimum baseflow discharges observed in 2016 (Alvarez-145 Zaldivar et al., 2018). To obtain sufficient amount of S-metolachlor for quantification and CSIA, samples were combined in composite samples based on hydrograph, i.e., base-flow, rising and falling limb, yielding volumes ≥ 990 mL (Alvarez-Zaldivar et al., 2018). The piezometric monitoring of the shallow aquifer could not be carried out due to a lack of wells on the study site.

## 2.3 S-metolachlor quantification and $\delta^{13}$C analysis

To separate dissolved and particulate phases, water samples were filtered through 0.7 µm glass fibre filters. Methods of S-metolachlor extraction from soil and water samples with an AutoTrace 280 Solid Phase Extraction (SPE) system (Dionex) with SolEx C18 cartridges (Dionex ®) and quantification with GC-MS (MS, ISQ™, Thermo Fisher 173 Scientific) were detailed previously (Alvarez-Zaldivar et al., 2018; Gilevska et al., 2022b). Quantification limits for S-metolachlor were 0.01 µg L$^{-1}$ from water and 0.001 µg g$^{-1}$ from soil samples (d.w.), with a total analytical uncertainty of 8 % and 16 %, respectively. 155 Changes of carbon isotope ratios of S-metolachlor in soil and water were analysed using a GC-C-IRMS system consisting of a TRACE™ Ultra gas chromatograph (ThermoFisher Scientific) coupled via a GC IsoLink/ Conflow IV interface to an isotope ratio mass spectrometer (DeltaVplus, ThermoFisher Scientific) (Alvarez-Zaldivar et al., 2018; Gilevska et al., 2022b) as described in section 5 of SI. The reproducibility of triplicate measurements was ≤0.2‰ (1σ). The minimum peak amplitude needed for accurate $\delta^{13}$C measurements was 300 mV, corresponding to about 10 ng of carbon injected on column.




## 2.4 PiBEACH model description

*PiBEACH development.* The Pesticide-isotopes BEACH model (PiBEACH) was developed in Python from the conceptual Bridge Event Continuous Hydrological (BEACH) model (Sheikh et al., 2009). BEACH was chosen in this study for its ability to predict both daily topsoil moisture and the daily catchment discharge (i.e., Nash-Sutcliffe efficiency of 0.79) on a 42 ha catchment (Sheikh et al., 2009), with loess soils and crops similar to those observed in the study catchment. The BEACH model is a grid-based model generating spatially distributed soil water content and discharge at the outlet, depending on prevailing hydrological processes, i.e., evaporation, plant transpiration, percolation, deep percolation, lateral flow and runoff (Sheikh et al., 2009). Daily vertical water fluxes across soil layers then lateral fluxes were considered at the cell size from upstream to downstream, with the surface flow direction extracted from digital elevation model, and using flow-accumulation grid-based functions without numerical scheme (Sheikh et al., 2009). BEACH was fed by daily meteorological records, including rainfall and mean air temperature, soil physical properties, including saturated hydraulic conductivity, bulk density, porosity, both for the plow layer and a deeper soil layer, and crop-specific agronomical information. The development of the PiBEACH conceptual model relied on field knowledge of hydrological dynamics and associated S-metolachlor flows in the Alteckendorf catchment (Lefrancq et al., 2017; Lutz et al., 2017; Alvarez-Zaldivar et al., 2018; Lange et al., 2018; Lefrancq et al., 2018), translating our experimentalist's understanding (perceptual model) into conceptual model as proposed by Ammann et al. (2020).

Compared with BEACH features, PiBEACH includes (i) a mixing topsoil layer (McGrath et al., 2008) and additional deepest soil layers to include the shallow groundwater contribution to discharge, (ii) daily changes of topsoil temperature (Neitsch et al., 2011), (iii) daily variations of topsoil hydraulic properties accounting for the impact of agricultural practices with a crop-specific agronomical model (Lefrancq et al., 2018), (iv) pesticides first-order degradation and linear sorption/desorption processes, and (v) pesticide carbon stable isotopic fractionation and transport of isotopologues (e.g., $^{13}$C and $^{12}$C) (Lutz et al., 2017; Alvarez-Zaldivar et al., 2018). PiBEACH requires amounts and dates of pesticide application for each plot.

*Comparison of PiBEACH with physically-based models.* The finite difference or finite volume method is typically used to simulate agricultural catchment hydrology in 3D physically-based models, e.g., Hydrogeosphere (De Schepper et al., 2015) or CATHY (Gatel et al., 2020). However, modelling pesticide reactive transport including pesticide sorption-desorption and degradation on the catchment scale remains challenging. This is because the numerical diffusion (Gatel et al., 2020) is generally of the same order of magnitude as pesticide export coefficient, i.e., from 1‰ to 1% of the applied amount of pesticides. These numerical diffusion issues can be mitigated, though this leads to longer computation times in both 2D (Lutz et al., 2013) and 3D (Gatel et al., 2020) models. PiBEACH simulates water and pesticide fluxes on a daily time step, tracking flow from upstream to downstream along branches of the drainage network using flow-accumulation grid-based functions (Sheikh et al., 2009). While it avoids numerical issues by not employing differential equations and associated numerical schemes, this approach also limits its ability to predict fast flow dynamics, i.e., runoff genesis and preferential water flow. This network can





be extracted from a LIDAR digital elevation model (Lefrancq et al., 2017). A vertical mass balance is computed using the sum
of discharges and pesticides masses of the upstream grid cells as lateral inputs for the downstream grid cells (section 6 of the
SI and equation 2). The daily resolution of PiBEACH reduced by one to two orders of magnitude lower the computation time
compared to 2D or 3D models for catchments of similar size (Gatel et al., 2020). The limitations associated with this conceptual
approach are discussed in the next section.

*PiBEACH description and sub-model components.* To integrate landscape components enhancing or limiting pesticide off-site
transport, i.e., grass strips or roads, the Alteckendorf catchment was represented with a $2 \times 2$ m resolution. The vertical soil
profile was described by five successive soil layers including, from topsoil to groundwater bottom (Fig. 2): (i) a mixing layer
(*z0* from 0 to 1 cm) (McGrath et al., 2008), (ii) a plow layer observed in the field (z1 from 1 to 30 cm), (iii) a layer controlled
by artificial drainage pipes, such as observed in the Alteckendorf catchment (z2 from 30 to 80 cm), (iv) a groundwater layer
divided into a variably saturated layer (*z3*), and (v) a deeper permanently saturated layer (*z4*). The depth of groundwater layers
(*z3 + z4*) varied constantly from upstream to downstream. In the catchment, the maximum depth of *z3 + z4* was 23.2 m. Depth
distribution ($z_f$) between z3 and z4 layers was considered as a calibration parameter (Table S3 in SI; *z3 + z4* = 23.2 m; *z3* =
$23.2 \times z_f$; z4 = $23.2 \times (1 - z_f)$). Evaporation affects layers *z0* to *z3*, while transpiration depends on plant root depth following
crop-specific development stages (see sections 6.5 and 6.6 in SI). Lateral flow and percolation are calculated from upstream
to downstream cells for layers *z0* to *z3* (Manfreda et al., 2005). Percolation across the saturated layer *z4* routed up to the outlet
as a global linear reservoir (Manfreda et al., 2005).

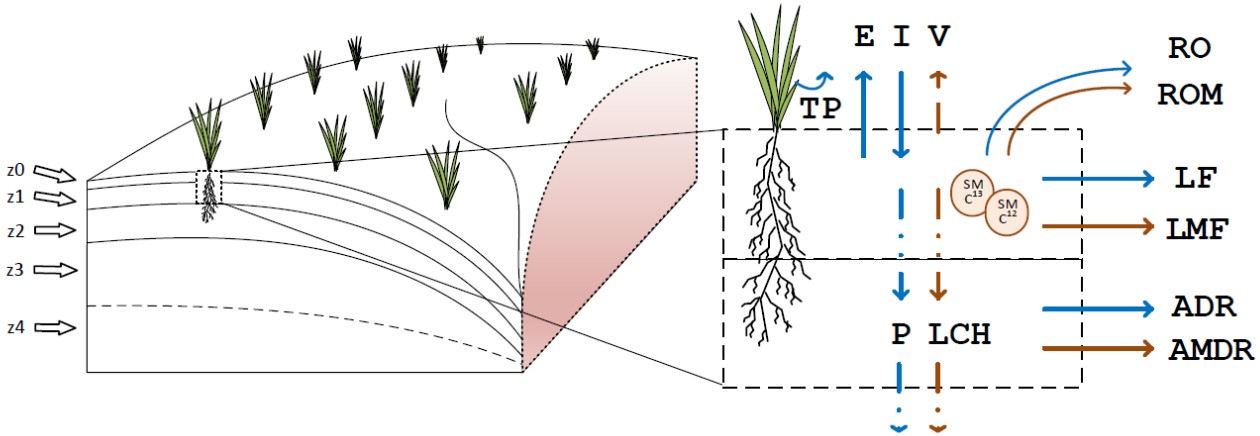

**Figure 2:** A conceptual 5-layer spatially distributed hydrological and reactive-transport PiBEACH model. Hydrological
processes include evaporation (*E*), transpiration (*TP*), percolation (*P*), volatilization (*V*), runoff (*RO*), lateral flow (*LF*), and
artificial drainage (*ADR*). Mass transfer processes include volatilization (*V*), runoff mass (*ROM*), lateral mass flow (*LMF*),



leaching (*LCH*), mass transfer via artificial drainage (*AMDR*), and degradation with isotope fractionation (symbolized by SM [13]C and SM [12]C).

The hydrological balance was described in Sheikh et al., (2009) and detailed implementation was provided for each process in section 6 of the SI. Overall, change in soil water content ($\theta$, m$^3$ m$^{-3}$) for each cell $i$ and for each layer with soil depth $Z$ (mm) was calculated from equation 1:

$$Z\frac{d\theta_i}{dt} = R_i - RO_i + \Delta LF_i - Ea_i - Ta_i - P_i \tag{1}$$

where rainfall ($R$), runoff ($RO$), net cell lateral inflow-outflow ($\Delta LF$), actual evaporation ($Ea$), actual transpiration ($Ta$) and
percolation ($P$) were expressed in mm H$_2$O d$^{-1}$.

The agronomical sub-model of PiBEACH (Lefrancq et al., 2017) (see SI section 7) provided temporal variation of topsoil soil hydraulic properties across the growing season, which significantly improved the hydrological balance of the headwater catchment (Lefrancq et al., 2017).

The pesticide dissipation sub-model of PiBEACH entailed the main processes to predict the pesticide reactive transport on the
catchment scale (Fig. 2). The pesticide mass ($M$, g) balance at each cell $i$ and for each layer was then given by:

$$\frac{dM_i}{dt} = A_i - ROM_i + \Delta LMF_i - V_i - LCH_i - DEG_i \tag{2}$$

where, for each mass component (g pesticide d$^{-1}$) including the mass applied ($A$) on $z0$, loss to runoff ($ROM$) only from z0, change due to lateral flux ($\Delta LMF$) from layers $z0$ to $z3$, volatilization ($V$) only from $z0$, leaching ($LCH$) from $z0$ to $z3$ and degradation ($DEG$) in the layers $z0$ to $z4$.


Mass distribution into the dissolved, sorbed and gaseous phases follows Leistra et al., (2001) and is detailed in section 8 of SI. Partition into the dissolved and adsorbed phases was determined by linear sorption, considering the organic carbon-water partition coefficient $K_{OC}$ (mL g$^{-1}$) normalized by the fraction of organic carbon $f_{OC}$ (kg kg$^{-1}$) in soil, where the pesticide dissociation coefficient ($K_d$, mL g$^{-1}$) was given by $K_d = K_{OC} \times f_{OC}$. Partition into the gas phase was obtained from the
dimensionless Henry constant, $K^{cc}_H = 9.55\ 10^{-5}$ for S-Metolachlor (Feigenbrugel et al., 2004). Generalized pesticide mass flux $J$ (µg d$^{-1}$) for each model layer was given by:

$$J = q_{x,y} \times C_{aq} \tag{3}$$

where $q_{x,y}$ was the water flux vector (mm d$^{-1}$) along the lateral ($x,y$) and vertical ($z$) direction and $c_{aq}$ was dissolved S-metolachlor concentration in the aqueous phase (µg mm$^{-1}$). For the topsoil layer ($z_0$), runoff and volatilization were also
considered such that:

$$J_{zo} = J + c_{aq}\left(RO e^{-\beta_{RO}D_{zo}} + \frac{1}{r_a + r_s}\right) \tag{4}$$

where $RO$ was runoff (mm) and $\beta_{RO}$ was a calibration constant ($1 \geq \beta_{RO} \geq 0$) and $D_{z0}$ (mm) was topsoil depth (Ahuja and Lehman, 1983). Volatilization was considered only during the first 5 days after application (Gish et al., 2011), (Prueger et al.,



2005), and follows (Leistra et al., 2001), with fluxes across topsoil controlled by air transport resistance, $r_a$, (d m$^{-1}$) and

diffusion resistance, $r_s$ (d m$^{-1}$) (Leistra et al., 2001) (see sub-section 8.2 of SI).

Biodegradation was assumed to occur only in bioavailable fractions of adsorbed (*ads*) and aqueous (*aq*) phases (Thullner et al., 2013). A similar isotope fractionation associated with degradation was then considered for the bioavailable fractions, including the adsorbed (Eq. 5, second term) and the aqueous (Eq. 6) phases. The bioavailable fraction was controlled kinetically

by an ageing rate $k_{age}$ (d$^{-1}$) on the adsorbed fraction (Schwarzenbach, 2003). As sorption generally does not significantly alter the isotope composition of pollutants (Schmidt et al., 2004; Droz et al., 2021), the ratios of light to heavy isotopologues did not significantly vary during sorption and ageing processes (Eq. 5, first term). Altogether, representing pesticide mass (*M*) as separate light (*l*) and heavy (*h*) isotopologues, the change in aqueous and adsorbed phases was given by:

$$\frac{dM_{ads}}{dt} = -k_{age}\left(M_{ads}^l + M_{ads}^h\right) - k_{deg}\left(M_{ads}^l + \alpha M_{ads}^h\right) \tag{5}$$


$$\frac{dM_{aq}}{dt} = -k_{deg}\left(M_{aq}^l + \alpha M_{aq}^h\right) \tag{6}$$

where $k_{deg} = ln\,(2)\,/\,DT_{50}$ and $DT_{50}$ (days) was the observed degradation half-life in soil. Stable isotope fractionation associated with S-metolachlor degradation in soil was considered through the carbon fractionation factor ($\alpha_C$), expressed as $\alpha_C = \varepsilon_C/1000 + 1$, where $\varepsilon_C$ (‰) was the carbon stable isotope fractionation value of the targeted pesticide retrieved from a degradation reference experiment. Although degradation rates generally decrease over soil depth, e.g., due to lower microbial activity (Cruz

et al., 2008; Lutz et al., 2017) or sorption (Arias-Estevez et al., 2008) in deeper soil layer, the lack of concentration and isotope data for S-metolachlor in deeper soil layers did not allow to consider depth-dependence degradation, and was thus not included in the model. Degradation rates for S-metolachlor freshly sorbed in the organic fraction of soil was assumed to be equal to degradation rates in the dissolved phase (Wu et al., 2011; Long et al., 2014). The reduction of bioavailability of the aged fraction of S-metolachlor as a function of the increasing fraction of irreversible sorbed S-metolachlor over time (Sander et al.,

2006; Arias-Estevez et al., 2008; Torabi et al., 2020) was taken into account with the rate $k_{irs}$ (d$^{-1}$) of irreversible sorption for the aged S-metolachlor mass ($M_{age}$, g):

$$\frac{dM_{age}}{dt} = -k_{irs}\left(M_{age}^l + M_{age}^h\right) \tag{7}$$

Decrease in $M_{age}$ due to abiotic degradation (Xu et al., 2018) was not included, since it was unlikely to be significant under abiotic conditions in the studied aerobic soil (Torabi et al., 2020), even after 200 days of incubation (Alvarez-Zaldivar et al.,

275    2018).

As implemented in advanced pesticide fate models such as MACRO (Garratt et al., 2003), the pesticide degradation rate ($k_{deg}$) in PiBEACH varies depending on soil hydro-climatic conditions, such as soil temperature and water content. Therefore, a dynamic degradation rate ($k_{Dynamic}$, d$^{-1}$) was adjusted daily with soil temperature ($F_T$) and water content ($F_\theta$), as follows:




$$k_{Dynamic} = k_{Ref} \times F_T \times F_\theta \tag{8}$$

where $k_{ref}$ (d$^{-1}$) was the degradation rate constant from degradation half-life (days) at reference conditions ($k_{Ref} = ln\,(2)\,/\,DT_{50,Ref}$). A dynamic half-time $DT_{50,Dynamic}$ was derived to be compared to $DT_{50,Ref}$ ($DT_{50,\,Dynamic} = ln\,(2)\,/\,k_{Dynamic}$).

For soil temperature dependence, the modified Arrhenius equation for low temperatures was considered (Boesten and Vanderlinden, 1991; Larsbo and Jarvis, 2003):

$$F_T = \begin{cases} 0, & if, T_C \leq 0 \\ \frac{T_{K,Obs}-273.15}{5} exp\left(\frac{E_a}{R}\left(\frac{1}{T_{K,Ref}} - \frac{1}{T_{K,Obs}}\right)\right), & if, 0 < T_C \leq 5 \\ exp\left(\frac{E_a}{R}\left(\frac{1}{T_{K,Ref}} - \frac{1}{T_{K,Obs}}\right)\right), & if\ T_C > 5 \end{cases} \tag{9}$$

where $T_K$ and $T_C$ were soil temperatures in Kelvin and Celsius, respectively and $T_{K,Ref}$ was the reference temperature at 293.15
K. $E_a$ was the S-metolachlor activation energy (23.91 10$^3$ J mol$^{-1}$) (Jaikaew et al., 2017) and $R$ was the gas constant (8.314 J mol$^{-1}$ K$^{-1}$). This modified Arrhenius equation was validated against $DT_{50}$ values derived from microcosm degradation experiments with S-metolachlor conducted at 20 and 30°C (Section 9 of SI, Figure S4).

Influence of water content followed Walker, (1974) and Larsbo and Jarvis (2003):

$$F_\theta = min\left(1.0, \left(\frac{\theta_t}{\theta_{Ref}}\right)^{\beta_\theta}\right) \tag{10}$$

where $\beta_\theta$ was a calibration constant and $\theta_{Ref}$ the reference water content, which was set at 0.2 m$^3$ m$^{-3}$. Note that $F_\theta$ was slightly modified from Walker (1974), as microcosm experiments for S-metolachlor did not show an increase in degradation rates with increases in moisture contents (Figure S4).

## 2.5 Model limitations


Primarily developed to predict pesticides dissipation in topsoil, four aspects of the PiBEACH design and features may limit its transposition to predict pesticide dynamics at the outlet of the headwater catchment scale. Considering a daily time step, PiBEACH was not designed to accurately simulate fast flow dynamics, i.e., runoff genesis and preferential water and soil flow at the headwater scale (Sheikh et al., 2009). This issue could be addressed by coupling PiBEACH with a distributed event-
based model, such as the Limburg Soil Erosion Model (OpenLISEM) (Baartman et al., 2012), which recently incorporated a pesticide module (OLP) (Commelin et al., 2024). Based on a conceptual approach with a linear reservoir to represent the dynamic of the shallow aquifer, PiBEACH does not explicitly consider the impact of transit time distribution on pesticide release from groundwater into the river (Hrachowitz et al., 2016). Whilst the implementation of transit time distribution in





lumped pesticide reactive transport model is promising (Lutz et al., 2017), implementing transit time distribution in distributed
model remains difficult, as shown for nitrate (Kaandorp et al., 2021).

In addition, pesticide leaching across soil layers was considered without differentiation of matrix and preferential flows. Macropores can also play a significant role on pesticide breakthrough in the vadose zone (Urbina et al., 2020). However, the explicit integration of macropore at the catchment scale would requires advanced in situ measurements (Weiler, 2017), and a combination of geostatistical methods, pedotransfer functions or meta-models, i.e., simplified statistical models built with 1D
soil reactive transport models such as MACRO (Lindahl et al. 2008).

PiBEACH gathers main compartments of the catchment, including soil layers, plants and crops, and major processes controlling pesticide fate. However, S-metolachlor uptake by plants was not considered and may be quantitatively insignificant since no correlation was observed between the remaining mass of S-metolachlor in the topsoil and the sugar beet and corn growth (Lefrancq et al., 2018). Pesticides wash off on plants was not considered in PiBEACH as S-metolachlor is a pre-
emergent herbicide applied on bare soil.

Considering degradation in reactive transport models as a lumped process with only one half-life parameter is a strong simplification which may alter the predictability of degradation in models. However, this limitation is shared among existing pesticide transport models (Leistra et al., 2001; Lindahl et al, 2008; Lutz et al., 2017; Gatel et al.,2020). The implementation of soil temperature and moisture dependence for pesticide degradation, as proposed in this study, may strengthen the
description of degradation within the catchment and across the hydrological or growing season (Gassmann 2021). Nonetheless, soil temperature and moisture are also forcing variables of the soil microbial activity, and thus of biodegradation, which remains extremely difficult to characterize and conceptualize with respect to pesticide transformation ( Imfeld and Vuilleumier, 2012; Bongiorno, 2020;  Höhener et al., 2022) for implementation in reactive models at the catchment scale.

## 2.6 Model uncertainty assessment

The GLUE and formal Bayesian methods are commonly used methods to quantify the uncertainty of hydrological models and provide distributions of parameters and water discharges. Bayesian methods are more convenient to calculate the uncertainty interval of one-step ahead forecasting with a formal or exact likelihood function (Jin et al., 2010). In contrast, the GLUE method integrates a real calibration process in which both inputs and model structural errors contribute to uncertainties of the model outputs (Beven et al., 2007). The GLUE method was thus adopted in our study to calibrate PiBEACH and retrieve
output uncertainties. Rather than seeking an optimal model solution, the GLUE approach recognizes that more than one model structure or parameter set may lead to acceptable model results, i.e. equifinality (Beven and Binley, 1992). The GLUE method involved a sampling method of PiBEACH parameters, an objective function incorporating observed dataset (i.e., topsoil S-metolachlor concentration only, then combined to S-metolachlor $\delta^{13}$C), a threshold of this objective function to select behavioural parameter sets, and the calculation of posterior probability distributions for parameters and uncertainties associated
to the outputs of PiBEACH.



For the sampling method of parameters, PiBEACH required the calibration of 43 parameters (Table S3 in SI). The range, i.e. min-max values, of these parameters were defined based either on literature or field data collected in 2012 and 2016 (Lefrancq et al., 2017; Lefrancq et al., 2016, 2018; Alvarez-Zaldivar et al., 2018). These parameters were assumed to be *a priori*

uniformly distributed within these min and max values (Table S3 in SI). To reduce the number of runs required by the GLUE method, three steps were successively applied. First, a pre-sensitivity global analysis based on the Morris method (Morris, 1991; Herman and Usher, 2017) was conducted (section 10 of SI) to select the most sensitive parameters. Although the Morris method yields a qualitative indication of relative parameter importance, it is efficient compared to other sensitivity approaches (Gan et al., 2014) that screen for sensitive parameters (Herman et al., 2013). The Morris method allowed to reduce the

PiBEACH parameter number from 43 to 25 (Table S3 in SI). Second, a Latin-Hypercube sampling (Herman and Usher, 2017) was used to reduce the numbers of runs (n = 2500) to cover the parameter space for the 25 parameters. To further reduce the computation time, the GLUE assessment focused on the growing period (March 19[th] to July 12[th], 2016), where pesticide degradation and exports are of most significance. Initial hydrological state was estimated from a spin-up period of one full hydrological year (Oct. 1[st], 2015 - Sept. 30[th], 2016) and hydrological parameters calibrated against observed discharge at the

catchment outlet (March 19[th] and July 12[th], 2016) using particle swarm optimization (Bratton et al., 2007).

For the second step of the GLUE method, the *KGE* (Gupta et al., 2009) metric was adopted as the objective function to maximize during calibration. Goodness of fit between simulations and observations are given relative to a maximum efficiency of 1 and given by:


$$KGE = 1 - \sqrt{(r-1)^2 + (\alpha_{KGE} - 1)^2 + (\beta_{KGE} - 1)^2} \qquad (11)$$

where r was a linear correlation coefficient, $\alpha_{KGE} = s_I / s_0$, and $\beta_{KGE} = \mu_i / \mu_0$, where $\sigma$ and $\mu$ represent the standard deviation and mean of simulated (*i*) and observed (*o*) values, respectively.


The *KGE* metric was selected to provide equal weight across correlation, bias and variability measures. *KGE* metric is also an improved measure of model performance compared to other metrics, such as the mean squared error and the Nash-Sutcliffe efficiency, which favour parameter values and underestimate variance of the model results (Gupta et al., 2009). Three *KGE* metrics were calculated with (i) $KGE_{SM}$ with weekly topsoil S-metolachlor concentration, (ii) $KGE_\delta$ with weekly topsoil $\delta^{13}$C

and (iii) $KGE_Q$ with daily discharge at the outlet. The *KGE* metrics were successively built to assess the benefit of CSIA by incorporating topsoil S-metolachlor concentration only or in combination with topsoil S-metolachlor $\delta^{13}$C observations, and the minimal topsoil S-metolachlor concentration and $\delta^{13}$C sampling effort to reduce the uncertainties of PiBEACH outputs. For the latter, KGE metric incorporates topsoil S-metolachlor concentration and $\delta^{13}$C from (i) individual plot observations, (ii) aggregated plot observation along three transects across the catchment and (iii) transect composite soil.




The threshold to retain acceptable model results runs (out of 2500 simulation runs) for topsoil S-metolachlor degradation and transport was set to $KGE_{SM}$ >0.5 and $KGE_Q$ >0.5. An additional isotope data constraint of $KGE_\delta$ >0.8 was then consider to evaluate the benefit of CSIA during the calibration process. We hypothesized that the uncertainty of degradation extent controlled by $DT_{50,Ref}$ (Eq. 8) should be reduced considering CSIA, resulting in a smaller standard deviation of $DT_{50\,Ref}$.


Distributions of the acceptable parameter sets (i.e. $KGE_{SM}$ >0.5 and $KGE_Q$ >0.5 and $KGE_\delta$ >0.8) with the 25 most sensitive parameters were extracted for the fifth and last step of GLUE. The PiBEACH outputs were then reported as an ensemble and with the 95 % confidence interval obtained from the acceptable parameter sets.

## 3 Results and discussion

**3.1 Topsoil hydro-climatic dynamics and effect on S-metolachlor degradation rates**

Topsoil ($z0$) water contents were highly dynamic, in agreement with weekly measurements of field water content in topsoil (Fig. 3A) and previous BEACH application on similar area, soil and crops catchment (Sheikh et al., 2009). Simulated outlet discharge fitted with observations (Fig. 3B), with maximum $KGE_Q$ (Kling-Gupta efficiency metric on discharge) values of 0.75 which underscores the ability of PiBEACH to reproduce prevailing hydrological trends. The tendency of PiBEACH to 390 overestimate runoff generation is related to the basic assumptions of the SCS-CN approach (see section 6.2, SI), and to the daily time-step concealing daily distribution of rainfall intensity. The hydro-climatic conditions yielded varying soil degradation rates $k_{Dynamic}$ and associated half-times $DT_{50,Dynamic}$ (Fig. 3C) over time. Predicted low S-metolachlor degradation rates in the first 10 days of observation ($k$ = 0.015 ± 0.005 d$^{-1}$) corresponded to an early cold (7.1 ± 1.7 °C) and dry period (rainfall of 7.4 mm) (Fig. 3A). In contrast, increasing degradation rates ($k$ = 0.031 ± 0.002 d$^{-1}$) from 50 to 120 days following 395 the first application of S-metolachlor were mainly associated with higher temperatures (16.9 ± 3.6 T°C). Simulated S-metolachlor concentrations (Fig. 3D) and stable isotope ratios (Fig. 3E) in topsoil fitted well with weekly field observations.





**Figure 3:** Predicted catchment topsoil ($z0$: 0-1cm) water content (m$^3$ m$^{-3}$) and temperatures expressed as a fraction relative to the maximum air temperature ($T_{z0}/T_{air, max}$ ; with $T_{air, max}$ = 27°C) between March 14$^{th}$ (day -5, before first application) and July 12$^{th}$ (day 115), with shaded area depicting the 95 % confidence intervals (CI) and observed water content ($\theta_{obs}$) weekly in topsoil (A). The associated error bars are linked to soil bulk density estimations required for gravimetric to volumetric conversions. Simulated mean and observed daily outlet discharge with shaded area depicting 95 % confidence interval for the model ensemble (B). Mean dynamic degradation half-life ($DT_{50,Dynamic}$) of S-metolachlor for the topsoil layer ($z0$) with shaded area representing 95 % confidence intervals and reference half-life ($DT_{50,Ref}$) (C). S-metolachlor concentrations (D) and $\delta^{13}$C values (E) for observed composite topsoil transects (i.e., $z0$) and simulated model ensemble mean and CI's (E). Error bars indicate standard deviations of measured S-metolachlor concentrations (D) and $\delta^{13}$C values (E). The three application periods are indicated as App. 1 (days 0 & 6), App. 2 (day 25), and App. 3 (Days 67 & 74) in panel E.

Out of a total of 2500 simulation runs associated with the Latin-Hypercube sampling, only 672 acceptable runs (i.e. with $KGE_Q$ >0.5 and $KGE_{SM}$ >0.5) were retained. An additional isotope data constraint with $KGE_\delta$ >0.8 resulted in 244 runs for the model ensemble. Hence, among 672 acceptable runs, considering hydrology ($KGE_Q$ >0.5) and S-metolachlor concentrations ($KGE_{SM}$ >0.5), only 36 % could reliably predict S-metolachlor degradation ($KGE_\delta$ >0.8). For the 64% other runs, S-metolachlor dissipation in soil were well predicted ($KGE_{SM}$ >0.5) but with a less accurate prediction of degradation (wider range of $DT_{50,Ref}$ value, Fig. 4) than runs constrained by $KGE_\delta$, including CSIA data in the calibration phase.

### 3.2 Uncertainty reduction through incorporation of CSIA data

The benefit of using in the calibration phase both topsoil pesticide concentration and CSIA data was examined by comparing the calibrations of PiBEACH parameters with S-metolachlor concentrations in the topsoil only (NIC: no isotope constraint), or with both S-metolachlor concentrations in the topsoil and associated CSIA data (WIC: with isotope constraint). For the composite transect soil (Fig. 4), mean $DT_{50,Ref}$ ($\mu$) (Eq. 8) values were similar for calibrations with (WIC) or without CSIA (NIC) data. However, standard deviations associated with $DT_{50,Ref}$ values for NIC calibration were twice larger than for WIC calibration (Fig. 4). This indicates that CSIA data significantly reduced the uncertainty associated with S-metolachlor degradation.





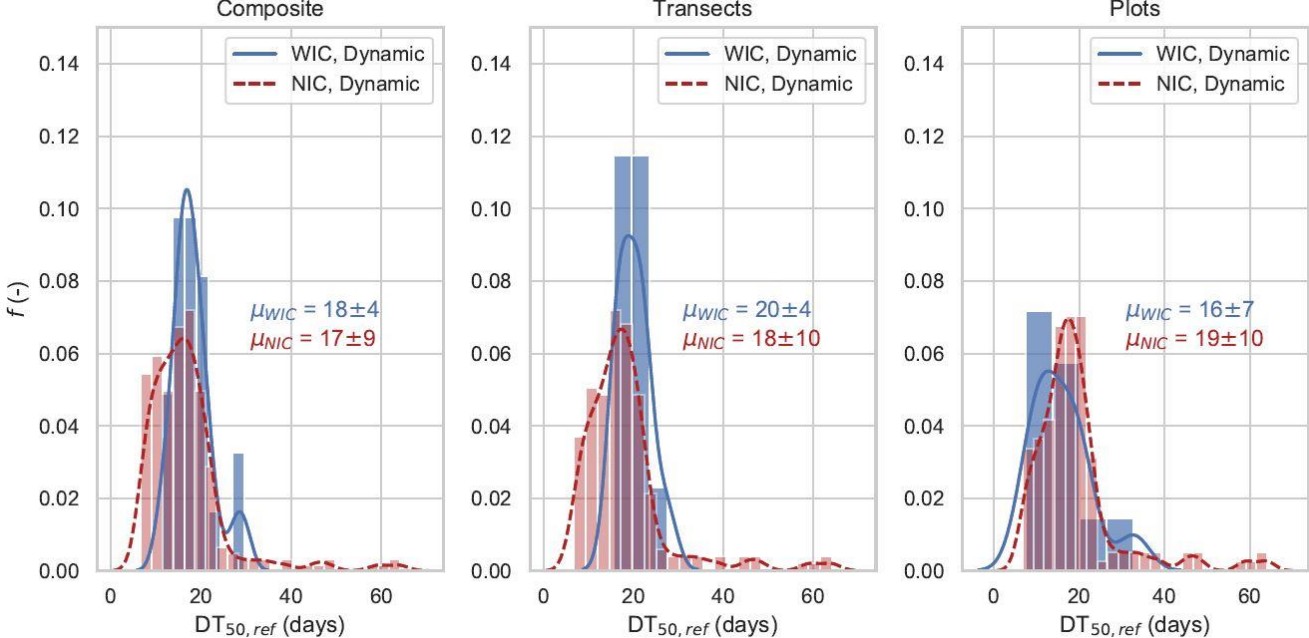

**Figure 4:** Distribution (out of a total of n = 2,500 runs) of $DT_{50}$ calibrated with no isotope constraint (NIC, n = 672) and with isotope constraint (WIC, n = 244) at three sampling resolutions (i.e., composite transect, transect and plot soils). NIC models considered $KGE_{SM} > 0.5$ and $KGE_Q > 0.5$, while WIC models considered $KGE_{SM} > 0.5$ and $KGE_Q > 0.5$ and $KGE_\delta > 0.8$. Statistics for $DT_{50,Ref}$ distributions are provided as mean (solid line for WIC and dotted line for NIC) and standard deviations ($\mu \pm \sigma$).

Reducing uncertainty associated with the extent of pesticide degradation in soil is necessary because degradation half-lives of a pesticide may span one order of magnitude depending of compounds (Wang et al., 2018), mainly depending on hydro-climatic and soil characteristics. For instance, mean values of $DT_{50,Ref}$ for WIC calibration (< 20 days) and low standard deviations ($\sigma$ <7 days; Fig. 4) suggested that aerobic degradation of S-metolachlor ($DT_{50}$ = 14 - 21 days (Lewis et al., 2016) prevailed in Alteckendorf topsoil, whereas anaerobic degradation ($DT_{50}$ = 23 - 62 days (Seybold et al., 2001; Long et al., 2014) was less significant.

Comparison of calibrated mean values of $DT_{50,Ref}$ for the transect composite soil, the transects across the catchment and the plot resolutions showed small differences (<4 days; Fig. 4). A similar reduction of the uncertainty (i.e., standard deviation twice smaller Fig. 4) was observed with WIC compared to NIC for the transect composite soil, the transects and the plot. This indicates that more detailed spatial soil sampling (i.e., plots > transects > transect composite sample) did not provide additional value to constrain model calibration for this catchment. Hence, pooling transect samples into one composite soil sample for CSIA may be representative and sufficient to constrain degradation parameters while reducing sampling and analytical efforts.





In addition, incorporation of CSIA in PiBEACH allowed determining the carbon isotope fractionation value ($\varepsilon_C$) for S-metolachlor degradation in topsoil. Modelling results yielded $\varepsilon_C = -2.7 \pm 0.6$ ‰, in agreement with the prevailing

transformation pathway suggested for this catchment (Alvarez-Zaldivar et al., 2018). Although the $\varepsilon_C$ value used in this previous study ($\varepsilon_C = -1.5 \pm 0.5$ ‰) may have slightly overestimated degradation extent, it remained within uncertainty ranges obtained in the present study from the model ensemble.

### 3.3 Metrics of pesticide persistence and transport risks

Six metrics were obtained from the PiBEACH outputs to evaluate the S-metolachlor persistence in soil and transport risk by leaching and runoff from topsoil across the catchment. The metrics were expressed as percentage of applied S-metolachlor and included (i) S-metolachlor degradation (Fig. 5A), (ii) remaining bioavailable mass of S-metolachlor (*BAM*) in topsoil (i.e. dissolved and reversibly-sorbed S-metolachlor; Fig. 5B), (iii) remaining mass of irreversibly sorbed S-metolachlor (*aged*; Fig. 5E), (iv) off-site transport of S-metolachlor by leaching (Fig. 5D), (v) S-metolachlor volatilization (Fig. 5F), and (vi) S-

metolachlor export to the catchment outlet by runoff and drainage (Fig. 5C), from the first application day on March 19th (day 0) to July 12th, 2016 (day 115). For two metrics, i.e., degradation and remaining mass in topsoil, PiBEACH estimates were similar to observations (Fig. 5A, 5B and section 4 of SI). The degradation extent obtained with the model and derived from the observed isotope measure on transect composite soil and the Rayleigh equation (section 5 of SI) were similar with $72 \pm 13$ % and $68 \pm 1$ %, respectively (day 87, corresponding to last quantifiable soil isotope measure, Fig. 5A). The remaining

dissolved and reversibly sorbed, i.e. bioavailable mass, of S-metolachlor (Remaining *BAM* %) were estimated from measured soil concentrations and transect areas (Fig. 1 and S1). The extrapolated remaining mass of S-metolachlor from transect composite soil ($12 \pm 8$ %) fell within the uncertainty range of the predicted values on the last measurement day (day 115, at $18 \pm 3$ %). S-metolachlor export to the catchment outlet ($2 \pm 6$ %) estimated with CSIA data was higher than that observed from March 19th to July 12th, 2016 ($0.5 \pm 0.1$ %), but within the uncertainty range. This latter metric depends on the ability of

PiBEACH to reproduce daily discharges. Although PiBEACH was primarily developed to initialize sub-hourly event-based distributed models, daily discharge dynamics were correctly simulated (Fig. 3B) as observed previously in similar hydro-climatic, soil and crop catchment (Sheikh et al., 2009). Finally, the predicted volatilization of S-metolachlor (<1 %) during the first 120 h following application was consistent with the low vapor pressure of S-metolachlor (= 1.7 mPa, Lewis et al., 2016). However, S-metolachlor volatilization ranging from 5 to 63 % of the applied mass has been reported, depending on

meteorological conditions (Gish et al., 2011).








**Figure 5:** Pesticide dissipation processes and associated ensemble mean out of a total of 672 and 244 simulations for NIC (dotted line) and WIC (solid line for WIC), respectively. 95 % confidence intervals (CI) with isotope constraints (WIC) and without isotope constraints (NIC) are depicted in blue and red, respectively. All metrics are reported as percent of applied cumulative masses for the entire catchment after the first day of application on March 19, 2016. Final values of dissipation processes ($\mu_{WIC}$ and $\mu_{NIC} \pm$ 95 % CI) are provided on day 115. Black markers depict estimated amounts derived from mass balances based on farmer surveys and extrapolated to the whole catchment (See SI: Sections S3 and S4).

Altogether, the six metrics used for pesticide transport risk confirmed that CSIA data reduced uncertainty of pesticide degradation estimates and other processes of pesticide dissipation modelled by the PiBEACH model. The comparison of 95 % confidence intervals for the six metrics indicated that the uncertainty at the end of the season (day 115) was two-fold lower in the WIC (with CSIA data) than in the NIC (without) model ensemble. The mean extent of S-metolachlor degradation was similar for both dynamic $DT_{50}$ models at the end of the season (i.e., $\Delta\mu = 1$ %). However, the uncertainty associated with the degradation extent was 60% larger without than with CSIA data. The larger uncertainty of degradation extent for NIC model ensemble propagated across other dissipation processes. For instance, both mean and uncertainties of S-metolachlor leaching were twice higher in the NIC than in the WIC model ensemble.

Our study highlights that even a moderate sampling effort, including CSIA data, such as weekly sampling of mixed topsoil samples across the catchment, can be sufficient to identify hot spots and hot moments of pesticide degradation at the catchment scale. As the field of pesticide CSIA continues to advance, for instance, through the development of multi-element CSIA (Torrentó et al., 2021; Höhener et al., 2022) and the development of passive sampling strategies for CSIA (Gilevska et al., 2022a), more reliable evaluations of pesticide degradation and a better understanding of competing pathways (Elsner and Imfeld, 2016; Hofstetter et al., 2024) will enable more detailed assessments of degradation processes in agricultural soil and at the headwater catchment scale.

## 3 Conclusion

This study addresses the disparity between the complexity that reactive transport models can handle and the available field datasets for their calibration and validation. The representation of the relationship between hydro-climatic factors, biogeochemical conditions, and the extent and pathways of degradation has improved significantly in recent decades. However, it often remains overly simplified in 2D catchment models, as noted by Lutz et al. (2013), or is not explicitly defined in other models (DeMars et al., 2018). To reduce model uncertainty and equifinality when incorporating more complex, physically, or biologically-based representations of degradation across the catchment, additional data are needed to identify valid parameter value ranges. Our findings indicate that relying solely on topsoil pesticide concentration and discharge data may be insufficient

to adequately constrain the complexity of reactive transport models for pesticides in 2D catchment models. We showed that incorporating complementary datasets, such as pesticide CSIA data (Fenner et al., 2013, Elsner and Imfeld, 2016; Hofstetter

et al., 2024), can help distinguish between degradative and non-degradative processes that result in a decrease in pesticide concentrations on the catchment scale. For providing evidence of in situ degradation, relying on pesticide carbon only ($\delta^{13}C$) may be sufficient. This can ultimately help close pesticide mass balances on the catchment scale and validate hydrological models. In the future, pesticide multi-element CSIA, beyond carbon CSIA, has the potential to enhance knowledge on competing pathways (Höhener et al., 2022). To enhance the accuracy of these models, one potential avenue for improvement

is to consider dynamics related to microbial functional diversity, biomass, and biodegradation extent, as suggested by Konig et al. (2018).

Importantly, our study also underscores the potential of topsoil CSIA data to guide pesticide management practices, as it offers more dependable estimates of pesticide degradation and transport at the headwater catchment scale. Given the influence of hydro-climatic factors on pesticide degradation and transport, particularly in the context of global warming with changing

local rainfall and temperature patterns (Barrios et al., 2020), considering these factors is essential. Moreover, the determination of metrics for pesticide risk assessment often requires the use of probability and confidence intervals (CI) (Lutz et al., 2013). Therefore, to support management decisions and ecotoxicological assessments, hydrological modeling approaches should consistently incorporate an uncertainty framework for analyzing the range of parameter values needed to evaluate pesticide transfer risks.

## 4 Code availability

The codes developed for this study are available from the corresponding author upon request.

## 5 Data availability

The data produced for this study are available from the corresponding author upon request.

## 6 Author contributions

PAZ performed conceptualization, data curation, formal analysis, investigation and writing. PVD developed the agronomical module, produced associated results, and performed writing. SP and GI performed conceptualization, funding acquisition, project administration, supervision and writing – review & editing.



## 7 Supplement

The supplement related to this article includes a summary of hydro-climatic conditions, detailed catchment and sampling descriptions, farmer survey results, δ¹³C analysis and detailed model descriptions and is available in assets section.

## 8 Competing interests

The authors declare that they have no conflict of interest.

## 9 Acknowledgements

P. Alvarez-Zaldivar was supported by an Initiatives d'excellence (IDEX) fellowship of the Strasbourg University. This work is funded by the French National research Agency ANR through grant ANR-18-CE04-0004-01, project DECISIVE. The authors thank S. Wisselmann, M. Levasseur, C. Wiegert, and B. Guyot for support in sampling and analysis and R. Coupe for helpful comments on the article's structure and organization. We would also like to thank the Alteckendorf farmers for their collaboration during the field study.

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
