# Peer review of "Constraining pesticide degradation in conceptual distributed catchment models with compound-specific isotope analysis (CSIA)"

_EGUsphere, 2024_

## Referee Comment (RC1)

**General comments**

The study showed that incorporating CSIA reduced uncertainty in PiBEACH model for estimating S-metolachlor concentration in soil. The study is interesting, with a good conclusion and abstract. However, the structure of the article requires MAJOR improvements. Some are mentioned below, but the authors' additional effort is also welcome.

**Specific comments**

**Ln 11**: What do you mean by "the calibration of pesticide dissipation"? Which parameters are related to pesticide dissipation?

**Ln 16** This is confusing; previously, you mentioned "calibration of pesticide dissipation." Typically, you calibrate the parameters of the model. Now you are mentioning "predict daily pesticide dissipation." In this case, you are saying that pesticide dissipation is like an output of the model PIBEACH. Can you explain this? It is very important to understand your research.

**Ln 23**, In the phrase "the incorporation of CSIA data into conceptual distributed hydrological models," you should mention that only for PIBEACH, you did not test more conceptual hydrological models (perhaps the results are different). For this article, it could be beneficial to incorporate one more model.

**Ln 41.** In the phrase *"In addition, existing models frequently fail to accurately quantify the contribution of concomitant pesticide dissipation processes for accurate estimation of off-site pesticide transport (Gassmann et al., 2021)."*

Previously, you provided information on the contribution of processes for the pesticide dissipation process, such as degradation, sorption, and volatilization. I think this term (pesticide dissipation) requires a small paragraph for a definition first because, in the phrase of Ln41, it is not clear what you mean by "the contribution of concomitant pesticide dissipation processes......" Commonly, hydrological models simulate (quantify) "concentrations" over time or "transport"; for example, you can see SWAP-PEARL, HYDRUS, or MACRO models. Therefore, the models do not quantify pesticide dissipation processes as an output; they are mechanisms **IN** the model. When I read about pesticide dissipation, the first thing that came to mind was DT50 (degradation). Therefore, I suggest including a small explanation of the term, changing it, or citing an author explaining the concept.

**Ln 45** In the phrase "data limitations to constrain model parameters," Please include at the end of this paragraph "by calibration". So, the final phrase should read "data

limitations to constrain model parameters by calibration." Remember that several pesticide parameters can be obtained by laboratory analysis.

**Ln 55-57** I think this phrase is key for your purpose. Maybe after this phrase, you should provide your opinion and mention that this is one of the research goals, increasing the understanding of your research.

**Ln 72** Is the incorporation of CSIA an "alternative" or "complementary" approach?

**Ln 138** It mentioned soil samples from 13 marked plots. I GUESS you mean the black dots in Figure 1. We still do not know which samples were obtained from weekly measurements (green dots in Fig. 1). The methodology needs to be organized better.

**Ln 149** I suggest including the total number of observations you have for calibration. You have three types of observations to include in an objective function. Concentrations of S-metolachlor in the topsoil, CSIA, and runoff discharge. It is important to know that number because you are calibrating 25 parameters (which is a lot!) In my experience, you should not calibrate more than five, which still is a lot. Otherwise, you get problems with parameter uncertainties.

**Ln 170**. OK, PiBEACH needs meteorological records, among other things, but it did not provide details of how those measurements were taken or analyzed in laboratory conditions.

**Ln 158** If you mean Supplemental information by "SI", you should put the definition of the acronym one time.

**Ln 178** Can you please mention the mixing topsoil layer length in this phrase? The mixing layer is generally a challenging topic and, in my opinion, an unsolved problem for hydrological models.

**Ln 201** The Model only accepts that number of layers. Or is that set by the user?

**Ln 220** In Eq.1, you mentioned that the balance is solved for each layer with soil depth Z. But the runoff term should not be included in deeper layers. This raises the question of whether you obtain a unique theta per cell or a theta for each layer (5 layers in the profile per cell). It would help if you clarified this, especially for HYDRUS, MACRO, openLISEM, or SWAP-PEARL users.

**Ln 230:** You mentioned the balance at each cell I was given by Eq. 2. My question is, does this model solve the mass balance per layer (5 in your case), or does it generate a mesh with smaller cells? (Connected with the previous question)

**Ln 246**. Where is the mixing layer depth used in Eq. 4? I think you should change **J** by **qx,y** in this equation.

**Ln 296** It is not better to include $h_{ref}$ than $\theta ref$. The water content depends highly on texture; 0.2 cm$^3$ cm$^{-3}$ of water content could be either high or low. Instead, the $h$ does not have that inconvenient.

**Ln 305** OpenLISEM cannot simulate preferential flow yet, to my knowledge. So, this phrase required modification.

**Ln 350** Which threshold did you use to reduce parameters from 43 to 25. Basically, how did you define a parameter as highly sensitive? Additionally, you should mention which outputs were analyzed during sensitivity analysis, i.e., did you include sensitivities to isotope output?

**Ln 370** Where did you previously mention "daily discharge at the outlet"? You mentioned weekly runoff. The observations are unclear, and you need to structure the methodology better.

**Ln 373** OK, you explain that in addition to incorporating CSIA data in the objective function, you will analyze the spatial distribution of this observation. For better understanding, you should use Fig 1 at this point.

**Ln 393** If the graph (Fig 3C), the "y" axis is in DT50, please use DT50 instead of $k_{eg}$ during the discussion. Additionally, I do not know how this graph and analysis (Fig 3C) provide useful information for the research. All that you mentioned in the graph and discussion can be easily obtained from Eq. 8, derived from another hydrological model, so basically, it is something simple and well-known.

**Ln 400** The graph should be improved **(.svg)**. Additionally, you mention the error bars linked to soil bulk density. I think this is important and should be mentioned in the methodology (please improve the organization of methods).

**Ln 404** Please indicate if metolachlor concentrations are in liquid or total concentrations (also, this should be explained in the methodology).

**Ln 420** OK WIC calibration reduced uncertainty, but what happens with the other 24 parameters? In the case of NIC or WIC calibration, do they maintain the uncertainty bounds? Or perhaps it changes, and how? This is important to analyze because that uncertainty may be translated to other parameters.  Additionally, is there some field (laboratory) information on the common parameter values for this compound? especially DT50. I agree that DT50 could be one order of magnitude different from the expected one. Still, it is welcome to have the laboratory reference to see if calibration

did not modify parameters to "unexplainable" values. That also holds for the 24 remaining parameters that you calibrate. That kind of analysis is expected from this kind of research.

**Ln 439**. OK, it would be nice to see if there are significant differences in the comparison variance. The plot situation showed a smaller reduction in the SD. The "plot" may not be recommended, but you need ANDEVA or something like that for that affirmation.

**Ln 444** "Modeling results": Do you mean calibration results? Ec was calibrated? In this case, did you compare it with previous studies on that parameter? Something like that should be expected for DT50 and perhaps other calibrated parameters. (Similar comment that above)

**Ln 457** Why did you mention that degradation obtained with the model was like observations? In which part of the methodology did you explain that compound degradation was performed under laboratory conditions or other? You need to improve the structure of the article.

**Ln461** Here, we explain how the BAM% was obtained, but this is a methodological part.

**Techincal comments**

**L10.** to the limited availability of field data **capable of distinguishing** between. Please change by "distinguishing".

**L17.** incorporating changes **of** the carbon isotopic signatures. Please change "of" by "in".

**L36.** leaching, volatilization, off-site transport, Please included **"and"** before off-site.

L46. each of which generally **accounting** for several physico-chemical. Please modify by accounts.

L59. in topsoil and at the catchment outlet **is** currently. Please change "is" by "are".

L70. it may also generate unknow and. Please change "unknow" by "unknown".

And some more that you can check.

---

## Author Comment (AC2)

Dear reviewers,

We warmly thank the two reviewers for their relevant and useful feedback and comments on our paper titled "Constraining pesticide degradation in conceptual distributed catchment models with compound-specific isotope analysis (CSIA)." We have now carefully addressed and incorporated these comments to emphasized the originality and potential of this study.

Please find below the detailed responses and modifications made (in blue) based on the comments, which have been incorporated into the revised version. The line numbers correspond to the revised manuscript, which already includes the following modifications.

Reviewer 1:

**General comments**
The study showed that incorporating CSIA reduced uncertainty in PiBEACH model for estimating S-metolachlor concentration in soil. The study is interesting, with a good conclusion and abstract. However, the structure of the article requires MAJOR improvements. Some are mentioned below, but the authors' additional effort is also welcome.

We thank Reviewer 1 for their valuable comments. We have now reorganized the structure of the article based on these suggestions and added a new section to the Materials and Methods, introducing the six metrics.

**Specific comments**
**Ln 11**: What do you mean by "the calibration of pesticide dissipation"? Which parameters are related to pesticide dissipation?

We have now clarified the sentence pointing the parameters involved in the pesticide dissipation.

*"This limitation complicates the calibration of parameters involved in the pesticide dissipation, e.g., biodegradation and volatilization half times ($DT_{50}$) or partition coefficient ($K_{oc}$) for sorption/desorption, often leading to equifinality. As a result, it hinders the reliable prediction of pesticide persistence in soil and its transportation from agricultural plots to the catchment outlet."*

**Ln 16** This is confusing; previously, you mentioned "calibration of pesticide dissipation." Typically, you calibrate the parameters of the model. Now you are mentioning "predict daily pesticide dissipation." In this case, you are saying that pesticide dissipation is like an output of the model PIBEACH. Can you explain this? It is very important to understand your research.

Thank you for highlighting this confusion. We have now clarified this issue. Pesticide dissipation in soil results from both degradation processes (biodegradation, photolysis, hydrolysis) and transport between compartments (soil to air via volatilization, soil to surface water via runoff, or soil to groundwater via leaching). Therefore, pesticide dissipation is not a direct output of the model, such as the pesticide concentration at the catchment outlet, but can be inferred from the model's internal compartments and fluxes. However, a mass balance

of pesticide fate can be calculated at each time step, allowing for the prediction of both overall pesticide dissipation and the specific fraction that is definitively degraded. The novelty of the PiBEACH model lies in its ability to refine this prediction by distinguishing between degradative and non-degradative processes, thereby reducing uncertainty.

We have now clarified the sentence to point this novelty:

*"The research was conducted in a 47-ha crop catchment, focusing on the widely used pre-emergence herbicide S-metolachlor. A novel conceptual model, named PIBEACH, was developed to predict pesticide dissipation in soil and its transport to rivers, reducing equifinality and associated uncertainty on the extent of the different dissipation processes by incorporating changes of the carbon isotopic signatures ($\delta^{13}C$) of the targeted pesticide during degradation."*

**Ln 23**, In the phrase "the incorporation of CSIA data into conceptual distributed hydrological models," you should mention that only for PIBEACH, you did not test more conceptual hydrological models (perhaps the results are different). For this article, it could be beneficial to incorporate one more model.

We have revised the following sentence to express that we focused on the PiBEACH model. We agreed to explore the integration of isotopic fractionation in other models in future work and this study highlights the integration of CSIA data in PIBEACH. Therefore, the primary goal of this paper was to lay the groundwork for applying a similar approach to other pesticide models, highlighting the benefits, limitations, and additional field data required, rather than focusing on comparing the added value of CSIA across different models, which is beyond the scope of the current study. This has now been specified:

*"In summary, the incorporation of CSIA data into the conceptual distributed hydrological model PiBEACH holds the potential to alleviate parameter equifinality, therewith significantly improving our ability to predict the dynamics of pesticide degradation on the catchment scale. With concentration of pesticide in topsoil distributed across the catchment and throughout the crop growing season, PiBEACH paves the way for coupling with distributed event-based models, such as OpenLISEM OLP, which are better suited to address the intra-event dynamics of pesticide transport."*

**Ln 41.** In the phrase *"In addition, existing models frequently fail to accurately quantify the contribution of concomitant pesticide dissipation processes for accurate estimation of off-site pesticide transport (Gassmann et al., 2021)."*
Previously, you provided information on the contribution of processes for the pesticide dissipation process, such as degradation, sorption, and volatilization. I think this term (pesticide dissipation) requires a small paragraph for a definition first because, in the phrase of Ln41, it is not clear what you mean by "the contribution of concomitant pesticide dissipation processes……" Commonly, hydrological models simulate (quantify) "concentrations" over time or "transport"; for example, you can see SWAP-PEARL, HYDRUS, or MACRO models. Therefore, the models do not quantify pesticide dissipation processes as an output; they are mechanisms **IN** the model. When I read about pesticide dissipation, the first thing that came to mind was DT50 (degradation). Therefore,

I suggest including a small explanation of the term, changing it, or citing an author explaining the concept.

We have now revised the sentences to address this comment and clarify how the reactive transport of pesticides, encompassing various processes, leads to pesticide dissipation at the catchment scale, i.e., a reduction in pesticide concentration in the soil compartment.

*While pressure on aquatic ecosystems is increasing, the contribution of each pesticide dissipation processes in soil driving to a mass decrease and pesticide off-site transport toward the catchment outlet remains difficult to evaluate and predict accurately. In this context, reliable and validated hydrological models that predict pesticide dissipation and off-site transport have the potential to address fundamental questions, such as the relationship between hydro-climatic factors and pesticide dissipation processes and the transfer risks of pesticides on the catchment scale. The contribution of degradative processes, including both biotic or abiotic degradation, as well as non-degradative pesticide dissipation processes, such as sorption, leaching, volatilization, off-site transport, must be quantified for this purpose in models (Larsbo and Jarvis, 2005; Steffens et al., 2015; Gassmann, 2021).*

**Ln 45** In the phrase "data limitations to constrain model parameters," Please include at the end of this paragraph "by calibration". So, the final phrase should read "data limitations to constrain model parameters by calibration." Remember that several pesticide parameters can be obtained by laboratory analysis.
We have now revised the sentence as suggested

**Ln 55-57** I think this phrase is key for your purpose. Maybe after this phrase, you should provide your opinion and mention that this is one of the research goals, increasing the understanding of your research.
We have added a final sentence in this section to address the challenge for the modelling community.

*Consequently, reducing the uncertainty of $DT_{50}$ values remains a challenge at the catchment scale, even though it is a priority for enhancing the predictive capacity of transport models.*

**Ln 72** Is the incorporation of CSIA an "alternative" or "complementary" approach?
We have revised the sentence as suggested:

*In this context, pesticide compound specific isotope analysis (CSIA) offers an complementary approach because information on parent pesticide degradation is derived independently from concentrations and produced transformation products, and relies on stable isotope ratios (Elsner and Imfeld, 2016; Hofstetter et al., 2024).*

**Ln 138** It mentioned soil samples from 13 marked plots. I GUESS you mean the black dots in Figure 1. We still do not know which samples were obtained from weekly measurements (green dots in Fig. 1). The methodology needs to be organized better.

We have now revised the caption of the Figure 1, and the following sentence, to explain the weekly weighted samples of topsoil for each transect from regular sampling (green dots) along transects (see Alvarez-Zaldivar et al., 2018).

*Figure 1: The headwater catchment of Alteckendorf (Bas-Rhin, France), illustrating the experimental setup, which includes transects (weighted samples collected at the green dots along the red transects) and plot sampling (black dots), as well as the land use for 2016. The "Other" category encompasses roads, grass strips, and orchards.*

*Lines 146-148*

*Soil samples were collected from 13 marked plots (before S-metolachlor applications and on days 1, 50, and 100 post-application), as well as from the north, valley, and south transects across the catchment (weekly weighted samples from the green dots in Fig. 1, spanning from March 19th, i.e., day 0, to July 12th, i.e., day 115). These samples were analyzed for the quantification and CSIA ($\delta^{13}$C) of S-metolachlor in soil*

**Ln 149** I suggest including the total number of observations you have for calibration. You have three types of observations to include in an objective function. Concentrations of S-metolachlor in the topsoil, CSIA, and runoff discharge. It is important to know that number because you are calibrating 25 parameters (which is a lot!) In my experience, you should not calibrate more than five, which still is a lot. Otherwise, you get problems with parameter uncertainties.

Thank you for the suggestion. We have mobilized a total of 103 topsoil samples (i.e., 13 plots × 4 sampling dates + 3 transects × 17 weeks) to measure S-metolachlor concentration and $\delta^{13}$C values across the catchment and throughout the crop growing season. This effort was aimed at constraining the topsoil dissipation of S-metolachlor, as well as 115 daily discharge measurements to assess the overall hydrological response of the catchment.

The S-metolachlor concentrations and associated $\delta^{13}$C values monitored at the catchment outlet were not incorporated into the calibration process, as PiBEACH, using daily rainfall data, was not designed to simulate accurate sub-daily intra-event runoff, which accounts for a significant portion of seasonal export (Lefrancq et al., 2012). However, the S-metolachlor concentrations at the outlet, derived from 51 samples, were used for external validation to assess the limitations of PiBEACH, particularly in terms of S-metolachlor export at the outlet (Panel C, Figure 5, and the new section in Materials and Methods). We recognize that calibrating 25 parameters is a considerable undertaking, which justifies the dataset collected across the catchment and throughout the growing season, including the original use of isotopic signatures. We have now addressed the issue of parameter uncertainty with a rigorous methodology, which allowed us to accurately associate our results with the corresponding confidence intervals, as shown in Figures 3, 4, and 5.

We added this information in the manuscript (Line 447):

Calibrating 25 parameters is challenging and justifies the extensive dataset collected across the catchment and throughout the growing season. This unique dataset included the use of isotopic signatures and comprises a total of 103 topsoil samples analyzed for S-metolachlor concentration and $\delta^{13}$C, 115 daily discharge measurements, and 51 river outlet samples with S-metolachlor concentration.

**Ln 170**. OK, PiBEACH needs meteorological records, among other things, but it did not provide details of how those measurements were taken or analyzed in laboratory conditions.

We acknowledge the reviewer for this useful comment, and we have now added the following sentences.

In section Field site - Lines 148-150
*Daily temperature and potential evapotranspiration were obtained from the MeteoFrance station in Waltenheim sur Zorn located 7 km southwest. Daily rainfall was derived from a tipping bucket rain gauge (Précis Mécanique®, Fig. 1).*

In section PiBEACH development - line 205
*BEACH was fed by daily meteorological records, including rainfall and mean air temperature and potential evaporation, [...].*

**Ln 158** If you mean Supplemental information by "SI", you should put the definition of the acronym one time.

Indeed, thank you. We now use "Supplement" without an acronym, as recommended in the HESS guidelines.

**Ln 178** Can you please mention the mixing topsoil layer length in this phrase? The mixing layer is generally a challenging topic and, in my opinion, an unsolved problem for hydrological models.

This information (first centimeter, from 0 to 1 cm) was originally provided 25 lines after the line 178. We have now included this information as recommended in line 214:

*Compared with BEACH features, PiBEACH includes (i) a mixing topsoil layer (from 0 to 1 cm, McGrath et al., 2008) and additional deepest soil layers to include the shallow groundwater contribution to discharge, (ii) daily changes of topsoil temperature (Neitsch et al., 2011), (iii) daily variations of topsoil hydraulic properties accounting for the impact of agricultural practices with a crop-specific agronomical model (Lefrancq et al., 2018), (iv) pesticides first-order degradation and linear sorption/desorption processes, and (v) pesticide carbon stable isotopic fractionation and transport of isotopologues (e.g., $^{13}C$ and $^{12}C$) (Lutz et al., 2017; Alvarez-Zaldivar et al., 2018). PiBEACH requires amounts and dates of pesticide application for each plot.*

As mentioned by the reviewer, the mixing layer concept is probably the most commonly used in pesticide fate models due to its numerical simplicity and its realistic representation, as hortonian runoff processes have repeatedly been shown to be initiated as a near-surface process (Gao et al., 2004; McGrath et al., 2008; Wallender et al., 2008). This concept assumes that the transport is controlled by a mixing layer below the soil surface, in which rainwater, soil solution and runoff water completely and instantaneously mix (Havis et al., 1992). The soil depth interacting with runoff water is variable ranging from 0.25 to 2 cm according to various experimental and modelling studies (Ahuja et al., 1981; Havis et al., 1992; Wallender et al., 2008). Previous studies have shown that the effective depth of interaction is related to the degree of soil aggregation and increases with soil slope, kinetic energy of raindrops and rainfall

intensity (Havis et al., 1992). In fact, the observed mixing-layer depth is often much shallower than the depth required for fitting models to field data (Gao et al., 2004).

Ahuja, L.R., Sharpley, A.N., Yamamoto, M., Menzel, R.G., 1981. The Depth of Rainfall-Runoff-Soil Interaction as Determined by P-32. Water Resources Research. 17, 969-974.

Gao, B., Walter, M.T., Steenhuis, T.S., Hogarth, W.L., Parlange, J.Y., 2004. Rainfall induced chemical transport from soil to runoff: theory and experiments. Journal of Hydrology. 295, 291-304.

McGrath, G.S., Hinz, C., Sivapalan, M., 2008. Modeling the effect of rainfall intermittency on the variability of solute persistence at the soil surface. Water Resources Research. 44.

Wallender, W.W., Joyce, B.A., Ginn, T.R., 2008. Modeling the Transport of Spray-Applied Pesticides from Fields with Vegetative Cover. Transactions of the Asabe. 51, 1963-1976.

Havis, R.N., Smith, R.E., Adrian, D.D., 1992. Partitioning Solute Transport between Infiltration and Overland-Flow under Rainfall. Water Resources Research. 28, 2569-2580.

The reference (McGrath et al., 2008) cited at the first mention of the mixing layer in our paper was included to provide a more in-depth understanding of this concept.

Line 214:

*Compared with BEACH features, PiBEACH includes (i) a mixing topsoil layer (assuming that the pesticide transport is controlled by a mixing layer from 0 to 1 cm below the soil surface, in which rainwater, soil solution and runoff water completely and instantaneously mix, McGrath et al., 2008) and additional deepest soil layers to include the shallow groundwater contribution to discharge.*

**Ln 201** The Model only accepts that number of layers. Or is that set by the user?

We relied on the pedological, agronomic, and hydrological knowledge of the studied catchment to develop the conceptual framework for PiBEACH. The number of layers chosen, i.e. five for the Alteckendorf catchment, seems to be the optimal number to simplify vertical heterogeneity based on agronomic criteria (cropping operations), pesticide reactivity, or hydrological functioning with variably saturated and saturated zones. Based on similar expertise, this number of layers could be adjusted for another catchment and implemented in a modified version of PiBEACH.

**Ln 220** In Eq.1, you mentioned that the balance is solved for each layer with soil depth Z. But the runoff term should not be included in deeper layers. This raises the question of whether you obtain a unique theta per cell or a theta for each layer (5 layers in the profile per cell). It would help if you clarified this, especially for HYDRUS, MACRO, openLISEM, or SWAP-PEARL users.

As noted in line 234 and in Equation 1, PiBEACH calculates the hydrological balance for each cell and layer. The hydrological balance is described in Sheikh et al. (2009), with a detailed implementation provided for each process in Section 6 of the SI. In general, the change in soil water content ($\theta$, m³ m⁻³) for each cell (i) and each layer, defined by soil depth (Z, mm), is calculated using Equation 1. The resulting soil water content is then available for each cell and layer, facilitating potential coupling with other models such as HYDRUS, MACRO, OpenLISEM OLP, or SWAP-PEARL.

We have moved Equation 1 to follow the description of the layers (line 255) in order to clarify how each term in the equation is applied depending on the specific layer. As noted by the

reviewer, we have added a sentence explaining that runoff (RO) is only computed for the topsoil layer ($z_0$).

Line 257:

*"Runoff (RO) is calculated exclusively for the topsoil layer ($z_0$).*"

**Ln 230:** You mentioned the balance at each cell I was given by Eq. 2. My question is, does this model solve the mass balance per layer (5 in your case), or does it generate a mesh with smaller cells? (Connected with the previous question)

As stated in line 212, equations 1 and 2 are solved for each cell of each layer z0 to z3 using a raster approach with a resolution of 2 x 2 m for layers z0 to z3 (four layers in total). The deeper layer, z4, is represented using a global linear reservoir.

The sentence was revised to clarify the application of the raster approach with a 2 x2 m resolution for solving the equations.

*To integrate landscape components enhancing or limiting pesticide off-site transport, i.e., grass strips or roads, the Alteckendorf catchment was modelled using a 2 × 2 m resolution raster. The vertical soil profile was represented by five successive layers including, from topsoil to groundwater bottom (Fig. 2):*

**Ln 246**. Where is the mixing layer depth used in Eq. 4? I think you should change *J* by *$q_{x,y}$* in this equation.

The equation 4 has now been corrected, and the depth of the mixing layer is now explicitly linked to $D_{z0}$.

Line 269:

*"where RO was runoff (mm) and $\beta_{RO}$ was a calibration constant ($1 \geq \beta_{RO} \geq 0$) and $D_{z0}$ (mm) was the mixing layer  depth (Ahuja and Lehman, 1983)."*

**Ln 296** It is not better to include $h_{ref}$ than $\theta ref$. The water content depends highly on texture; 0.2 $cm_3$ $cm_{-3}$ of water content could be either high or low. Instead, the *h* does not have that inconvenient.

We acknowledge this suggestion and we agree that water content depends on soil texture. However, we have retained and applied in this study the formulation developed by Walker (1974) and Larsbo and Jarvis (2003), which uses a ratio of soil moisture expressed in θ ($m^3$ $m^{-3}$) rather than h (mm $H_2O$). The reference value, $\theta_{ref}$, corresponds to the experimental conditions associated with the $DT_{50}ref$ (and $k_{ref}$) characterization. In these lab-scale experiments, soil saturation is typically expressed in terms of soil moisture.

**Ln 305** OpenLISEM cannot simulate preferential flow yet, to my knowledge. So, this phrase required modification.

We agree with the reviewer's assessment. OpenLISEM and its pesticide module OLP are not capable of simulating preferential flow and are better suited for catchments with a significant risk of runoff and associated erosion, such as in Alteckendorf during storm events. As a result, we have revised the sentence to focus solely on runoff.

Line 363:
Considering a daily time step, PiBEACH was not designed to accurately simulate  runoff genesis and dynamic across the catchment  (Sheikh et al., 2009).

**Ln 350** Which threshold did you use to reduce parameters from 43 to 25. Basically, how did you define a parameter as highly sensitive? Additionally, you should mention which outputs were analyzed during sensitivity analysis, i.e., did you include sensitivities to isotope output?

We have calculated and analyzed the mean and standard deviation of the elementary effects (EE) for each parameter, as required by the Morris method. The mean value represents the overall effect of each parameter on the model output, while the standard deviation reflects the interaction between inputs. We excluded 21 parameters where the mean value was zero or close to zero, indicating no significant influence on the output. The outputs considered in this sensitivity analysis included S-metolachlor concentrations at the outlet, composite soil transects (North, Valley, and South), discharge at the outlet, and isotope signatures at both the outlet and composite soil transects. The mean and standard deviation values for the 25 retained parameters are presented in Figures S5 and S6 in the Supplement, for the different output variables.

**Ln 370** Where did you previously mention "daily discharge at the outlet"? You mentioned weekly runoff. The observations are unclear, and you need to structure the methodology better.

As described in a previous paper (Alvarez-Zaldivar et al., 2018), *"discharge at the catchment outlet was continuously measured by a Doppler flow-meter (2150 Isco, Lincoln, Nebraska, USA) with discharge precision of 3% and a time step of 2 minutes. Water was collected by flow proportional sampling using a refrigerated automatic sampler with a total capacity of 3.96 L divided into 12 glass vials each of 330 mL (Isco Avalanche, Lincoln, Nebraska, USA). A predefined discharged volume based on the seasonal rainfall intensity expected for April (50 $m^3$), May (100 $m^3$) and June (150 $m^3$) was chosen allowing for 36 aliquots of 110mL each per week. Water samples were then combined into composite samples according to hydrograph characteristics (base-flow, rising and/or falling limb), yielding one to four samples weekly of volumes >= 990 mL. "*

The weekly time step refers only to the field visits for collecting and replacing samples, not to the sampling frequency used to acquire data. We have revised the relevant sentences to clarify this distinction.

*Line 175:*

*Outlet runoff discharge was measured using a Doppler flowmeter (2150 Isco) with a precision of 3% and a time step of 2 minutes. Automatic, refrigerated continuous flow proportional sampling (Isco Avalanche with twelve bottles of 330 mL) was conducted at fixed weekly discharge volumes ranging from 50 to 150 m³ from April to June to capture increasing minimum baseflow discharges observed in 2016 (Alvarez-Zaldivar et al., 2018). To obtain sufficient amount of S-metolachlor for quantification and CSIA, samples were combined weekly in composite samples based on hydrograph, i.e., base-flow, rising and falling limb, yielding one to four samples weekly of volumes ≥ 990 mL (Alvarez-Zaldivar et al., 2018).*

**Ln 373** OK, you explain that in addition to incorporating CSIA data in the objective function, you will analyze the spatial distribution of this observation. For better understanding, you should use Fig 1 at this point.

As suggested by the reviewer, we have revised the relevant sentence to clarify this point.

Lines 436:

*For the latter, KGE metric incorporates topsoil S-metolachlor concentration and $\delta^{13}C$ from (i) individual plot observations (13 black dots in Fig. 1), (ii) aggregated plot observation along three transects across the catchment (North, Valley and South transect, Fig. 1) and (iii) transect composite soil (green dots, Fig. 1).*

**Ln 393** If the graph (Fig 3C), the "γ" axis is in DT50, please use DT50 instead of $k_{eg}$ during the discussion. Additionally, I do not know how this graph and analysis (Fig 3C) provide useful information for the research. All that you mentioned in the graph and discussion can be easily obtained from Eq. 8, derived from another hydrological model, so basically, it is something simple and well-known.

As suggested by the reviewer, we have now revised the relevant sentence, using $DT_{50}$ instead of $k$ to discuss the temporal variation in S-metolachlor persistence.

Lines 476:

*Predicted low S-metolachlor degradation rates in the first 10 days of observation ($DT_{50} = 46 \pm 17$ days) corresponded to an early cold (7.1 ± 1.7 °C) and dry period (rainfall of 7.4 mm) (Fig. 3A). In contrast, increasing degradation rates ($DT_{50} = 22 \pm 2$ days) from 50 to 120 days following the first application of S-metolachlor were mainly associated with higher temperatures (16.9 ± 3.6 T°C).*

**Ln 400** The graph should be improved (**.svg**). Additionally, you mention the error bars linked to soil bulk density. I think this is important and should be mentioned in the methodology (please improve the organization of methods).

The figures 3, 4, 5 will be send in HD in the final version.

As stated in the caption of Figure 3, both observed and predicted soil water content are expressed in m³ water m⁻³ soil. Additional details have been provided in the Materials and Methods section, explaining how gravimetric measurements obtained in the laboratory (drying samples in an oven set at 110°C, NF ISO 1146) were converted into volumetric values using soil bulk density. The topsoil bulk density fluctuates throughout the growing season and was estimated daily based on the reference soil bulk density measured below the plow layer. (1.5 ± 0.09 kg m⁻³) and the agronomic module of PiBEACH (Section 7.2 in the Supplement), which was previously validated in Lefrancq et al. (2018, HP).

This information was added in line 170:
*Volumetric topsoil water content ($m^3_{water}$ $m^{-3}_{soil}$) was derived from gravimetric water content after drying the samples in an oven set at 110°C (NF ISO 1146; Lefrancq et al., 2018). This calculation accounted for the seasonal dynamics of topsoil bulk density, as predicted by PiBEACH and detailed in the Supplement (section 7.2).*

In caption of Figure 3 we have revised the targeted sentence:

*The associated error bars are linked to soil bulk density estimations required for gravimetric to volumetric conversions and from the standard error on gravimetric water content measurements.*

**Ln 404** Please indicate if metolachlor concentrations are in liquid or total concentrations (also, this should be explained in the methodology).

We agree with the reviewer's that information on S-metolachlor concentration was missing. Although the filtration methodology was already described (see below and in line 183 of the manuscript), we have now clarified the methodology for river samples and added a sentence (Line 557) in the results section to clarify this point. S-metolachlor concentrations in the particulate phase (i.e., > 0.7 μm) collected in the river were consistently below the quantification limit. Therefore, S-metolachlor concentrations used in the calibration step for the 2016 campaign were based solely on the dissolved phase of the samples collected in the river at the outlet of the catchment.

Line 183:
*To separate S-metolachlor dissolved and particulate phases, river water samples were filtered through 0.7 μm glass fibre filters. Methods of S-metolachlor extraction from soil and water samples with an AutoTrace 280 Solid Phase Extraction (SPE) system (Dionex) with SolEx C18 cartridges (Dionex ®) and quantification with GC-MS (MS, ISQ™, Thermo Fisher 173 Scientific) were detailed previously (Alvarez-Zaldivar et al., 2018; Gilevska et al., 2022b). Quantification limits for S-metolachlor were 0.01 μg L⁻¹ from water and 0.001 μg g⁻¹ from soil and total suspended matter samples (d.w.), with a total analytical uncertainty of 8 % and 16 %, respectively.*

New sentence (line 557)

*It should be noted that the S-metolachlor concentrations used to estimate observed S-metolachlor export in river at the outlet of the catchment were based solely on the dissolved phase of river samples. S-metolachlor concentrations in the particulate phase (i.e., > 0.7 μm) of these samples were consistently below the limit of quantification.*

**Ln 420** OK WIC calibration reduced uncertainty, but what happens with the other 24 parameters? In the case of NIC or WIC calibration, do they maintain the uncertainty bounds? Or perhaps it changes, and how? This is important to analyze because that uncertainty may be translated to other parameters. Additionally, is there some field (laboratory) information on the common parameter values for this compound? especially DT50. I agree that DT50 could be one order of magnitude different from the expected one. Still, it is welcome to have the laboratory reference to see if calibration did not modify parameters to "unexplainable" values. That also holds for the 24 remaining parameters that you calibrate. That kind of analysis is expected from this kind of research.

We hypothesized that $DT_{50, reference}$ was the primary parameter affected by incorporating stable isotope signatures, as it is the sole parameter governing degradation in PiBEACH. As expected, $DT_{50,reference}$ was affected. Specifically, its uncertainty bounds were reduced when $\delta^{13}C$ was included in the objective function (WIC calibration). The newly calibrated WIC range for $DT_{50, reference}$ (18 ± 4 days) remained consistent with reported experimental values in the literature. In contrast, the uncertainty bounds for the other 24 parameters were not significantly altered when comparing NIC and WIC calibrations (data not shown in the current version).

In response to the Reviewer's comment, we have now added a new section in the Supplement (Section 12). This includes a new Figure S8, demonstrating, using $K_{OC}$ as an example, that CSIA data (WIC) did not reduce the uncertainty of 24 other parameters compared to NIC. This may be due to the pesticide extraction protocols used for soil samples, which account for both the dissolved phase and the extractable sorbed phase. Additionally, sorption generally does not lead to significant isotope fractionation (see also the response to Questions 2 and 3 in the Introduction, addressed by Reviewer 2). In this case, CSIA may not help evaluate sorption and thus constrain $K_{OC}$ values as isotope signatures for pesticides in the dissolved and particulate phase of soil samples are generally similar.

[Figure]

*Figure S8: Distribution (out of a total of n = 2,500 runs) of $K_{OC}$ calibrated with no isotope constraint (NIC, n = 672) and with isotope constraint (WIC, n = 244) at three sampling resolutions (i.e., composite transect, transect and plot soils). NIC models considered $KGE_{SM} > 0.5$ and $KGE_Q > 0.5$, while WIC models considered $KGE_{SM} > 0.5$ and $KGE_Q > 0.5$ and $KGE_\delta > 0.8$. Statistics for $K_{OC}$ distributions are provided as mean (blue for WIC, purple and red for NIC, with $DT_{50dynamic}$ depending of soil moisture and temperature or $DT_{50\ constant}$, respectively) and standard deviations ($\mu \pm SD$).*

**Ln 439**. OK, it would be nice to see if there are significant differences in the comparison variance. The plot situation showed a smaller reduction in the SD. The "plot" may not be recommended, but you need ANDEVA or something like that for that affirmation.

We agree with the reviewer's observation. The $DT_{50}$ distributions calibrated with and without isotopic constraints were similar at the plot resolution but differed at the transect and transect composite sample resolutions. We have now entirely revised this statement to avoid any affirmation.

Line 521:

*The mean calibrated $DT_{50,Ref}$ values for transect composite soil, transects across the catchment, and plot resolutions differed by less than four days (Fig. 4). However, the uncertainties, as indicated by standard deviations, were significantly lower—approximately half—with isotopic constraints (WIC) compared to no isotopic constraints (NIC) for transect composite soil and transects, while they remained unchanged at the plot resolution. This indicates that higher-resolution soil sampling (plots > transects > transect composite samples) did not improve significantly model calibration in this catchment. Therefore, pooling transect samples into a single composite soil sample for CSIA appears sufficient to constrain degradation parameters, while also reducing sampling and analytical efforts. This approach may effectively capture temporal variations in pesticide degradation at the catchment scale.*

**Ln 444** "Modeling results": Do you mean calibration results? Ec was calibrated? In this case, did you compare it with previous studies on that parameter? Something like that should be expected for DT50 and perhaps other calibrated parameters. (Similar comment that above)

We have now revised the sentence to clarify that $\varepsilon_c$ was indeed one of the 24 calibrated parameters, and the calibrating value is closed to the experimental reference derived from lab-scale study on the Alteckendorf soil (Alvarez-Zaldivar et al., 2018). In response to the Reviewer's question (see Point 16 in Reviewer 2's section), we have now clarified the different values of the isotope fractionation factor ($\varepsilon_C$) used in this study.

Line 536:

*Additionally, the incorporation of CSIA into PiBEACH enabled the calibration of the carbon isotope fractionation value ($\varepsilon_C$) for S-metolachlor degradation in topsoil. The calibration yielded $\varepsilon_C = -2.7 \pm 0.6$ ‰, which aligns with the prevailing transformation pathway suggested for this catchment (Alvarez-Zaldivar et al., 2018). Although the $\varepsilon_C$ value used in this previous study ($\varepsilon_C = -1.5 \pm 0.5$ ‰) may have slightly overestimated degradation extent, it remained within uncertainty ranges obtained in the present study from the model ensemble.*

**Ln 457** Why did you mention that degradation obtained with the model was like observations? In which part of the methodology did you explain that compound degradation was performed under laboratory conditions or other? You need to improve the structure of the article.

We thank the reviewer for their comments. We have now carefully checked and reorganized the structure of the article by adding a new section to the Materials and Methods, which introduces the six metrics. Additionally, we have now provided a detailed explanation of how the observed values, i.e. specifically topsoil concentration, $\delta^{13}C$, and discharge and concentration in river at the outlet, were used to derive the observed values for three of the six metrics.

Line 449:

**2.7 Metrics of pesticide persistence and transport risks**

*Six metrics were derived from PiBEACH outputs to assess the persistence and transport risk of S-metolachlor at the catchment scale. These metrics were expressed as percentages of the applied S-metolachlor and included: (1) S-metolachlor degradation, (2) the remaining bioavailable mass of S-metolachlor (BAM) in the topsoil (i.e., dissolved and reversibly sorbed S-metolachlor), (3) S-metolachlor export to the catchment outlet via runoff and drainage. The metrics 1 to 3 were derived from observations to evaluate the predictability of PiBEACH. Specifically, the observed remaining bioavailable mass of S-metolachlor (BAM) in the topsoil was calculated through spatial extrapolation of weekly topsoil samples. The observed S-metolachlor degradation was estimated using the mean $\delta^{13}C$ derived from the same spatial extrapolation, combined with the enrichment factor ($\varepsilon$) obtained from microcosm lab-scale experiments using soil from Alteckendorf. Finally, S-metolachlor export to the catchment outlet was calculated by integrating continuous discharge measurements with sub-weekly S-metolachlor concentration samples from the river. (4) off-site transport of S-metolachlor*

*through leaching, (5) the remaining mass of irreversibly sorbed S-metolachlor (i.e., aged S-metolachlor), and (6) S-metolachlor volatilization, from the first application on March 19 (day 0) to July 12, 2016 (day 115). The metrics 4 to 6 remained difficult to estimate using the catchment-scale sampling campaign and were not computed.*

**Ln461** Here, we explain how the BAM% was obtained, but this is a methodological part.
This explanation is now included in the newly added Materials and Methods section.

Line 460:
*Specifically, the observed remaining bioavailable mass of S-metolachlor (BAM) in the topsoil was calculated through spatial extrapolation of weekly topsoil samples*

**Technical comments**
**L10.** to the limited availability of field data **capable of distinguishing** between. Please change by "distinguishing".
As suggested by the reviewer, we have revised the sentence:

Line 11:
*Predicting pesticide dissipation at the catchment scale using hydrological models is often challenging due to the limited availability of field data that can distinguish between degradative and non-degradative processes.*

**L17.** incorporating changes **of** the carbon isotopic signatures. Please change "of" by "in".
we modified the sentence as suggested

**L36.** leaching, volatilization, off-site transport, Please included **"and"** before off-site.
we modified the sentence as suggested

L46. each of which generally **accounting** for several physico-chemical. Please modify by accounts.
we modified the sentence as suggested

L59. in topsoil and at the catchment outlet **is** currently. Please change "is" by "are".
we modified the sentence as suggested

L70. it may also generate unknow and. Please change "unknow" by "unknown".
we modified the sentence as suggested

And some more that you can check.

We have thoroughly revised the manuscript to correct any spelling errors.
* * *
**General comment**

Payraudeau et al. present a novel modelling study advancing the field of simulation of pesticide and isotope ratios at catchment scale. Studies as this one are rarely conducted but very important to understand more on pesticide fate at catchment scale. I have two main concerns that I strongly recommend that the authors take into consideration. These are elaborated below.

We thank Reviewer 2 for the relevant comments. In response, we have now revised in details the manuscript, specifically addressing the two main concerns: (i) the added benefits of this distributed model approach compared to the earlier lumped (non-spatially distributed) model study by Lutz et al. (2017), and (ii) the use of pesticide concentrations and isotope ratios at the catchment outlet.

**Main comments**

1. A key novelty of this study is the application of a spatially distributed catchment model to simulate pesticide concentrations and isotope ratios (only in topsoil unfortunately, see main comment #2), as stated also on line 92. My first main critique to this study is that the added benefit of a spatially distributed model for this purpose is barely demonstrated nor discussed, albeit expectations are raised in lines 87-89. In that sense the added benefit of this distributed model approach compared to the earlier lumped (not spatially-distributed) model study by Lutz et al. (2017) - applied to the same catchment but with an earlier dataset limited to catchment outlet (but no topsoil) data - does not become clear. The focus in the model is more on linking degradation rates to temporal variations in temperature and soil moisture but that can also be simulated with a lumped model approach. There must be spatial heterogeneity in the catchment like in pesticide application at plot level, but also in spatial variation in soil texture and soil moisture content (Figure 1 nicely shows the spatial variations in crop types; and the descriptive text mentions various soil types). The added benefit of simulating such spatial variations much better in a distributed model than in a lumped model is not discussed. I recommend that the authors show and discuss this key advantage better as that seems the main novelty of this study. Or could a lumped model not have resulted in an equally good model fit as page 5 states that the spatial variation in soil parameters was in fact low in this (specific) catchment?

PiBEACH was initially developed as a companion model for distributed event-based pesticide transport models, such as OpenLISEM OLP (Commelin et al., 2024), to enhance understanding and support management decisions and risk assessments at the headwater catchment scale. Event-based models are valuable tools for predicting intra-hourly rainfall-runoff dynamics and the associated off-site transport of pesticides. However, their accuracy depends on the initial conditions, which are affected by hydrological dynamics and pesticide dissipation processes occurring between rainfall events. PiBEACH was designed to predict catchment properties that evolve throughout the growing season and across the catchment, such as plow layer bulk density, soil moisture, and pesticide concentrations in both dissolved and particulate fractions. PiBEACH considers the spatial distribution of (i) agricultural plots with different crops and associated pesticide applications based on farmers' technical practices (e.g., pesticide type, application date, and dosage, as shown in Fig. 1 and Table S2 in the Supplement), (ii) non-target areas, i.e. areas where pesticide applications are not intended,

such as grass strips, hedges, and roads. (Fig. 1), which affected runoff generation and propagation (Lefrancq et al., 2017), (iii) different soil types and slope profiles (Lefrancq et al., 2017), and (iv) soil moisture and temperature dynamics as a function of hillslope position.

While lumped models can capture pesticide dynamics at the catchment scale, as highlighted by Lutz et al. (2017), we hypothesized that only a distributed model can effectively support management decisions related to the crop mosaics within the catchment to mitigate pesticide fluxes at this scale.

In summary, both lumped and distributed approaches can be used to investigate pesticide dynamics throughout the growing season. However, only a distributed model can account for spatial heterogeneity across the catchment and support further management decisions.

As outlined in the following responses to the Reviewer's comments, we have now clarified in the manuscript the complementarity of these two approaches, which were developed and applied to the same catchment.

My second main critique is that the authors did not make use of the pesticide concentrations and isotope ratios at the catchment outlet (besides the topsoil data) as also reported by Alvarez-Zaldivar et al. (2018). I strongly recommend that the authors add the simulation of the catchment outlet data as reported by Alvarez-Zaldivar et al. (2018) to this work. I see this as a pre-requisite to allow to draw conclusions on pesticide degradation at the catchment scale. The model is now only calibrated to observations on the first cm of soil in the catchment. The (degradation) model is not calibrated to anything deeper than this top one cm. Calibrating the model also to the available catchment outlet data enables to draw conclusions on pesticide degradation at the whole catchment scale, and would make this a really unique study. All conclusions drawn on catchment scale are in the current version based on extrapolation of the topsoil data. Also include a discussion on the pros/cons of calibrating such models with only topsoil data (as done in this current work) vs. outlet data (as done by Lutz et al. 2017), or both. See later "other comments" that elaborate on this main comment #2.

We acknowledge the comment of the Reviewer to clarify this issue in the manuscript. We used river discharge at the outlet in the calibration function and the S-metolachlor fluxes to examine the predictability of PiBEACH (Fig. 5C). As suggested by Reviewer 1, we have now reorganized the article's structure by adding a new section in the Materials and Methods section, which introduces the six metrics simulated after calibration and derived from soil and river samples. Additionally, we have provided a detailed explanation of how the observed values, including discharge and S-metolachlor concentration at the outlet, were used to derive the observed values for three of the six metrics. As expected, given PiBEACH's limited ability to capture intra-daily runoff generation, it is unable to accurately simulate S-metolachlor fluxes at the catchment scale (Fig. 5C), representing less than 2% of the applied amount. Therefore, we opted not to include outlet concentration and isotopic signature, mainly collected during rainfall-runoff events (Alvarez-Zaldivar et al., 2018), in the calibration function to avoid a biased calibration process, which could lead to parameter fitting for the wrong reasons in an attempt to compensate for the model's conceptual limitations.

As emphasized by the reviewer, we prioritized topsoil data during the PiBEACH calibration process, specifically using 103 samples with S-metolachlor concentration and isotopic

signatures. We derived two internal variables, i.e. predicted soil temperature and water content, to extrapolate degradation across time, space, and depth. The degradation dynamics, incorporating soil temperature and water content in PiBEACH, was validated through specific lab-scale experiments on Alteckendorf soil (see Supplement, Section 9). The results enhanced the interpretation of S-metolachlor export, considering the daily limitations of PiBEACH, with predicted and observed export rates of 2 ± 6% and 0.3% of the applied S-metolachlor, respectively.

To summarize, we leveraged a substantial portion of the data collected at the catchment outlet for both calibration (discharge in the KGE objective function) and predictability assessment (discharge and S-metolachlor concentrations to estimate export). Due to the conceptual limitations of PiBEACH, particularly its daily time step, the comparison between simulated and observed S-metolachlor reactive transport was only partially conducted, in contrast to previous lumped modelling approaches (Lutz et al., 2017).

The added value of integrating both observed S-metolachlor concentrations and isotopic signatures, at the outlet and in the topsoil, into the calibration process of distributed modelling will be further explored in the upcoming PiBEACH-OpenLISEM OLP modelling study. This study is now in collaboration with Wageningen University, the developer of the OLP pesticide module (Commelin et al., 2024).

This second major concern of the Reviewer, along with the relevant data from the river outlet, has now been incorporated into the revised manuscript, as detailed below.

**Additional comments:**

1. Title: why "models"? Plural? It seems only one model has been applied?

As suggested by the reviewer, we have now revised the title to use the singular form of model.

2. Introduction:
   1. Line 72: Explain the concept of CSIA better such that the reader can understand how information on degradation can be derived from isotope ratios.

We have revised the following sentence to clarify the connection between isotopic signature and degradation:

*This results in a kinetic isotope effect leaving a chemical imprint in the form of characteristic changes in isotope ratios of reacting molecules (Elsner, 2010). The isotope enrichment factor ($\varepsilon_{lab}$, e.g for carbon) can be derived from closed microcosm degradation experiments and used to quantify pesticide degradation in field studies (Alvarez-Zaldivar et al., 2018). In contrast, non-degradative pesticide dissipation processes, such as sorption generally do not result in significant isotope fractionation (Schmidt et al., 2004; Alvarez-Zaldivar et al., 2018; Droz et al., 2021).*

2. Line 78: some publications have shown that sorption can lead to significant isotope fractionation effects in the spreading direction of pollution which might

be relevant in catchment studies as well. Why not test this with this model, as this seems straightforward?

Non-destructive processes, such as dispersion and sorption, can induce significant isotope fractionation, but only under specific conditions, including strong sorption behavior, early transient diffusion, and multi-step sorption in aquifers. In such environments, the slightly stronger sorption of light isotopologues can enhance enrichment in the heavier isotopologues at the leading edge of a sorptive contaminant pulse (early stages) while reducing enrichment at the trailing edge (later stages) (Eckert et al., 2013; van Breukelen and Prommer, 2008; van Breukelen and Rolle, 2012). This effect is particularly relevant when biodegradation is slow.

In our case, S-metolachlor is not expected to exhibit strong sorption behavior, as strongly sorbing contaminants typically have log $K_{ow}$ > 2 and water solubility < 1 ppm, and S-metolachlor degradation in soil was fast. Molecular diffusion, which may lead to non-equilibrium behavior due to concentration gradients between mobile and immobile phases, plays a negligible role in contaminant redistribution when S-metolachlor is first introduced into the field. Furthermore, unlike highly sorptive contaminants in aquifers, S-metolachlor does not form distinct front and tail effects since it is applied uniformly across the catchment and gradually diffuses toward the outlet. Although low but measurable isotope fractionation may occur following multiple sorption–desorption cycles (Kopinke et al., 2017), sorption-induced stable isotope fractionation is generally considered negligible for moderately sorptive pesticides like S-metolachlor in surface agro-ecosystems during seasonal transport.

For these reasons and because sorption-induced isotope fractionation is expected to be very low and negligible in our case, we did not simulate sorption-induced isotope fractionation in our model.

We have now explicitly stated this in the manuscript (line 89), and four additional references have been included:

*Non-destructive processes such as dispersion and sorption can induce isotope fractionation under conditions of strong sorption behavior, early transient diffusion, and multiple sorption–desorption cycles (Eckert et al., 2013; van Breukelen and Prommer, 2008; van Breukelen and Rolle, 2012; Kopinke et al., 2017). However, as sorption-induced isotope fractionation is considered negligible for S-metolachlor during seasonal transport in surface agro-ecosystems and biodegradation in soil was fast, it was not included in the model.*

3. Lines 77-83: I feel that neglecting isotope fractionation effects of non-transformation processes can be more convincingly elaborated.

We agree with the reviewer.

We have now specified in the manuscript (line 89):

*"Given the physico-chemical properties of S-metolachlor, abiotic degradation processes (e.g., photolysis) and their associated isotope fractionation effects were excluded from the modelling. Non-destructive processes such as dispersion and sorption can induce isotope*

*fractionation under conditions of strong sorption behavior, early transient diffusion, and multiple sorption–desorption cycles (Eckert et al., 2013; van Breukelen and Prommer, 2008; van Breukelen and Rolle, 2012; Kopinke et al., 2017). However, as sorption-induced isotope fractionation is considered negligible for S-metolachlor during seasonal transport in surface agro-ecosystems and biodegradation in soil was fast, it was not included in the model."*

3.  Methods:
    1.  Line 177: explain what a mixing topsoil layer is. Ploughed?

As also mentioned by the Reviewer 1, the mixing layer concept is probably the most commonly used in pesticide fate models due to its numerical simplicity and its realistic representation, as hortonian runoff processes have repeatedly been shown to be initiated as a near-surface process (Gao et al., 2004; McGrath et al., 2008; Wallender et al., 2008). This concept assumes that the transport is controlled by a mixing layer below the soil surface, in which rainwater, soil solution and runoff water completely and instantaneously mix (Havis et al., 1992). The soil depth interacting with runoff water is variable ranging from 0.25 to 2 cm according to various experimental and modelling studies (Havis et al., 1992; Wallender et al., 2008). Previous studies have shown that the effective depth of interaction is related to the degree of soil aggregation and increases with soil slope, kinetic energy of raindrops and rainfall intensity (Havis et al., 1992). In fact, the observed mixing-layer depth is often much shallower than the depth required for fitting models to field data (Gao et al., 2004).

The ploughed layer extends from 1 to 30 cm below the surface (second layer in PiBEACH). We have revised the sentence to provide more detailed information about the mixing layer directly in the article.

Gao, B., Walter, M.T., Steenhuis, T.S., Hogarth, W.L., Parlange, J.Y., 2004. Rainfall induced chemical transport from soil to runoff: theory and experiments. Journal of Hydrology. 295, 291-304.

Havis, R.N., Smith, R.E., Adrian D.D., 1992. Partitioning solute transport between infiltration and overland flow under rainfall. Water Resour. Res., 28, 2569-2580.

Wallender, W.W., Joyce, B.A., Ginn, T.R., 2008. Modeling the Transport of Spray-Applied Pesticides from Fields with Vegetative Cover. Transactions of the Asabe. 51, 1963-1976.

Additionally, the reference (McGrath et al., 2008) has been cited at the first mention of the mixing layer to offer a more comprehensive understanding of this concept.

Line 215:

*Compared with BEACH features, PiBEACH includes (i) a mixing topsoil layer (assuming that the pesticide transport is controlled by a mixing layer from 0 to 1 cm below the soil surface, in which rainwater, soil solution and runoff water completely and instantaneously mix, McGrath et al., 2008) and additional deepest soil layers to include the shallow groundwater contribution to discharge,*

2.  Some textual errors on lines 168 and 177.

We revised the targeted sentences:

*Daily vertical water fluxes across soil layers, followed by lateral fluxes, were considered at the cell scale from upstream to downstream. The surface flow direction was extracted from the digital elevation model, and flow-accumulation was determined using grid-based functions, without relying on a numerical scheme.*

*Compared with BEACH, PiBEACH includes (i) a mixing topsoil layer (assuming that the pesticide transport is controlled by a mixing layer from 0.25 to 2 cm below the soil surface, in which rainwater, soil solution and runoff water completely and instantaneously mix, McGrath et al., 2008) and additional deepest soil layers to account for the contribution of the shallow groundwater contribution to discharge, …*

3. Line 181: I think that not the various isotopologues are simulated in the model but the bulk heavy and light carbon in the pesticide?

As suggested by the reviewer, we have revised the sentence.

*[…] and (v) pesticide carbon stable isotopic fractionation and transport of bulk heavy and light carbon in the pesticide (e.g., $^{13}C$ and $^{12}C$) (Lutz et al., 2017; Alvarez-Zaldivar et al., 2018).*

4. Line 187: "numerical diffusion"? Do they authors mean numerical dispersion?

We agree with the reviewer that 'numerical dispersion' refers to a type of truncation error that occurs when solving the diffusion-advection equation using finite differences, and have revised the relevant sentence accordingly.

5. Line 188: explain what the export coefficient is.

We provided the definition of the export coefficient:

*This is because the numerical dispersion  (Gatel et al., 2020) is generally of the same order of magnitude as pesticide export coefficient (i.e., the ratio of the mass transported at the outlet to the applied mass at the catchment scale), i.e., from 1‰ to 1% of the applied amount of pesticides.*

6. Line 192-193: does this then not prohibit the simulation of hot-moments?

We assumed that the aspect targeted by this comment is the daily time step, which constrains its ability to predict fast flow dynamics, such as runoff generation and preferential water flow. We fully acknowledge that a daily model, using daily rainfall data, cannot capture intra-hourly rainfall intensity, which drives runoff intensity and surface off-site pesticide transport. However, this limitation, which was one of the key motivations for developing PiBEACH in conjunction with an event-based model like OpenLISEM OLP, does not, in our opinion, preclude the simulation of hot moments for degradation. Considering "soil moisture and temperature"- dependent degradation, we can account for periods of slower or faster degradation in the topsoil across the catchment, primarily driven by temperature. For example, during the first application in April, the $DT_{50}$ was close to 100 days, with an average temperature of 9.1 ± 2.9°C (Fig. 3C and Table S2). As topsoil temperatures progressively increased (Fig. 3A), the degradation rate accelerated, reaching a $DT_{50}$ of approximately 25 days

in May, four times faster than in April. The impact of topsoil soil moisture on the $DT_{50}$ was secondary (Fig. 3A and C). Between March and July, the topsoil temperature increased, displaying a more pronounced temperature gradient with depth, which resulted in reduced reactivity at greater depths. For example, in July, the $DT_{50}$ values were 25 days in the topsoil (Fig. 3C) and 34 days at a 2-meter depth (data not shown).

In summary, the "soil moisture and temperature" dependence, as a proxy for microbial activity, provided a standardized approach for depicting degradation kinetics across the catchment and throughout the year.

Model approach general: I do not understand why this model has not been applied to simulate data at the catchment outlet as it seems to be designed to this end. Could not a simpler model been applied to simulate only the data in the topsoil?

We thank the reviewer for this comment. We agree that a lumped and parsimonious modelling approach, such as that developed by Lutz et al. (2017), effectively captures the dynamics of pesticide degradation and transport at the catchment scale. However, this approach was not designed to provide a distributed representation of pesticide concentrations in the topsoil throughout the growing season to assess off-site pesticide transport resulting from runoff and erosion. Such information is critical for informing management decisions and conducting risk assessments. PiBEACH was initially developed for this purpose as a companion model for distributed, event-based pesticide transport models, such as OpenLISEM OLP (Commelin et al., 2024), and cannot be used independently to simulate data at the catchment outlet. We agree that a lumped and parcimonious modelling approach such a as developed by Lutz et al., (2017) was able to capture the dynamic of pesticide degradation and transport at the catchment scale. Therefore, this lumped approach was not designed to provide distributed extent of pesticide amount in topsoil when a rainfall event causes runoff and erosion with associated pesticide transport to support management decisions and risk assessments. PiBEACH was initially developed for this purpose as a companion model for distributed event-based pesticide transport models, such as OpenLISEM OLP (Commelin et al., 2024) and cannot be directly applied to simulate data at the catchment outlet. PiBEACH was not originally designed to simulate discharge and the associated pesticide concentrations in headwater catchments, where surface runoff and off-site transport can play a significant role in pesticide fluxes in rivers, such as in the Alteckendorf catchment with loamy soils (Lefrancq et al., 2018). Additionally, PiBEACH does not account for erosion and the transport of pesticides sorbed in the suspended matter, which is another limitation for simulating pesticide concentrations in rivers. The particulate fraction of some pesticides can be substantial at the outlet following intense rainfall events (Lefrancq et al., 2018).

As discussed previously, PiBEACH has a dual objective:

- To simulate both degradation and non-degradative processes across the catchment and throughout the crop growing season, providing key inputs for distributed, event-based models such as OpenLISEM OLP, which simulates the transport of both dissolved and particulate pesticides, based on the remaining pesticides in the mixing layer. This contributes significantly to pesticide modelling at the catchment scale.
- To model based-flow and the associated pesticide concentrations, enabling the integration of discharge and pesticide concentrations between rainfall events.

We thank the reviewer for highlighting the importance of considering the complexity or simplicity of a model in providing targeted answers. We have not verified in the section on model description in the Materials and Methods and this issue has been addressed: "the development of the PiBEACH conceptual model relied on field knowledge of hydrological dynamics and associated S-metolachlor flows in the Alteckendorf catchment, translating our experimentalist's understanding (perceptual model) into conceptual model", with an appropriate level of complexity to meet the two objectives.

To clarify the application domain of PiBEACH and prevent any misinterpretation of its capabilities and uses, we have now provided additional details in the title:

*Constraining topsoil pesticide degradation in a conceptual distributed catchment model with compound-specific isotope analysis (CSIA)*

In the Abstract (Line 28):

*Based on pesticide concentrations in topsoil distributed across the catchment and throughout the crop growing season, PiBEACH paves the way for coupling with a distributed event-based models, such as OpenLISEM OLP, which are better suited to address the intra-event dynamics of pesticide transport towards the catchment outlet.*

As detailed below, this information is also provided in the Materials and Methods section (see the new section on metric calculation, "2.7 Metrics of Pesticide Persistence and Transport Risks," Line 449, page 13 of this document) and in the Results and Discussion section (Line 589, with further details on the last page of this document).

Lines 205-206: what is meant with the depth of the groundwater layers varying constantly from upstream to downstream?

As described, the Alteckendorf catchment was not equipped with piezometers or wells to monitor the dynamics of the water table. Therefore, we adopted a conceptual approach based on our understanding of soil structure (Lefrancq et al., 2017, 2018), capable of simulating the observed seasonal fluctuations in the water table when monitoring data is available (Molenat et al., 2005). A two-layer shallow aquifer (z3 and z4) was defined, with its thickness decreasing from the upper hillslopes (maximum of 23.2 m) to the riverbanks. The shallow aquifer is encased by unsaturated layers (Z0 to Z2, with a total thickness of 80 cm), which include a mixing layer, the plough layer, and a layer affected by artificial drainage, as determined by our soil structure investigation (Lefrancq et al., 2017, 2018). The shallow aquifer was divided into an upper variably saturated layer (z3), influenced by aquifer recharge, and a lower saturated layer that drains towards the river, functioning as a global linear reservoir (Manfreda et al., 2005). A calibration parameter (Zf) was used to distinguish the upper and lower portions of the shallow aquifer, accounting for the gradual decrease in thickness of both the saturated and variably saturated layers. Consequently, we revised the targeted sentence by replacing 'constantly' with 'gradually' to more accurately reflect our approach.

*The depth of groundwater layers (z3 + z4) varied gradually from upstream to downstream, exhibiting seasonal fluctuations driven by aquifer recharge, as observed in the hillslopes of monitored catchments (Molenat et al., 2005).*

With a new reference:

Molénat, J., Gascuel-Odoux, C., Davy, P., Durand P.: How to model shallow water-table depth variations: the case of the Kervidy-Naizin catchment, France, Hydrol. Process., 19, 901-920, https://doi.org/10.1002/hyp.5546, 2005.

7. Page 9: as sorption was simulated in the model, why not test / show that sorption isotope fractionation has limited/negligible effects on the simulated isotope ratios in topsoil (and especially at the outlet?)?

As discussed previously in the point 2 raised by the Reviewer, S-metolachlor is not expected to exhibit strong sorption behavior, as strongly sorbing contaminants typically have log $K_{ow}$ > 2 and water solubility < 1 ppm, and S-metolachlor degradation in soil was fast. Molecular diffusion plays a negligible role in contaminant redistribution when S-metolachlor is first introduced into the field, potentially leading to non-equilibrium behavior due to concentration gradients between mobile and immobile phases. Furthermore, unlike highly sorptive contaminants in aquifers, S-metolachlor does not form distinct front and tail effects since it is applied uniformly to corn and sugar beet plots across the catchment and gradually diffuses toward the outlet. Although low but measurable isotope fractionation may occur following multiple sorption–desorption cycles (Kopinke et al., 2017), sorption-induced stable isotope fractionation is generally considered negligible for moderately sorptive pesticides like S-metolachlor in surface agro-ecosystems during seasonal transport. Thus, we did not simulate sorption-induced isotope fractionation in our model.

8. Page 10: add bit of explanation to argue why it was chosen for to simulate only the bioavailable fraction? How would it influence model outcomes compared to the more common model formulation that simulates biodegradation of the total dissolved fraction? Is this in line with earlier catchment degradation model studies?

As noted by the reviewer regarding common model formulations, we also accounted for the biodegradation of the total dissolved fraction. Additionally, we assumed that the reversibly sorbed fraction of pesticides, subject to daily sorption/desorption equilibrium, undergoes biodegradation at a similar rate, expressed by the $DT_{50,reference}$ value. However, as in several pesticide transport models, e.g. PEARL, PELMO, PRZM or MACRO, we included an aged fraction that is more strongly sorbed and undergoes limited degradation.

In summary, consistent with the following models, we considered three pesticide pools: the bioavailable fraction, comprising both the (1) total dissolved fraction and (2) the reversibly sorbed fraction, and (3) a non-extractable fraction.

9. Line 265: "depth-dependent degradation". The argument not to include this in the model seems rather weak (because there are no data available from deeper

soil layers; but there are in fact catchment outlet data available). If that reasoning is followed the authors should also only draw conclusions on what happens in the top soil layer and not in the catchment as a whole. Lutz et al. 2017 included "depth-dependent degradation" to enable fitting the concentration and isotope ratios of pesticides at the catchment outlet. But the current model is not constrained with such outlet data and therefore there will be high uncertainty on what happens at the catchment scale. In fact as the model is only calibrated with topsoil data (and discharge at the outlet but that is not that relevant for catchment scale degradation extent), the model is therefore only informative on what happens in the upper 1 cm of the catchment. See also main comment #2.

We fully agree with the Reviewer that the apparent depth-dependent degradation may be a parsimonious approach to represent complexity of depth-dependent microorganism activity as shown in different recent papers. We have chosen in this paper to explicitly link the persistence of pesticide with soil moisture and temperature as key drivers of microorganism activity integrating both temporal, spatial and depth dynamics such as in MACRO or PEARL models. We tested at the lab scale on the Alteckendorf soil, the predictability of this approach (Figure S4) before to implement it in PiBEACH. With this "soil moisture and temperature"-dependent degradation, we can account for periods of slower or faster degradation in the topsoil across the catchment. Figure 3C highlights cold and hot moments, i.e., slower and then faster degradation, primarily driven by temperature. The $DT_{50}$ was close to 100 days during the first application in April at 9.1 ± 2.9°C, before decreasing to 25 days in May. As topsoil temperatures progressively increased (Fig. 3A), the degradation rate stabilized in May, with a $DT_{50}$ of approximately 25 days—four times faster than in April. The impact of topsoil soil moisture on the predicted $DT_{50}$ was secondary (Fig. 3A and C). Between March and July, the topsoil temperature increased, displaying a more pronounced temperature gradient with depth, which resulted in reduced reactivity at greater depths. For example, in July, the $DT_{50}$ values were 25 days in the topsoil (Fig. 3C) and 34 days at a 2-meter depth (data not shown). The adopted "soil moisture and temperature" dependence as a proxy for microorganism activity allow an uniformized method to depict degradation kinetic across the catchment and throughout the year.

This decrease resulted in a moderate, depth-dependent degradation of PiBEACH, in line with predictions from the calibration of the lumped model developed by Lutz et al. (2017) for this catchment. In both cases, i.e., the depth-dependent degradation (Lutz et al., 2017) and soil moisture and temperature-dependent degradation, the adopted approach can only tackle tendencies of microbial activity which remains largely unknown and complex to monitor along soil depth and at the catchment scale.

The next step, opened with the PiBEACH development, will be the combination with an event-based model such as OpenLISEM OLP which is expected to significantly improve the intra-daily runoff and associated dissolved and particulate transport from the plots to the outlet. This combined approach opens the way to open the grey box of soil reactivity by introducing new dependence and calibrate associated parameters.

We have now clarified the assumptions of soil moisture and temperature dependence in the Materials and Methods section and completed the discussion (line 316), as suggested by the Reviewer.

Line 316:

*Degradation rates generally decrease over soil depth, e.g., due to lower microbial activity (Cruz et al., 2008; Lutz et al., 2017) or sorption (Arias-Estevez et al., 2008) in deeper soil layer. The lack of concentration and isotope data for S-metolachlor in deeper soil layers prevented direct consideration of depth-dependent degradation.*

Line 353:

Between March and July, the topsoil temperature increased, displaying a more pronounced temperature gradient with depth, which resulted in reduced reactivity at greater depths. For example, in July, the $DT_{50}$ values were 25 days in the topsoil (Fig. 3C) and 34 days at a 2-meter depth (data not shown), as predicted by the lumped model developed by Lutz et al. (2017) for this catchment. In both cases, the depth-dependent degradation (Lutz et al., 2017) and the soil moisture and temperature-dependent degradation, the adopted framework can only capture general trends in microbial activity, which remains largely unknown and challenging to monitor across soil depth and at the catchment scale.

10. The three applications as shown in Fig. 3 are not described in the methods section (or I could not find it): how much pesticide was applied when and where? Exactly this heterogeneity in source zone variation calls for a model like this one but the advantage of using a distributed model has not been illustrated.

The application details were provided in the initial version, in the Supplement (Table S2).

Table S2. Applied mass (Kg) of active ingredient (S-metolachlor) per transect by date and days since 1st application. Ranges indicates uncertainty of exact application date (Alvarez-Zaldivar et al., 2018).

| App. No. | Date | Days | North | Valley | South |
|---|---|---|---|---|---|
| A1 | March 20 - 25th | 0 - 5 | 5.1 | 1.6 | 11.1 |
| A2 | April 13 - 14th | 25 - 26 | 8.0 | 1.8 | 2.9 |
| A3 | May 25 - 31st | 67 – 73 | 7.2 | 2.4 | 0.0 |
| Total | | | 20.2 | 5.9 | 14.0 |

We have now revised the sentence in the Materials and Methods to summarize Table S2, emphasizing the spatial and temporal variability of S-metolachlor applications at the plot scale. In our view, this variability highlights the necessity of a distributed modelling approach such as PiBEACH.

Line 161:
*The three application dates with, associated doses and formulation of S-metolachlor were collected for each plot through farmer surveys, providing spatially distributed data for the modelling. In sumary, applications took place from late March to late May, with total amounts*

*of 20.2 kg, 5.9 kg, and 14 kg applied to plots in the North, Valley, and South transects, respectively (Table S2).*

Results & Discussion:

11. Generally: explain why concentrations and isotope ratios of pesticide at the catchment outlet were not used. I assumed this was done when starting reading the paper but only in the R&D section is became clear that only data from the upper 1cm of the catchment skin were measured and simulated. How then can conclusions be drawn on overall degradation at the catchment body when these simulations are not constrained with outlet data? Thus relatedly, what is the relevance of the simulations of the upper 1 cm in the soil as presented in the key result Figure 3 on what happens at catchment scale? See also main comment #2.

We acknowledge Reviewer 2's comments regarding the relevance of the catchment results (Figure 5). As suggested by Reviewer 1, we have now reorganized the article's structure by adding a new section to the Materials and Methods, which introduces the six metrics simulated after calibration and derived from soil and river samples. Additionally, we have provided a detailed explanation of how the observed values, i.e., topsoil S-metolachlor concentration and $\delta^{13}C$, but also river discharge, and S-metolachlor concentration at the outlet, were used to derive the observed values for three of the six metrics.

The model was calibrated using topsoil concentration (KGESM > 0.5) and isotopic signature (KGEδ > 0.8), along with the flow rate at the outlet (KGEQ > 0.5), to represent hydrological functioning at the catchment scale. As mentioned in the Results section (lines 438–440), "*The tendency of PiBEACH to overestimate runoff generation is related to the basic assumptions of the SCS-CN approach (see section 6.2, SI), and to the daily time-step concealing daily distribution of rainfall intensity*". This result (Figure 3B) confirms the limitations of PiBEACH in accurately representing intra-daily and intra-hourly runoff generation and the associated fast-flow pesticide transport.

The S-metolachlor concentrations measured at the outlet throughout the growing season were also used to derive the observed export metric (Figure 5C). As expected, given PiBEACH's limited ability to capture intra-daily runoff generation, it is unable to accurately simulate S-metolachlor fluxes at the catchment scale representing less than 2% of the applied amount. Therefore, we opted not to include outlet concentration and isotopic signature in the calibration function to avoid a biased calibration process, which could lead to parameter fitting for the wrong reasons in an attempt to compensate for the model's conceptual limitations.

The added value of incorporating observed S-metolachlor concentrations and isotopic signatures at the outlet in the calibration process will be explored in the upcoming PiBEACH-OpenLISEM OLP modelling study.

It is important to emphasize that the primary goal of PiBEACH, which has been now clarified in the manuscript, is to simulate the degradation and persistence of pesticides in topsoil layer

for coupling with event-based models such as OpenLISEM OLP. Figures 5A (isotopic dynamics) and 5B (remaining mass in topsoil) demonstrate that PiBEACH performs well in achieving this objective.

> 12. The three levels of sampling resolution as presented in Figure 4: composite, transects, plots are unclear, and not clearly described and explained in the method section. This would help also on improving on main comment #1.

Following the suggestion of both reviewers, we have now revised the caption of the Figure 1, and the following sentence, to explain the weekly weighted samples of topsoil for each transect from regular sampling (green dots) along transects (see Alvarez-Zaldivar et al., 2018).

*Figure 1: The headwater catchment of Alteckendorf (Bas-Rhin, France), illustrating the experimental setup, which includes transects (weighted samples collected at the green dots along the red transects) and plot sampling (black dots), as well as the land use for 2016. The "Other" category encompasses roads, grass strips, and orchards.*

*Lines 146-148*

*Soil samples were collected from 13 marked plots (before S-metolachlor applications and on days 1, 50, and 100 post-application), as well as from the north, valley, and south transects across the catchment (weekly weighted samples from the green dots in Fig. 1, spanning from March 19th, i.e., day 0, to July 12th, i.e., day 115). These samples were analyzed for the quantification and CSIA ($\delta^{13}C$) of S-metolachlor in soil*

> 13. Lines 440-441: in my words: "more detailed spatial soil sampling did not help to better constrain the model". Why then was a spatially distributed model needed?

Mobilizing data of isotopic signatures through higher-resolution soil sampling (plots > transects > transect composite samples) did not significantly improve model calibration in this catchment. Therefore, pooling weekly transect samples into a single composite soil sample for CSIA appears sufficient to constrain the degradation parameter, i.e. $DT_{50\ reference}$, while also reducing sampling and analytical efforts. This composite approach may effectively capture temporal variations in pesticide degradation at the catchment scale.

We hypothesized that the transect composite approach, with a single averaged topsoil $\delta^{13}C$ value per week, could also enhance the calibration of lumped models such as the one developed by Lutz et al. (2017). However, this result does not disqualify distributed modelling approaches such as PiBEACH, which was primarily designed to spatially represent pesticide concentrations, hydraulic properties, and topsoil moisture as initial conditions for distributed event-based models.

> 14. Lines 443-447: the difference in fractionation factor -2.7 vs. -1.5 is quite large and needs further explanation. Also compare and discuss the outcome with the fractionation factor calibrated by Lutz et al (2017): - 1.3 permil.

The incorporation of CSIA data into PiBEACH yielded an isotope fractionation factor of $\varepsilon C = -2.7 \pm 0.6$ ‰ for S-metolachlor degradation in topsoil. This value is notably larger than those reported in previous studies, including Alvarez-Zaldivar et al. (2018) ($\varepsilon C = -1.5 \pm 0.5$ ‰) and Lutz et al. (2017) ($\varepsilon C = -1.3 \pm 0.5$ ‰). While this discrepancy suggested differences in degradation dynamics or environmental conditions, it remained within the uncertainty range obtained from the model ensemble in this study.

The observed stronger fractionation effect ($\varepsilon C = -2.7 \pm 0.6$ ‰) may result from variability in environmental conditions, microbial activity, and transformation kinetics, supporting a more refined assessment of S-metolachlor degradation. The fractionation factor reported by Alvarez-Zaldivar et al. (2018) ($\varepsilon C = -1.5$ ‰) was obtained under laboratory conditions, where experimental constraints may have reduced isotope fractionation compared to field-scale observations. Similarly, the $\varepsilon C = -1.3 \pm 0.5$ ‰ reported by Lutz et al. (2017) was derived from a different dataset and modelling approach.

The present study incorporates a more extensive dataset and employs model ensembles, capturing a broader range of degradation scenarios and yielding a stronger fractionation effect. This suggests that enzymatic degradation may have been more pronounced, potentially due to higher microbial activity or the presence of a dominant microbial consortia with different metabolic capabilities.

While the previously reported $\varepsilon C$ values ($-1.5$ ‰ and $-1.3 \pm 0.5$ ‰) may have slightly overestimated degradation extent, the discrepancy remains within the uncertainty bounds determined in this study. These findings emphasize the need for site-specific calibration when applying CSIA-based degradation assessments and highlight the importance of model ensemble approaches in accounting for uncertainties in isotope fractionation factors under field conditions.

The difference in isotope fractionation in previous and this study is now discussed in the manuscript (line 536):

*"The integration of CSIA data into PiBEACH yielded an isotope fractionation factor of $\varepsilon C = -2.7 \pm 0.6$ ‰ for S-metolachlor degradation in topsoil, exceeding previously reported values (Alvarez-Zaldivar et al., 2018: $-1.5 \pm 0.5$ ‰; Lutz et al., 2017: $-1.3 \pm 0.5$ ‰). This discrepancy likely reflects differences in environmental conditions, microbial activity, and transformation pathways, yet remains within the uncertainty range derived from the model ensemble. Compared to laboratory-derived estimates, the stronger fractionation observed in this study suggests alternative degradation pathways or the influence of distinct microbial consortia at the catchment scale. The broader dataset and model ensemble approach provide a more comprehensive assessment of degradation processes, capturing a wider range of field-scale conditions. While previously reported $\varepsilon C$ values may have slightly overestimated degradation extent, these findings emphasize the need for site-specific calibration in CSIA-based assessments. Furthermore, they highlight the importance of model ensembles in addressing uncertainties associated with isotope fractionation factors in heterogeneous environmental settings, such as agro-ecosystems."*

      15. Line 458+459: "measure", change into "measurements"

We have corrected the sentence as suggested.

> 16. Figure 5: what is the relevance if these results at catchment scale when the model is only calibrated on topsoil data but not on outlet pesticide data?

We acknowledge the Reviewer's comments regarding the relevance of the catchment results (Figure 5). As mentioned in the previous point n°12, this has now been clarified by specifying how the soil and river samples were used in a new section to the Materials and Methods. As discussed previously, it should be mentioned that the primary goal of PiBEACH, is to simulate the degradation and persistence of pesticides in topsoil layer for coupling with event-based models such as OpenLISEM OLP. This has now been clarified in the manuscript. Figures 5A (isotopic dynamics) and 5B (remaining mass in topsoil) demonstrate that PiBEACH performs well in achieving this objective.

> 4. Discussion is largely lacking. Add a larger discussion section to include for example:

We acknowledge the Reviewer's feedback and we have clarified the discussion by incorporating elements related to the three highlighted points and previously discussed.

> 1. Comparison with findings Lutz et al. (2017) who applied a similar model to the same catchment but then calibrated to earlier data at the catchment outlet instead of the topsoil. Add some discussion to discuss why this study did find that having isotope data let to better model results, whereas this effect was not clearly found by Lutz et al. (2017). Note that Lutz et al (2027) had to include depth-dependent degradation to enable to simulate the outlet pesticide and isotope ratio data.

We have now added the following section, discussing the main contributions of each modelling study (Lutz et al., 2017 and this one), in the results section (line 600), before the conclusion:

*In summary, $\delta^{13}C$ data from both soil samples, used to constrain degradation, and the river samples, used to capture additional degradation processes along pesticide transport, can strengthen the robustness of the calibrated parameters. Both lumped (Lutz et al., 2017) and distributed approaches, such as PiBEACH, can be used to investigate pesticide dynamics throughout the growing season. However, only a distributed model can account for spatial heterogeneity across the catchment to support further management decisions. Additionally, PiBEACH provides spatially explicit data on pesticide concentrations, hydraulic properties, and topsoil moisture, which are crucial as initial conditions for distributed event-based models. We leveraged a substantial portion of the data collected at the catchment outlet for both calibration (discharge in the KGE objective function) and predictability assessment (discharge and S-metolachlor concentrations to estimate export). Due to the conceptual limitations of PiBEACH, particularly its daily time step, the comparison between simulated and observed S-metolachlor reactive transport was only partially conducted, in contrast to previous lumped modelling approaches (Lutz et al., 2017). The added value of incorporating observed S-*

*metolachlor concentrations and isotopic signatures at the outlet in the calibration process will be explored in the upcoming PiBEACH-OpenLISEM OLP modelling study.*

2. Pros/cons topsoil vs. catchment outlet data.

We have now added the following section, discussing the strengths and weaknesses of using soil and river $\delta^{13}$C data to constrain reactive transport models, in the results section (line 594), before the conclusion:

*This new modelling approach, applied to a similar compound and catchment as in Lutz et al. (2017), allows for a discussion of the strengths and weaknesses of using soil and river $\delta^{13}$C data to constrain reactive transport models. The lumped modelling approach (Lutz et al., 2017) focuses on $\delta^{13}$C of S-metolachlor in the river at the catchment outlet, representing an integrated signature of ongoing degradation at the catchment scale. This model was developed as a standalone approach to evaluate the utility of additional S-metolachlor $\delta^{13}$C data in river systems during calibration. In the PiBEACH framework, we further investigated the added value of the topsoil $\delta^{13}$C data to reduce uncertainty in the degradation kinetics. For three components of the S-metolachlor mass balance, namely, degraded mass, remaining mass in the topsoil, and export at the catchment outlet, their predicted cumulative contributions against observed data could be validated. We did not explore the combined use of both topsoil and river S-metolachlor $\delta^{13}$C in PiBEACH due to conceptual limitations in accurately accounting for rapid surface off-site transport associated with runoff and erosion. However, with the planned coupling of PiBEACH and an event-based model, such as OpenLISEM OLP, integrating both topsoil and river $\delta^{13}$C data may substantially improve the constraints on ongoing degradation at the catchment scale and enhance the representation of surface fast-flow transport during intense rainfall-runoff events.*

3. Line 493: I thought the conclusion was that there are no "hot-spots" in this catchment, as there is limited spatial variation among the topsoil samples? Therefore, add some discussion what the added benefit of a spatial distributed model was in this case.
5. Hot-spot and hot-moments. The paper does not discuss hot-spots in the catchment. The advantage of this model is that it enables to account for spatial heterogeneity but it seems that all model parameters were taken spatially homogeneously except with depth. Therefore, the added benefit to the model applied by Lutz et al. 2017 remains unexplored.

We have now added the following section discussing the points 3 and 5 as suggested by the reviewer 2

*The degradation process in PiBEACH, dependent on soil temperature and moisture, was primarily governed by temperature, exhibiting a distinct seasonal and vertical pattern. Consequently, the predicted degradation hotspot was located in the topsoil layer. Soil*

*moisture, which showed greater spatial variability than temperature, played a secondary role in degradation, with no significant spatial variability observed between plots treated with S-metolachlor throughout the growing season. The limited predicted spatial variation of DT$_{50}$ across the catchment does not diminish the relevance of PiBEACH's distributed nature, particularly in the context of its coupling with distributed event-based models such as OpenLISEM OLP.*

6. Line 510-511: see earlier comments: as pesticide concentrations and isotope ratios were only measured on the topsoil one cannot simply extrapolate to the catchment scale. Degradation rates are likely much higher in the topsoil than in the deeper soil layers leading to overestimation of a catchment to degrade pesticide. The data to constrain the model also at catchment scale are available. Why not use them?

We have now addressed the second major critique by clarifying in the manuscript (new section to the Materials and Methods, introducing the six metrics and in the new element of discussion) how discharge and S-metolachlor concentration in the river at the catchment outlet were used to calibrate the parameters and assess the simulated S-metolachlor export from the catchment. As previously discussed, we opted not to include outlet concentration and isotopic signature, mainly collected during rainfall-runoff events (Alvarez-Zaldivar et al., 2018), in the calibration function to avoid a biased calibration process, which could lead to parameter fitting for the wrong reasons in an attempt to compensate for the model's conceptual limitations.

*We leveraged a substantial portion of the data collected at the catchment outlet for both calibration (discharge in the KGE objective function) and predictability assessment (discharge and S-metolachlor concentrations to estimate export). Due to the conceptual limitations of PiBEACH, particularly its daily time step, the comparison between simulated and observed S-metolachlor reactive transport was only partially conducted, in contrast to previous lumped modelling approaches (Lutz et al., 2017). The added value of incorporating observed S-metolachlor concentrations and isotopic signatures at the outlet in the calibration process will be explored in the upcoming PiBEACH-OpenLISEM OLP modelling study.*

*Sylvain Payraudeau, on behalf of the co-authors*

*Strasbourg, February 17$^{th}$ 2025*

---

## Author Response (AR1)

Dear reviewers,

We warmly thank the two reviewers for their relevant and useful feedback and comments on our paper titled "Constraining pesticide degradation in conceptual distributed catchment models with compound-specific isotope analysis (CSIA)." We have now carefully addressed and incorporated these comments to emphasized the originality and potential of this study.

Please find below the detailed responses and modifications made (in blue) based on the comments, which have been incorporated into the revised version. The line numbers correspond to the revised manuscript, which already includes the following modifications.

Reviewer 1:

**General comments**
The study showed that incorporating CSIA reduced uncertainty in PiBEACH model for estimating S-metolachlor concentration in soil. The study is interesting, with a good conclusion and abstract. However, the structure of the article requires MAJOR improvements. Some are mentioned below, but the authors' additional effort is also welcome.

We thank Reviewer 1 for their valuable comments. We have now reorganized the structure of the article based on these suggestions and added a new section to the Materials and Methods, introducing the six metrics.

**Specific comments**
**Ln 11**: What do you mean by "the calibration of pesticide dissipation"? Which parameters are related to pesticide dissipation?

We have now clarified the sentence pointing the parameters involved in the pesticide dissipation (Lines 9-13).

"*Predicting pesticide dissipation at the catchment scale using hydrological models is challenging due to limited field data distinguishing degradative from non-degradative processes.* This limitation hampers the *calibration of key parameters such as biodegradation and volatilization half-lives (DT$_{50}$), and the carbon-water partition coefficient (K$_{oc}$), often leading to* equifinality and reducing confidence in predictions of pesticide persistence in topsoil and transport from agricultural field to catchment outlets.*"

**Ln 16** This is confusing; previously, you mentioned "calibration of pesticide dissipation." Typically, you calibrate the parameters of the model. Now you are mentioning "predict daily pesticide dissipation." In this case, you are saying that pesticide dissipation is like an output of the model PIBEACH. Can you explain this? It is very important to understand your research.

Thank you for highlighting this confusion. We have now clarified this issue. Pesticide dissipation in soil results from both degradation processes (biodegradation, photolysis, hydrolysis) and transport between compartments (soil to air via volatilization, soil to surface water via runoff, or soil to groundwater via leaching). Therefore, pesticide dissipation is not a direct output of the model, such as the pesticide concentration at the catchment outlet, but

can be inferred from the model's internal compartments and fluxes. However, a mass balance of pesticide fate can be calculated at each time step, allowing for the prediction of both overall pesticide dissipation and the specific fraction that is definitively degraded. The novelty of the PiBEACH model lies in its ability to refine this prediction by distinguishing between degradative and non-degradative processes, thereby reducing uncertainty.

We have now clarified the sentence to point this novelty (lines 13-18):

*"This study examines the use of pesticide Compound-Specific Isotope Analysis (CSIA) data to improve model predictions of pesticide persistence in topsoil and off-site transport at the catchment scale. The study was conducted in a 47-ha crop catchment using the pre-emergence herbicide S-metolachlor. A new conceptual distributed hydrological model, PIBEACH, was developed to simulate daily pesticide dissipation in soils and its transport to surface waters. The model integrates changes in the carbon isotopic signatures ($\delta^{13}C$) of S-metolachlor during degradation to constrain key parameters and reduce equifinality."*

**Ln 23**, In the phrase "the incorporation of CSIA data into conceptual distributed hydrological models," you should mention that only for PIBEACH, you did not test more conceptual hydrological models (perhaps the results are different). For this article, it could be beneficial to incorporate one more model.

We have revised the following sentence to express that we focused on the PiBEACH model. We agreed to explore the integration of isotopic fractionation in other models in future work and this study highlights the integration of CSIA data in PIBEACH. Therefore, the primary goal of this paper was to lay the groundwork for applying a similar approach to other pesticide models, highlighting the benefits, limitations, and additional field data required, rather than focusing on comparing the added value of CSIA across different models, which is beyond the scope of the current study. This has now been specified (lines 19-25):

*"Incorporating $\delta^{13}C$ data and S-metolachlor concentrations from topsoil samples reduced uncertainty in the degradation half-life by more than half and improved six key metrics related to pesticide persistence and transport. The results highlighted that moderate, targeted sampling effort can identify degradation hot-spots and hot-moments in agricultural soil when stable isotope fractionation is integrated into the model. Overall, integrating CSIA data into PiBEACH model significantly enhances the reliability of pesticide degradation predictions at the catchment scale. In addition, PiBEACH, which accounts for spatial and seasonal variations in topsoil pesticide concentrations, enables coupling with distributed, event-based hydrological models such as OpenLISEM OLP to capture intra-event pesticide transport dynamics more accurately."*

**Ln 41.** In the phrase *"In addition, existing models frequently fail to accurately quantify the contribution of concomitant pesticide dissipation processes for accurate estimation of off-site pesticide transport (Gassmann et al., 2021)."*
Previously, you provided information on the contribution of processes for the pesticide dissipation process, such as degradation, sorption, and volatilization. I think this term (pesticide dissipation) requires a small paragraph for a definition first because, in the phrase of Ln41, it is not clear what you mean by "the contribution of concomitant pesticide dissipation processes……" Commonly, hydrological models simulate (quantify) "concentrations" over time or "transport"; for example, you can see SWAP-PEARL, HYDRUS, or MACRO models. Therefore, the models do not quantify

pesticide dissipation processes as an output; they are mechanisms **IN** the model. When I read about pesticide dissipation, the first thing that came to mind was DT50 (degradation). Therefore, I suggest including a small explanation of the term, changing it, or citing an author explaining the concept.

We have now revised the sentences to address this comment and clarify how the reactive transport of pesticides, encompassing various processes, leads to pesticide dissipation at the catchment scale, i.e., a reduction in pesticide concentration in the soil compartment (lines 30-38).

*"While pressure on aquatic ecosystems is increasing, the contribution of each pesticide dissipation processes in soil driving to a mass decrease and pesticide off-site transport toward the catchment outlet remains difficult to evaluate and predict accurately. In this context, reliable and validated hydrological models that predict pesticide dissipation and off-site transport have the potential to address fundamental questions, such as the relationship between hydro-climatic factors and pesticide dissipation processes with respects to the transfer risks of pesticides on the catchment scale. The contribution of degradative processes, including both biotic or abiotic degradation, as well as non-degradative pesticide dissipation processes, including sorption, leaching, volatilization and off-site transport, must be quantified for this purpose in models (Larsbo and Jarvis, 2005; Steffens et al., 2015; Gassmann, 2021)."*

**Ln 45** In the phrase "data limitations to constrain model parameters," Please include at the end of this paragraph "by calibration". So, the final phrase should read "data limitations to constrain model parameters by calibration." Remember that several pesticide parameters can be obtained by laboratory analysis.
We have now revised the sentence as suggested

**Ln 55-57** I think this phrase is key for your purpose. Maybe after this phrase, you should provide your opinion and mention that this is one of the research goals, increasing the understanding of your research.
We have added a final sentence in this section to address the challenge for the modelling community (lines 57-59).

*"As a result, reducing uncertainty in DT50 values remains challenging at the catchment scale, despite being crucial for improving the predictive accuracy of pesticide transport models that can serve water quality management".*

**Ln 72** Is the incorporation of CSIA an "alternative" or "complementary" approach?
We have revised the sentence as suggested (line 72-75):

*"In this context, pesticide compound specific isotope analysis (CSIA) offers an alternative approach, as it infers degradation independently of concentration data and transformation product identification, relying instead on changes in stable isotope ratios (Elsner and Imfeld, 2016; Hofstetter et al., 2024)."*

**Ln 138** It mentioned soil samples from 13 marked plots. I GUESS you mean the black dots in Figure 1. We still do not know which samples were obtained from weekly measurements (green dots in Fig. 1). The methodology needs to be organized better.

We have now revised the caption of the Figure 1, and the following sentence, to explain the weekly weighted samples of topsoil for each transect from regular sampling (green dots) along transects (see Alvarez-Zaldivar et al., 2018) (lines 120-122).

*"Figure 1: The Alteckendorf headwater catchment (Bas-Rhin, France), showing the experimental setup, including transects (weighted samples collected at green dots along red lines) and plot sampling (black dots). Land use for 2016 is also displayed. The "Other" category includes roads, grass strips and orchards."*

*Lines 141-144*

*"Topsoils (0-1 cm) were collected from individual plots and upstream-downstream transects across the catchment (Fig. 1 and S1; Alvarez-Zaldivar et al., 2018). Thirteen marked plots were sampled before S-metolachlor application and on days 1, 50, and 100 after application. In addition, weekly weighted composite samples were taken along the north, valley, and south transects (green dots in Fig. 1) from March 19 (day 0) to July 12 (day 115)."*

**Ln 149** I suggest including the total number of observations you have for calibration. You have three types of observations to include in an objective function. Concentrations of S-metolachlor in the topsoil, CSIA, and runoff discharge. It is important to know that number because you are calibrating 25 parameters (which is a lot!) In my experience, you should not calibrate more than five, which still is a lot. Otherwise, you get problems with parameter uncertainties.

Thank you for the suggestion. We have mobilized a total of 103 topsoil samples (i.e., 13 plots × 4 sampling dates + 3 transects × 17 weeks) to measure S-metolachlor concentration and $\delta^{13}C$ values across the catchment and throughout the crop growing season. This effort was aimed at constraining the topsoil dissipation of S-metolachlor, as well as 115 daily discharge measurements to assess the overall hydrological response of the catchment.

The S-metolachlor concentrations and associated $\delta^{13}C$ values monitored at the catchment outlet were not incorporated into the calibration process, as PiBEACH, using daily rainfall data, was not designed to simulate accurate sub-daily intra-event runoff, which accounts for a significant portion of seasonal export (Lefrancq et al., 2012). However, the S-metolachlor concentrations at the outlet, derived from 51 samples, were used for external validation to assess the limitations of PiBEACH, particularly in terms of S-metolachlor export at the outlet (Panel C, Figure 5, and the new section in Materials and Methods). We recognize that calibrating 25 parameters is a considerable undertaking, which justifies the dataset collected across the catchment and throughout the growing season, including the original use of isotopic signatures. We have now addressed the issue of parameter uncertainty with a rigorous methodology, which allowed us to accurately associate our results with the corresponding confidence intervals, as shown in Figures 3, 4, and 5.

We added this information in the manuscript (Lines 409-412):

*"Calibrating 25 parameters is challenging and justifies the extensive dataset collected across the catchment and throughout the growing season. This unique dataset included the use of isotopic signatures and comprises a total of 103 topsoil samples analyzed for S-metolachlor*

*concentration and δ¹³C, 115 daily discharge measurements, and 51 river outlet samples with S-metolachlor concentration."*

**Ln 170**. OK, PiBEACH needs meteorological records, among other things, but it did not provide details of how those measurements were taken or analyzed in laboratory conditions.

We acknowledge the reviewer for this useful comment, and we have now added the following sentences.

In section Field site - Lines 125-126:
*"Daily temperature and potential evapotranspiration were obtained from the MétéoFrance station in Waltenheim-sur-Zorn, located 7 km southwest of the study site. Daily rainfall was measured using a tipping bucket rain gauge (Précis Mécanique®, Fig. 1)."*

*In section PiBEACH development - line 181-183:*
*"The model is driven by daily meteorological records (rainfall, air temperature and potential evaporation), soil physical properties (saturated hydraulic conductivity, bulk density, porosity, both for the plow layer and a deeper soil layer), and crop-specific agronomical parameters."*

**Ln 158** If you mean Supplemental information by "SI", you should put the definition of the acronym one time.

Indeed, thank you. We now use "Supplement" without an acronym, as recommended in the HESS guidelines.

**Ln 178** Can you please mention the mixing topsoil layer length in this phrase? The mixing layer is generally a challenging topic and, in my opinion, an unsolved problem for hydrological models.

This information (first centimeter, from 0 to 1 cm) was originally provided 25 lines after the line 178. We have now included this information as recommended in line 187-194:

*"Key modifications in PiBEACH include: (i) a topsoil mixing layer, assuming that pesticide transport is controlled by a mixing layer from 0.25 to 2 cm below the soil surface, in which rainfall, soil solution and runoff are assumed to mix instantaneously (McGrath et al., 2008), and deeper soil layers to capture groundwater contributions to discharge, (ii) daily simulation of topsoil temperature (Neitsch et al., 2011), (iii) daily dynamic topsoil hydraulic properties that reflect crop-specific agronomical practices (Lefrancq et al., 2018), (iv) first-order pesticides degradation and linear sorption/desorption, and (v) pesticide carbon stable isotopic fractionation and transport of bulk heavy and light carbon in the pesticide molecule (e.g., ¹³C and ¹²C) (Lutz et al., 2017; Alvarez-Zaldivar et al., 2018). PiBEACH requires spatially explicit inputs of pesticide application dates and quantities for each plot."*

As mentioned by the reviewer, the mixing layer concept is probably the most commonly used in pesticide fate models due to its numerical simplicity and its realistic representation, as hortonian runoff processes have repeatedly been shown to be initiated as a near-surface process (Gao et al., 2004; McGrath et al., 2008; Wallender et al., 2008). This concept assumes that the transport is controlled by a mixing layer below the soil surface, in which rainwater,

soil solution and runoff water completely and instantaneously mix (Havis et al., 1992). The soil depth interacting with runoff water is variable ranging from 0.25 to 2 cm according to various experimental and modelling studies (Ahuja et al., 1981; Havis et al., 1992; Wallender et al., 2008). Previous studies have shown that the effective depth of interaction is related to the degree of soil aggregation and increases with soil slope, kinetic energy of raindrops and rainfall intensity (Havis et al., 1992). In fact, the observed mixing-layer depth is often much shallower than the depth required for fitting models to field data (Gao et al., 2004).

Ahuja, L.R., Sharpley, A.N., Yamamoto, M., Menzel, R.G., 1981. The Depth of Rainfall-Runoff-Soil Interaction as Determined by P-32. Water Resources Research. 17, 969-974.

Gao, B., Walter, M.T., Steenhuis, T.S., Hogarth, W.L., Parlange, J.Y., 2004. Rainfall induced chemical transport from soil to runoff: theory and experiments. Journal of Hydrology. 295, 291-304.

McGrath, G.S., Hinz, C., Sivapalan, M., 2008. Modeling the effect of rainfall intermittency on the variability of solute persistence at the soil surface. Water Resources Research. 44.

Wallender, W.W., Joyce, B.A., Ginn, T.R., 2008. Modeling the Transport of Spray-Applied Pesticides from Fields with Vegetative Cover. Transactions of the Asabe. 51, 1963-1976.

Havis, R.N., Smith, R.E., Adrian, D.D., 1992. Partitioning Solute Transport between Infiltration and Overland-Flow under Rainfall. Water Resources Research. 28, 2569-2580.

The reference (McGrath et al., 2008) cited at the first mention of the mixing layer in our paper was included to provide a more in-depth understanding of this concept.

Lines 187-189:

*"Key modifications in PiBEACH include: (i) a topsoil mixing layer, assuming that pesticide transport is controlled by a mixing layer from 0.25 to 2 cm below the soil surface, in which rainfall, soil solution and runoff are assumed to mix instantaneously (McGrath et al., 2008), and deeper soil layers to capture groundwater contributions to discharge,…"*

**Ln 201** The Model only accepts that number of layers. Or is that set by the user?

We relied on the pedological, agronomic, and hydrological knowledge of the studied catchment to develop the conceptual framework for PiBEACH. The number of layers chosen, i.e. five for the Alteckendorf catchment, seems to be the optimal number to simplify vertical heterogeneity based on agronomic criteria (cropping operations), pesticide reactivity, or hydrological functioning with variably saturated and saturated zones. Based on similar expertise, this number of layers could be adjusted for another catchment and implemented in a modified version of PiBEACH.

**Ln 220** In Eq.1, you mentioned that the balance is solved for each layer with soil depth Z. But the runoff term should not be included in deeper layers. This raises the question of whether you obtain a unique theta per cell or a theta for each layer (5 layers in the profile per cell). It would help if you clarified this, especially for HYDRUS, MACRO, openLISEM, or SWAP-PEARL users.

As noted in line 178 and in Equation 1, PiBEACH calculates the hydrological balance for each cell and layer. The hydrological balance is described in Sheikh et al. (2009), with a detailed implementation provided for each process in Section 6 of the SI. In general, the change in soil water content ($\theta$, m³ m⁻³) for each cell (i) and each layer, defined by soil depth (Z, mm), is calculated using Equation 1. The resulting soil water content is then available for each cell and

layer, facilitating potential coupling with other models such as HYDRUS, MACRO, OpenLISEM OLP, or SWAP-PEARL.

We have moved Equation 1 to follow the description of the layers (line 226) in order to clarify how each term in the equation is applied depending on the specific layer. As noted by the reviewer, we have added a sentence explaining that runoff (RO) is only computed for the topsoil layer ($z_0$).

Line 222-233:

*"The hydrological balance follows the formulation of Sheikh et al., (2009), with process-specific implementations detailed in Section 6 of the Supplement. For each grid cell i and soil depth Z (mm), the daily change in volumetric soil water content ($\vartheta$, $m^3\,m^{-3}$) is calculated using:*

$$Z\Delta\theta_i = R_i - RO_i + \Delta LF_i - Ea_i - Ta_i - P_i \qquad (1)$$

*where daily rainfall (R), runoff (RO) (calculated for $z_0$), net cell lateral inflow-outflow ($\Delta LF$), actual evaporation (Ea), actual transpiration (Ta) and percolation (P) were expressed in mm $H_2O\ d^{-1}$. Evaporation (Ea) affects layers $z_0$ to $z_3$, while transpiration (Ta) depends on plant root depth following crop-specific development stages (see sections 6.5 and 6.6 in the Supplement). Lateral flow ($\Delta LF$) and percolation (P) are calculated from upstream to downstream cells for layers $z_0$ to $z_3$ (Manfreda et al., 2005). Percolation across the saturated layer $z_4$ is routed to the outlet as a global linear reservoir (Manfreda et al., 2005). "*

**Ln 230:** You mentioned the balance at each cell I was given by Eq. 2. My question is, does this model solve the mass balance per layer (5 in your case), or does it generate a mesh with smaller cells? (Connected with the previous question)

As stated in line 213, equations 1 and 2 are solved for each cell of each layer $z_0$ to $z_3$ using a raster approach with a resolution of 2 x 2 m for layers $z_0$ to $z_3$ (four layers in total). The deeper layer, $z_4$, is represented using a global linear reservoir.

The sentence was revised to clarify the application of the raster approach with a 2 x2 m resolution for solving the equations.

*"To account for landscape features affecting pesticide off-site transport, such as grass strips or roads, the Alteckendorf catchment was modelled using a 2 × 2 m raster resolution. The vertical soil profile was divided into five successive layers from the topsoil to the groundwater (Fig. 2):"*

**Ln 246.** Where is the mixing layer depth used in Eq. 4? I think you should change *J* by *qx,y* in this equation.

The equation 4 has now been corrected, and the depth of the mixing layer is now explicitly linked to $D_{zo}$.

Line 254-264:

"General pesticide mass flux J (µg d$^{-1}$) for each model cell is computed as:

$$J = q_{x,y} \times C_{aq} \qquad\qquad (3)$$

where $q_{x,y}$ is the water flux vector (mm d$^{-1}$) in the lateral (x) and vertical (z) direction and $C_{aq}$ is the dissolved S-metolachlor concentration in the aqueous phase (µg mm$^{-1}$). In the topsoil layer ($z_0$), runoff and volatilization are included:

$$J_{zo} = q_{x,y} + c_{aq}\left( RO\, e^{-\beta_{RO} D_{zo}} + \frac{1}{r_a + r_s} \right) \qquad\qquad (4)$$

where RO is runoff (mm), $\beta_{RO}$ is a calibration constant (1 ≥ $\beta_{RO}$ ≥ 0) and $D_{z0}$ (mm) is the mixing layer depth (Ahuja and Lehman, 1983). Volatilization was limited to the first 5 days after application (Gish et al., 2011), (Prueger et al., 2005), and follows (Leistra et al., 2001), with fluxes across topsoil controlled by air transport resistance, $r_a$, (d m$^{-1}$) and diffusion resistance, $r_s$ (d m$^{-1}$) (Leistra et al., 2001) (see sub-section 8.2 of the Supplement).

**Ln 296** It is not better to include href than $\theta ref$. The water content depends highly on texture; 0.2 cm3 cm-3 of water content could be either high or low. Instead, the h does not have that inconvenient."

We acknowledge this suggestion and we agree that water content depends on soil texture. However, we have retained and applied in this study the formulation developed by Walker (1974) and Larsbo and Jarvis (2003), which uses a ratio of soil moisture expressed in θ (m³ m⁻³) rather than h (mm H₂O). The reference value, θref, corresponds to the experimental conditions associated with the DT₅₀ref (and k_ref) characterization. In these lab-scale experiments, soil saturation is typically expressed in terms of soil moisture.

**Ln 305** OpenLISEM cannot simulate preferential flow yet, to my knowledge. So, this phrase required modification.

We agree with the reviewer's assessment. OpenLISEM and its pesticide module OLP are not capable of simulating preferential flow and are better suited for catchments with a significant risk of runoff and associated erosion, such as in Alteckendorf during storm events. As a result, we have revised the sentence to focus solely on runoff.

Line 325-330:
"First, PiBEACH operates on a daily time step and is not suited to capture rapid flow dynamics, such as runoff generation and event-scale hydrological dynamics (Sheikh et al., 2009). Consequently, S-metolachlor concentrations and corresponding $\delta^{13}C$ values measured at the catchment outlet were not included in the model calibration objective function. This limitation could be addressed by coupling PiBEACH with a distributed event-based model, such as the Limburg Soil Erosion Model (OpenLISEM) (Baartman et al., 2012), which recently integrated a pesticide module (OLP) (Commelin et al., 2024)."

**Ln 350** Which threshold did you use to reduce parameters from 43 to 25. Basically, how did you define a parameter as highly sensitive? Additionally, you should mention which outputs were analyzed during sensitivity analysis, i.e., did you include sensitivities to isotope output?

We have calculated and analyzed the mean and standard deviation of the elementary effects (EE) for each parameter, as required by the Morris method. The mean value represents the overall effect of each parameter on the model output, while the standard deviation reflects the interaction between inputs. We excluded 21 parameters where the mean value was zero or close to zero, indicating no significant influence on the output. The outputs considered in this sensitivity analysis included S-metolachlor concentrations at the outlet, composite soil transects (North, Valley, and South), discharge at the outlet, and isotope signatures at both the outlet and composite soil transects. The mean and standard deviation values for the 25 retained parameters are presented in Figures S5 and S6 in the Supplement, for the different output variables.

**Ln 370** Where did you previously mention "daily discharge at the outlet"? You mentioned weekly runoff. The observations are unclear, and you need to structure the methodology better.

As described in a previous paper (Alvarez-Zaldivar et al., 2018), *"discharge at the catchment outlet was continuously measured by a Doppler flow-meter (2150 Isco, Lincoln, Nebraska, USA) with discharge precision of 3% and a time step of 2 minutes. Water was collected by flow proportional sampling using a refrigerated automatic sampler with a total capacity of 3.96 L divided into 12 glass vials each of 330 mL (Isco Avalanche, Lincoln, Nebraska, USA). A predefined discharged volume based on the seasonal rainfall intensity expected for April (50 m³), May (100 m³) and June (150 m³) was chosen allowing for 36 aliquots of 110mL each per week. Water samples were then combined into composite samples according to hydrograph characteristics (base-flow, rising and/or falling limb), yielding one to four samples weekly of volumes >= 990 mL. "*

The weekly time step refers only to the field visits for collecting and replacing samples, not to the sampling frequency used to acquire data. We have revised the relevant sentences to clarify this distinction.

*Line 150-156:*

*"Runoff discharge at the catchment outlet was measured using a Doppler flowmeter (2150 Isco) with 3% accuracy and a 2 min resolution. Automatic, refrigerated continuous flow proportional sampling was performed using an Isco Avalanche autosampler equipped with twelve 330 mL bottles. Samples were collected at fixed weekly discharge volumes from 50 to 150 m³, capturing the increasing baseflow discharges from April to June 2016 (Alvarez-Zaldivar et al., 2018). To obtain sufficient S-metolachlor for quantification and CSIA, samples were combined weekly into composite samples based on hydrograph phase (base-flow, rising limb, and falling limb), resulting in one to four samples per week with volumes ≥ 990 mL (Alvarez-Zaldivar et al., 2018)."*

**Ln 373** OK, you explain that in addition to incorporating CSIA data in the objective function, you will analyze the spatial distribution of this observation. For better understanding, you should use Fig 1 at this point.

As suggested by the reviewer, we have revised the relevant sentence to clarify this point.

Lines 388-390:

*"For the latter, KGE metric incorporates topsoil S-metolachlor concentration and δ$^{13}$C from (i) individual plot observations (13 black dots, Fig. 1), (ii) aggregated plot observation along three transects across the catchment (North, Valley and South transect, Fig. 1) and (iii) transect composite soil (green dots, Fig. 1)."*

**Ln 393** If the graph (Fig 3C), the "γ" axis is in DT50, please use DT50 instead of $k_{eg}$ during the discussion. Additionally, I do not know how this graph and analysis (Fig 3C) provide useful information for the research. All that you mentioned in the graph and discussion can be easily obtained from Eq. 8, derived from another hydrological model, so basically, it is something simple and well-known.

As suggested by the reviewer, we have now revised the relevant sentence, using $DT_{50}$ instead of $k$ to discuss the temporal variation in S-metolachlor persistence.

Lines 440-445:

*"Hydro-climatic variability drove changes in the simulated degradation rates ($k_{Dynamic}$) and associated half-times ($DT_{50,Dynamic}$) of S-metolachlor over time (Fig. 3C). Low S-metolachlor degradation rates during the first 10 days ($DT_{50} = 46 \pm 17\ d$) coincided with cold (7.1 ± 1.7 °C) and dry period (7.4 mm of rainfall) (Fig. 3A). In contrast, increasing degradation rates ($DT_{50} = 22 \pm 2\ d$) between 50 to 120 days following the first application of S-metolachlor were associated with warmer conditions (16.9 ± 3.6 T°C)."*

**Ln 400** The graph should be improved **(.svg)**. Additionally, you mention the error bars linked to soil bulk density. I think this is important and should be mentioned in the methodology (please improve the organization of methods).

The figures 3, 4, 5 will be send in HD in the final version.

As stated in the caption of Figure 3, both observed and predicted soil water content are expressed in m³ water m⁻³ soil. Additional details have been provided in the Materials and Methods section, explaining how gravimetric measurements obtained in the laboratory (drying samples in an oven set at 110°C, NF ISO 1146) were converted into volumetric values using soil bulk density. The topsoil bulk density fluctuates throughout the growing season and was estimated daily based on the reference soil bulk density measured below the plow layer. (1.5 ± 0.09 kg m⁻³) and the agronomic module of PiBEACH (Section 7.2 in the Supplement), which was previously validated in Lefrancq et al. (2018, HP).

This information was added in lines 145-149:

*"Volumetric topsoil water content ($m^3_{water}$ $m^{-3}_{soil}$) was derived from gravimetric measurements after drying at 110°C following NF ISO 1146 (Lefrancq et al., 2018). This calculation incorporated seasonal variations of topsoil bulk density, as predicted by PiBEACH and described in the Supplement, Section 7.2."*

In caption of Figure 3 we have revised the targeted sentence (lines 448-456):

*"**Figure 3:** Predicted water content ($m^3$ $m^{-3}$) and relative topsoil temperatures ($T_{z0}/T_{air, max}$ ; with $T_{air, max}$ = 27°C) in the topsoil layer ($z_0$: 0-1cm) from March 14 (day -5, before first application) and July 12 (day 115). Shaded area indicates the 95 % confidence intervals (CI). Weekly observed water content ($\theta_{obs}$) in topsoil is also shown (A), with error bars reflecting uncertainties in soil bulk density (used for gravimetric-to-volumetric conversion) and the standard error of gravimetric water content measurements. (B) Simulated and observed daily outlet discharge at the catchment scale, with the shaded area representing 95 % CI of the model ensemble. (C) Simulated mean dynamic degradation half-life ($DT_{50, Dynamic}$) of S-metolachlor in the topsoil ($z_0$), 95 % CI and the reference half-life ($DT_{50, Ref}$) shown for comparison. (D) Simulated and observed S-metolachlor concentrations composite topsoil transects ($z_0$), with error bars showing the standard deviation of measured values. (E) Simulated and observed $\delta^{13}C$ values of S-metolachlor in topsoil, with error bars showing the standard deviation. Application periods are marked as App. 1 (days 0 and 6), App. 2 (day 25), and App. 3 (days 67 and 74)."*

**Ln 404** Please indicate if metolachlor concentrations are in liquid or total concentrations (also, this should be explained in the methodology).

We agree with the reviewer's that information on S-metolachlor concentration was missing. Although the filtration methodology was already described (see below and in line 183 of the manuscript), we have now clarified the methodology for river samples and added a sentence (Line 557) in the results section to clarify this point. S-metolachlor concentrations in the particulate phase (i.e., > 0.7 µm) collected in the river were consistently below the quantification limit. Therefore, S-metolachlor concentrations used in the calibration step for the 2016 campaign were based solely on the dissolved phase of the samples collected in the river at the outlet of the catchment.

Line 159-164:
*"To separate dissolved and particulate phases of S-metolachlor, river samples were filtered through 0.7 µm glass fibre filters. Extraction of S-metolachor from soil and water samples was performed using an AutoTrace 280 Solid Phase Extraction (SPE) system (Dionex) with SolEx C18 cartridges (Dionex ®), as previously described (Alvarez-Zaldivar et al., 2018; Gilevska et al., 2022b). Quantification was carried out using chromatography–mass spectrometry (GC-MS, ISQ™, Thermo Fisher 173 Scientific). Quantification limits were 0.01 µg $L^{-1}$ for water and 0.001 µg $g^{-1}$ (dry weight) for soil and suspended matter, with a total analytical uncertainty of 8 % and 16 %, respectively."*

New sentence (line 520-522)

*"It is important to note that observed export estimates were based solely on the dissolved phase, as particulate-bound S-metolachlor (> 0.7 µm) remained below quantification limits in all river samples."*

**Ln 420** OK WIC calibration reduced uncertainty, but what happens with the other 24 parameters? In the case of NIC or WIC calibration, do they maintain the uncertainty bounds? Or perhaps it changes, and how? This is important to analyze because that uncertainty may be translated to other parameters. Additionally, is there some field (laboratory) information on the common parameter values for this compound? especially DT50. I agree that DT50 could be one order of magnitude different from the expected one. Still, it is welcome to have the laboratory reference to see if calibration did not modify parameters to "unexplainable" values. That also holds for the 24 remaining parameters that you calibrate. That kind of analysis is expected from this kind of research.

We hypothesized that $DT_{50, \text{reference}}$ was the primary parameter affected by incorporating stable isotope signatures, as it is the sole parameter governing degradation in PiBEACH. As expected, $DT_{50, reference}$ was affected. Specifically, its uncertainty bounds were reduced when $\delta^{13}C$ was included in the objective function (WIC calibration). The newly calibrated WIC range for $DT_{50, \text{reference}}$ (18 ± 4 days) remained consistent with reported experimental values in the literature. In contrast, the uncertainty bounds for the other 24 parameters were not significantly altered when comparing NIC and WIC calibrations (data not shown in the current version).

In response to the Reviewer's comment, we have now added a new section in the Supplement (Section 12). This includes a new Figure S8, demonstrating, using $K_{OC}$ as an example, that CSIA data (WIC) did not reduce the uncertainty of 24 other parameters compared to NIC. This may be due to the pesticide extraction protocols used for soil samples, which account for both the dissolved phase and the extractable sorbed phase. Additionally, sorption generally does not lead to significant isotope fractionation (see also the response to Questions 2 and 3 in the Introduction, addressed by Reviewer 2). In this case, CSIA may not help evaluate sorption and thus constrain $K_{OC}$ values as isotope signatures for pesticides in the dissolved and particulate phase of soil samples are generally similar.

*"12 KOC sensitivity*

*CSIA information in soils did not permit to reduce uncertainty for $K_{OC}$ values across all sample resolutions. In a virtual experiment evaluating leaching extent based on $DT_{50}$ and $K_{OC}$ correlation scenarios, Lindahl et al., (2008) find that when $DT_{50}$ and $K_{OC}$ were negatively correlated, larger variance in leaching extent was observed in the field, as lower degradation rates complement with higher mobility. In our study, $DT_{50}$ and $K_{OC}$ values were negatively correlated (-0.47, P<0.001), suggesting that spatial variations in organic carbon significantly altered mobility and degradation (Wu et al., 2012) as previously observed for S-metolachlor (Rice et al., 2002; Long et al., 2014). Namely, although catchment $K_{OC}$ values below 500 L/Kg could be discarded on average (i.e., as shown by WIC models in bulk soils, Fig. S8), improvements in degradation parameter constraints based on temperature and moisture alone were not useful to constrain spatial variability of $K_{OC}$ values (i.e., as shown by transect*

and plot $K_{OC}$ distributions, Fig. S8). More detailed and explicit representation of organic carbon content evolution in both space and time using available information such as soil type, land-use and agricultural management (Meersmans et al., 2011), as it was done in this study for soil hydraulic properties, could further help constraining spatial variability of degradation rates and mobility parameters regulating pesticide leaching.

[Figure]

*Figure S8: Distribution (out of a total of n = 2,500 runs) of $K_{OC}$ calibrated with no isotope constraint (NIC, n = 672) and with isotope constraint (WIC, n = 244) at three sampling resolutions (i.e., composite transect, transect and plot soils). NIC models considered $KGE_{SM} > 0.5$ and $KGE_Q > 0.5$, while WIC models considered $KGE_{SM} > 0.5$ and $KGE_Q > 0.5$ and $KGE_\delta > 0.8$. Statistics for $K_{OC}$ distributions are provided as mean (blue for WIC, purple and red for NIC, with $DT_{50\ dynamic}$ depending of soil moisture and temperature or $DT_{50\ constant}$, respectively) and standard deviations (μ ± SD).*

*With new references:*

Lindahl, A. M. L., Soderstrom, M. and Jarvis, N.: Influence of input uncertainty on prediction of within-field pesticide leaching risks. J. Contam. Hydrol. 98, 106, https://doi.org/10.1016/j.jconhyd.2008.03.006, 2008.

Long, Y. H., Li, R. Y. and Wu, X. M.: Degradation of S-metolachlor in soil as affected by environmental factors. J. Soil Sci. Plant Nutr., 14, 189-198, http://dx.doi.org/10.4067/S0718-95162014005000015, 2014.

Meersmans, J., VanWesemael, B., Goidts, E., Van Molle, M., De Baets, S., De Ridder, F.: Spatial analysis of soil organic carbon evolution in Belgian croplands and grasslands, 1960-2006. Glob. Change Biol., 17, 466-479. https://doi.org/10.1111/j.1365-2486.2010.02183.x, 2011.

Rice, P. J., Anderson, T. A. and Coats, J. R.: Degradation and persistence of metolachlor in soil: effects of concentration, soil moisture, soil depth, and sterilization. Environ. Toxicol. Chem. / SETAC 21, 2640-8. https://doi.org/10.1002/etc.5620211216, 2002.

Wu, X., Li, M., Long, Y. and Liu, R.: Effects of adsorption on degradation and bioavailability of metolachlor in soil. J. Soil Sci. Plant Nutr., 11, 83-97. http://dx.doi.org/10.4067/S0718-95162011000300007, 2011.

**Ln 439**. OK, it would be nice to see if there are significant differences in the comparison variance. The plot situation showed a smaller reduction in the SD. The "plot" may not be recommended, but you need ANDEVA or something like that for that affirmation.
We agree with the reviewer's observation. The $DT_{50}$ distributions calibrated with and without isotopic constraints were similar at the plot resolution but differed at the transect and transect composite sample resolutions. We have now entirely revised this statement to avoid any affirmation.

Line 486-494:

*"The mean calibrated $DT_{50,Ref}$ values for transect composite soil, individual transects across the catchment, and plot resolutions differed by less than four days (Fig. 4). However, standard deviations reflecting uncertainty, were approximately 50% lower when isotopic constraints (WIC) were applied, compared to calibrations without isotopic constraints (NIC), for both transect composite and transect-scale data. At the plot scale, uncertainty remained unchanged. These results indicated that increasing resolution (plots > transects > transect composite samples) did not significantly improve model calibration in this catchment. Therefore, pooling transect samples into a single composite soil sample was sufficient for CSIA-based calibration, while also reducing sampling and analytical efforts. This approach may reliably capture variations in pesticide degradation at the catchment scale."*

**Ln 444** "Modeling results": Do you mean calibration results? Ec was calibrated? In this case, did you compare it with previous studies on that parameter? Something like that should be expected for DT50 and perhaps other calibrated parameters. (Similar comment that above)

We have now revised the sentence to clarify that $\varepsilon_c$ was indeed one of the 24 calibrated parameters, and the calibrating value is closed to the experimental reference derived from lab-scale study on the Alteckendorf soil (Alvarez-Zaldivar et al., 2018). In response to the Reviewer's question (see Point 16 in Reviewer 2's section), we have now clarified the different values of the isotope fractionation factor ($\varepsilon_C$) used in this study.

Line 495-504:
*"Integration of CSIA data into PiBEACH yielded an isotope fractionation factor of $\varepsilon C = -2.7 \pm 0.6$ ‰ for S-metolachlor degradation in topsoil, exceeding previously reported values (Alvarez-Zaldivar et al., 2018: $-1.5 \pm 0.5$ ‰; Lutz et al., 2017: $-1.3 \pm 0.5$ ‰). This discrepancy likely reflects differences in environmental conditions, microbial activity, and transformation pathways, though it remains within the model ensemble's uncertainty range. Compared to laboratory-derived estimates, the stronger fractionation observed suggests either alternative*

*degradation pathways or the influence of distinct microbial consortia at the catchment scale, relative to laboratory conditions. These findings underscore the importance of site-specific calibration in CSIA applications and highlight the value of model ensemble approaches in capturing the range of degradation processes in heterogeneous agro-ecosystems. While previously reported εC values may slightly overestimate degradation in some field settings, ensemble modelling with integrated CSIA data provides a more robust, field-relevant assessment of pesticide transformation."*

**Ln 457** Why did you mention that degradation obtained with the model was like observations? In which part of the methodology did you explain that compound degradation was performed under laboratory conditions or other? You need to improve the structure of the article.

We thank the reviewer for their comments. We have now carefully checked and reorganized the structure of the article by adding a new section to the Materials and Methods, which introduces the six metrics. Additionally, we have now provided a detailed explanation of how the observed values, i.e. specifically topsoil concentration, $\delta^{13}C$, and discharge and concentration in river at the outlet, were used to derive the observed values for three of the six metrics.

Lines 416-429:  (attention decalage des pages 416 au dessus à reprendre avant)

"2.7 Metrics of pesticide persistence and transport risks

Six metrics were derived from PiBEACH outputs to assess S-metolachlor persistence and transport risk at the catchment scale. Expressed as percentages of the applied mass, these metrics included: (1) S-metolachlor degradation, (2) the remaining bioavailable mass of S-metolachlor (BAM) in the topsoil (i.e., dissolved and reversibly sorbed S-metolachlor), (3) S-metolachlor export to the catchment outlet via runoff and drainage, (4) off-site transport of S-metolachlor through leaching, (5) the remaining mass of irreversibly sorbed S-metolachlor (i.e., aged S-metolachlor), and (6) S-metolachlor volatilization, from the first application on March 19 (day 0) to July 12, 2016 (day 115). The first three metrics were also derived from field observations to evaluate the predictive accuracy of PiBEACH. Observed BAM in the topsoil was estimated through spatial extrapolation of weekly soil samples. The observed S-metolachlor degradation was estimated using the mean $\delta^{13}C$ derived from the same spatial extrapolation, combined with the enrichment factor (ε) obtained from microcosm lab-scale experiments using soil from Alteckendorf. Export to the outlet was calculated by integrating continuous discharge measurements with sub-weekly river S-metolachlor concentrations. The remaining three metrics, leaching, volatilization, and aging, could not be estimated from field data due to limitations of the catchment-scale sampling campaign and were therefore not computed from observations."

**Ln461** Here, we explain how the BAM% was obtained, but this is a methodological part.
This explanation is now included in the newly added Materials and Methods section.

Line 416-419:

*"Six metrics were derived from PiBEACH outputs to assess S-metolachlor persistence and transport risk at the catchment scale. Expressed as percentages of the applied mass, these metrics included: (1) S-metolachlor degradation, (2) the remaining bioavailable mass of S-metolachlor (BAM) in the topsoil (i.e., dissolved and reversibly sorbed S-metolachlor), …"*

**Technical comments**
**L10.** to the limited availability of field data **capable of distinguishing** between. Please change by "distinguishing".
As suggested by the reviewer, we have revised the sentence:

Lines 9-10:
*"Predicting pesticide dissipation at the catchment scale using hydrological models is challenging due to limited field data distinguishing degradative from non-degradative processes."*

**L17.** incorporating changes **of** the carbon isotopic signatures. Please change "of" by "in".
we modified the sentence as suggested

**L36.** leaching, volatilization, off-site transport, Please included **"and"** before off-site.
we modified the sentence as suggested

L46. each of which generally **accounting** for several physico-chemical. Please modify by accounts.
we modified the sentence as suggested

L59. in topsoil and at the catchment outlet **is** currently. Please change "is" by "are".
we modified the sentence as suggested

L70. it may also generate unknow and. Please change "unknow" by "unknown".
we modified the sentence as suggested

And some more that you can check.

We have thoroughly revised the manuscript to correct any spelling errors.
* * *
**General comment**

Payraudeau et al. present a novel modelling study advancing the field of simulation of pesticide and isotope ratios at catchment scale. Studies as this one are rarely conducted but very important to understand more on pesticide fate at catchment scale. I have two main concerns that I strongly recommend that the authors take into consideration. These are elaborated below.

We thank Reviewer 2 for the relevant comments. In response, we have now revised in details the manuscript, specifically addressing the two main concerns: (i) the added benefits of this

distributed model approach compared to the earlier lumped (non-spatially distributed) model study by Lutz et al. (2017), and (ii) the use of pesticide concentrations and isotope ratios at the catchment outlet.

**Main comments**

1. A key novelty of this study is the application of a spatially distributed catchment model to simulate pesticide concentrations and isotope ratios (only in topsoil unfortunately, see main comment #2), as stated also on line 92. My first main critique to this study is that the added benefit of a spatially distributed model for this purpose is barely demonstrated nor discussed, albeit expectations are raised in lines 87-89. In that sense the added benefit of this distributed model approach compared to the earlier lumped (not spatially-distributed) model study by Lutz et al. (2017) - applied to the same catchment but with an earlier dataset limited to catchment outlet (but no topsoil) data - does not become clear. The focus in the model is more on linking degradation rates to temporal variations in temperature and soil moisture but that can also be simulated with a lumped model approach. There must be spatial heterogeneity in the catchment like in pesticide application at plot level, but also in spatial variation in soil texture and soil moisture content (Figure 1 nicely shows the spatial variations in crop types; and the descriptive text mentions various soil types). The added benefit of simulating such spatial variations much better in a distributed model than in a lumped model is not discussed. I recommend that the authors show and discuss this key advantage better as that seems the main novelty of this study. Or could a lumped model not have resulted in an equally good model fit as page 5 states that the spatial variation in soil parameters was in fact low in this (specific) catchment?

PiBEACH was initially developed as a companion model for distributed event-based pesticide transport models, such as OpenLISEM OLP (Commelin et al., 2024), to enhance understanding and support management decisions and risk assessments at the headwater catchment scale. Event-based models are valuable tools for predicting intra-hourly rainfall-runoff dynamics and the associated off-site transport of pesticides. However, their accuracy depends on the initial conditions, which are affected by hydrological dynamics and pesticide dissipation processes occurring between rainfall events. PiBEACH was designed to predict catchment properties that evolve throughout the growing season and across the catchment, such as plow layer bulk density, soil moisture, and pesticide concentrations in both dissolved and particulate fractions. PiBEACH considers the spatial distribution of (i) agricultural plots with different crops and associated pesticide applications based on farmers' technical practices (e.g., pesticide type, application date, and dosage, as shown in Fig. 1 and Table S2 in the Supplement), (ii) non-target areas, i.e. areas where pesticide applications are not intended, such as grass strips, hedges, and roads. (Fig. 1), which affected runoff generation and propagation (Lefrancq et al., 2017), (iii) different soil types and slope profiles (Lefrancq et al., 2017), and (iv) soil moisture and temperature dynamics as a function of hillslope position.

While lumped models can capture pesticide dynamics at the catchment scale, as highlighted by Lutz et al. (2017), we hypothesized that only a distributed model can effectively support management decisions related to the crop mosaics within the catchment to mitigate pesticide fluxes at this scale.

In summary, both lumped and distributed approaches can be used to investigate pesticide dynamics throughout the growing season. However, only a distributed model can account for spatial heterogeneity across the catchment and support further management decisions.

As outlined in the following responses to the Reviewer's comments, we have now clarified in the manuscript the complementarity of these two approaches, which were developed and applied to the same catchment.

My second main critique is that the authors did not make use of the pesticide concentrations and isotope ratios at the catchment outlet (besides the topsoil data) as also reported by Alvarez-Zaldivar et al. (2018). I strongly recommend that the authors add the simulation of the catchment outlet data as reported by Alvarez-Zaldivar et al. (2018) to this work. I see this as a pre-requisite to allow to draw conclusions on pesticide degradation at the catchment scale. The model is now only calibrated to observations on the first cm of soil in the catchment. The (degradation) model is not calibrated to anything deeper than this top one cm. Calibrating the model also to the available catchment outlet data enables to draw conclusions on pesticide degradation at the whole catchment scale, and would make this a really unique study. All conclusions drawn on catchment scale are in the current version based on extrapolation of the topsoil data. Also include a discussion on the pros/cons of calibrating such models with only topsoil data (as done in this current work) vs. outlet data (as done by Lutz et al. 2017), or both. See later "other comments" that elaborate on this main comment #2.

We acknowledge the comment of the Reviewer to clarify this issue in the manuscript. We used river discharge at the outlet in the calibration function and the S-metolachlor fluxes to examine the predictability of PiBEACH (Fig. 5C). As suggested by Reviewer 1, we have now reorganized the article's structure by adding a new section in the Materials and Methods section, which introduces the six metrics simulated after calibration and derived from soil and river samples. Additionally, we have provided a detailed explanation of how the observed values, including discharge and S-metolachlor concentration at the outlet, were used to derive the observed values for three of the six metrics. As expected, given PiBEACH's limited ability to capture intra-daily runoff generation, it is unable to accurately simulate S-metolachlor fluxes at the catchment scale (Fig. 5C), representing less than 2% of the applied amount. Therefore, we opted not to include outlet concentration and isotopic signature, mainly collected during rainfall-runoff events (Alvarez-Zaldivar et al., 2018), in the calibration function to avoid a biased calibration process, which could lead to parameter fitting for the wrong reasons in an attempt to compensate for the model's conceptual limitations.

As emphasized by the reviewer, we prioritized topsoil data during the PiBEACH calibration process, specifically using 103 samples with S-metolachlor concentration and isotopic signatures. We derived two internal variables, i.e. predicted soil temperature and water content, to extrapolate degradation across time, space, and depth. The degradation dynamics, incorporating soil temperature and water content in PiBEACH, was validated through specific lab-scale experiments on Alteckendorf soil (see Supplement, Section 9). The results enhanced the interpretation of S-metolachlor export, considering the daily limitations of PiBEACH, with predicted and observed export rates of 2 ± 6% and 0.3% of the applied S-metolachlor, respectively.

To summarize, we leveraged a substantial portion of the data collected at the catchment outlet for both calibration (discharge in the KGE objective function) and predictability assessment (discharge and S-metolachlor concentrations to estimate export). Due to the conceptual limitations of PiBEACH, particularly its daily time step, the comparison between simulated and observed S-metolachlor reactive transport was only partially conducted, in contrast to previous lumped modelling approaches (Lutz et al., 2017).

The added value of integrating both observed S-metolachlor concentrations and isotopic signatures, at the outlet and in the topsoil, into the calibration process of distributed modelling will be further explored in the upcoming PiBEACH-OpenLISEM OLP modelling study. This study is now in collaboration with Wageningen University, the developer of the OLP pesticide module (Commelin et al., 2024).

This second major concern of the Reviewer, along with the relevant data from the river outlet, has now been incorporated into the revised manuscript, as detailed below.

**Additional comments:**

1. Title: why "models"? Plural? It seems only one model has been applied?

As suggested by the reviewer, we have now revised the title to use the singular form of model.

Lines 1-2:

*"Constraining topsoil pesticide degradation in a conceptual distributed catchment model with compound-specific isotope analysis (CSIA)"*

2. Introduction:
   1. Line 72: Explain the concept of CSIA better such that the reader can understand how information on degradation can be derived from isotope ratios.

We have revised the following sentence to clarify the connection between isotopic signature and degradation:

*Lines 72-83:*

*In this context, pesticide compound specific isotope analysis (CSIA) offers an alternative approach, as it infers degradation independently of concentration data and transformation product identification, relying instead on changes in stable isotope ratios (Elsner and Imfeld, 2016; Hofstetter et al., 2024). During pesticide degradation, lighter isotopes (e.g., $^{12}C$) typically react slightly faster than heavier ones (e.g., $^{13}C$), producing a kinetic isotope effect. This effect creates a distinct isotopic signature, reflected in changes in the isotope ratios of degrading molecules (Elsner, 2010). The stable isotope fractionation ($\varepsilon_{lab}$, e.g., for carbon) can be determined through closed-system microcosm experiments and applied under specific conditions to quantify pesticide degradation under field conditions (Alvarez-Zaldivar et al., 2018). In contrast, non-destructive processes like dispersion and sorption are not expected to induce significant isotope fractionation under agricultural field conditions (Eckert et al., 2013; van Breukelen and Prommer, 2008; van Breukelen and Rolle, 2012; Kopinke et al., 2017).*

*Hence, incorporating pesticide CSIA data into hydrological models can reduce uncertainties in pesticide dissipation processes by distinguishing between degradative and non-degradative processes, as well as transformation pathways.*

Line 78: some publications have shown that sorption can lead to significant isotope fractionation effects in the spreading direction of pollution which might be relevant in catchment studies as well. Why not test this with this model, as this seems straightforward?

Non-destructive processes, such as dispersion and sorption, can induce significant isotope fractionation, but only under specific conditions, including strong sorption behavior, early transient diffusion, and multi-step sorption in aquifers. In such environments, the slightly stronger sorption of light isotopologues can enhance enrichment in the heavier isotopologues at the leading edge of a sorptive contaminant pulse (early stages) while reducing enrichment at the trailing edge (later stages) (Eckert et al., 2013; van Breukelen and Prommer, 2008; van Breukelen and Rolle, 2012). This effect is particularly relevant when biodegradation is slow.

In our case, S-metolachlor is not expected to exhibit strong sorption behavior, as strongly sorbing contaminants typically have log $K_{ow}$ > 2 and water solubility < 1 ppm, and S-metolachlor degradation in soil was fast. Molecular diffusion, which may lead to non-equilibrium behavior due to concentration gradients between mobile and immobile phases, plays a negligible role in contaminant redistribution when S-metolachlor is first introduced into the field. Furthermore, unlike highly sorptive contaminants in aquifers, S-metolachlor does not form distinct front and tail effects since it is applied uniformly across the catchment and gradually diffuses toward the outlet. Although low but measurable isotope fractionation may occur following multiple sorption–desorption cycles (Kopinke et al., 2017), sorption-induced stable isotope fractionation is generally considered negligible for moderately sorptive pesticides like S-metolachlor in surface agro-ecosystems during seasonal transport.

For these reasons and because sorption-induced isotope fractionation is expected to be very low and negligible in our case, we did not simulate sorption-induced isotope fractionation in our model.

We have moved this section to Materials & Methods and explicitly stated this in the manuscript (lines 268-276), and four additional references have been included:

*"Due to the physicochemical properties of S-metolachlor, abiotic degradation processes such as photolysis, and their associated isotope fractionation, were excluded from the model. While non-destructive processes like dispersion and sorption may induce isotope fractionation under conditions of strong sorption behavior, transient diffusion, or repeated sorption–desorption cycles (Eckert et al., 2013; van Breukelen and Prommer, 2008; van Breukelen and Rolle, 2012; Kopinke et al., 2017), these effects were considered negligible. Given the rapid biodegradation of S-metolachlor in soil and the limited impact of sorption-induced fractionation in surface agro-ecosystems, only biodegradation was included in the model. Since not significant alteration of the isotope composition of pollutants was expected during sorption and ageing*

*(Schmidt et al., 2004; Droz et al., 2021), the ratios of light to heavy isotopologues did not significantly vary during sorption and ageing processes (Eq. 5, first term). »*

2. Lines 77-83: I feel that neglecting isotope fractionation effects of non-transformation processes can be more convincingly elaborated.

We agree with the reviewer.

We have now specified in the manuscript (lines 268-276):

*"Due to the physicochemical properties of S-metolachlor, abiotic degradation processes such as photolysis, and their associated isotope fractionation, were excluded from the model. While non-destructive processes like dispersion and sorption may induce isotope fractionation under conditions of strong sorption behavior, transient diffusion, or repeated sorption–desorption cycles (Eckert et al., 2013; van Breukelen and Prommer, 2008; van Breukelen and Rolle, 2012; Kopinke et al., 2017), these effects were considered negligible. Given the rapid biodegradation of S-metolachlor in soil and the limited impact of sorption-induced fractionation in surface agro-ecosystems, only biodegradation was included in the model. Since not significant alteration of the isotope composition of pollutants was expected during sorption and ageing (Schmidt et al., 2004; Droz et al., 2021), the ratios of light to heavy isotopologues did not significantly vary during sorption and ageing processes (Eq. 5, first term). »*

3. Methods:
   1. Line 177: explain what a mixing topsoil layer is. Ploughed?

As also mentioned by the Reviewer 1, the mixing layer concept is probably the most commonly used in pesticide fate models due to its numerical simplicity and its realistic representation, as hortonian runoff processes have repeatedly been shown to be initiated as a near-surface process (Gao et al., 2004; McGrath et al., 2008; Wallender et al., 2008). This concept assumes that the transport is controlled by a mixing layer below the soil surface, in which rainwater, soil solution and runoff water completely and instantaneously mix (Havis et al., 1992). The soil depth interacting with runoff water is variable ranging from 0.25 to 2 cm according to various experimental and modelling studies (Havis et al., 1992; Wallender et al., 2008). Previous studies have shown that the effective depth of interaction is related to the degree of soil aggregation and increases with soil slope, kinetic energy of raindrops and rainfall intensity (Havis et al., 1992). In fact, the observed mixing-layer depth is often much shallower than the depth required for fitting models to field data (Gao et al., 2004).

The ploughed layer extends from 1 to 30 cm below the surface (second layer in PiBEACH). We have revised the sentence to provide more detailed information about the mixing layer directly in the article.

Gao, B., Walter, M.T., Steenhuis, T.S., Hogarth, W.L., Parlange, J.Y., 2004. Rainfall induced chemical transport from soil to runoff: theory and experiments. Journal of Hydrology. 295, 291-304.

Havis, R.N., Smith, R.E., Adrian D.D., 1992. Partitioning solute transport between infiltration and overland flow under rainfall. Water Resour. Res., 28, 2569-2580.

Wallender, W.W., Joyce, B.A., Ginn, T.R., 2008. Modeling the Transport of Spray-Applied Pesticides from Fields with Vegetative Cover. Transactions of the Asabe. 51, 1963-1976.

Additionally, the reference (McGrath et al., 2008) has been cited at the first mention of the mixing layer to offer a more comprehensive understanding of this concept.

Lines 187-189:

*"Key modifications in PiBEACH include: (i) a topsoil mixing layer, assuming that pesticide transport is controlled by a mixing layer from 0.25 to 2 cm below the soil surface, in which rainfall, soil solution and runoff are assumed to mix instantaneously (McGrath et al., 2008),…"*

2. Some textual errors on lines 168 and 177.

We revised the targeted sentences:

Lines (178-180)

*"At the cell scale, daily vertical water fluxes are computed across soil layers, followed by lateral fluxes upstream to downstream. Surface flow direction is derived from the digital elevation model using flow-accumulation functions, without employing a numerical scheme (Sheikh et al., 2009)."*

*Lines (187-193):*

*"Key modifications in PiBEACH include: (i) a topsoil mixing layer, assuming that pesticide transport is controlled by a mixing layer from 0.25 to 2 cm below the soil surface, in which rainfall, soil solution and runoff are assumed to mix instantaneously (McGrath et al., 2008), and deeper soil layers to capture groundwater contributions to discharge, (ii) daily simulation of topsoil temperature (Neitsch et al., 2011), (iii) daily dynamic topsoil hydraulic properties that reflect crop-specific agronomical practices (Lefrancq et al., 2018), (iv) first-order pesticides degradation and linear sorption/desorption, and (v) pesticide carbon stable isotopic fractionation and transport of bulk heavy and light carbon in the pesticide molecule (e.g., $^{13}C$ and $^{12}C$) (Lutz et al., 2017; Alvarez-Zaldivar et al., 2018)."*

3. Line 181: I think that not the various isotopologues are simulated in the model but the bulk heavy and light carbon in the pesticide?

As suggested by the reviewer, we have revised the sentence, (lines 191-193):

*[…] and (v) pesticide carbon stable isotopic fractionation and transport of bulk heavy and light carbon in the pesticide molecule (e.g., $^{13}C$ and $^{12}C$) (Lutz et al., 2017; Alvarez-Zaldivar et al., 2018)."*

4. Line 187: "numerical diffusion"? Do they authors mean numerical dispersion?

We agree with the reviewer that 'numerical dispersion' refers to a type of truncation error that occurs when solving the diffusion-advection equation using finite differences, and have revised the relevant sentence accordingly (lines 199-203):

*"This is because the numerical dispersion (Gatel et al., 2020) is generally of the same order of magnitude as pesticide export coefficient (i.e., the ratio of the mass transported at the outlet to the applied mass at the catchment scale), i.e., from 1‰ to 1% of the applied amount of pesticides. The numerical dispersion issues can be mitigated, though this leads to longer computation times in both 2D (Lutz et al., 2013) and 3D (Gatel et al., 2020) models."*

5.   Line 188: explain what the export coefficient is.

We provided the definition of the export coefficient (lines 200-201):

*"[…] as pesticide export coefficient (i.e., the ratio of the mass transported at the outlet to the applied mass at the catchment scale), … […]"*

Line 192-193: does this then not prohibit the simulation of hot-moments?

We assumed that the aspect targeted by this comment is the daily time step, which constrains its ability to predict fast flow dynamics, such as runoff generation and preferential water flow. We fully acknowledge that a daily model, using daily rainfall data, cannot capture intra-hourly rainfall intensity, which drives runoff intensity and surface off-site pesticide transport. However, this limitation, which was one of the key motivations for developing PiBEACH in conjunction with an event-based model like OpenLISEM OLP, does not, in our opinion, preclude the simulation of hot moments for degradation. Considering "soil moisture and temperature"- dependent degradation, we can account for periods of slower or faster degradation in the topsoil across the catchment, primarily driven by temperature. For example, during the first application in April, the $DT_{50}$ was close to 100 days, with an average temperature of 9.1 ± 2.9°C (Fig. 3C and Table S2). As topsoil temperatures progressively increased (Fig. 3A), the degradation rate accelerated, reaching a $DT_{50}$ of approximately 25 days in May, four times faster than in April. The impact of topsoil soil moisture on the $DT_{50}$ was secondary (Fig. 3A and C). Between March and July, the topsoil temperature increased, displaying a more pronounced temperature gradient with depth, which resulted in reduced reactivity at greater depths. For example, in July, the $DT_{50}$ values were 25 days in the topsoil (Fig. 3C) and 34 days at a 2-meter depth (data not shown).

In summary, the "soil moisture and temperature" dependence, as a proxy for microbial activity, provided a standardized approach for depicting degradation kinetics across the catchment and throughout the year.

We have now clarified (Lines (437-441):

*"Hydro-climatic variability drove changes in the simulated degradation rates ($k_{Dynamic}$) and associated half-times ($DT_{50,Dynamic}$) of S-metolachlor over time (Fig. 3C). Low S-metolachlor degradation rates during the first 10 days ($DT_{50} = 46 ± 17$ d) coincided with cold (7.1 ± 1.7 °C)*

*and dry period (7.4 mm of rainfall) (Fig. 3A). In contrast, increasing degradation rates ($DT_{50}$ = 22 ± 2 d) between 50 to 120 days following the first application of S-metolachlor were associated with warmer conditions (16.9 ± 3.6 T°C)."*

And in lines 565-573:

*"This study demonstrates that even a moderate sampling effort, including weekly CSIA measurements from composite topsoil samples across the catchment, can effectively identify hot spots and hot moments of pesticide degradation at the catchment scale. In PiBEACH, degradation is primarily driven by soil temperature and moisture, which exhibits clear seasonal and vertical gradients. As a result, degradation hotspots were consistently located in the topsoil. In contrast, soil moisture, although more spatial variable than temperature, played a secondary role. No significant spatial differences in $DT_{50}$ were observed among plots treated with S-metolachlor over the growing season. The limited predicted spatial variation in $DT_{50}$ across the catchment does not diminish the value of PiBEACH's distributed nature, particularly when considering its future integration with distributed event-based models such as OpenLISEM-OLP. This coupling remains essential for simulating surface transport during rainfall events and refining predictions of pesticide fate under dynamic hydrological conditions."*

Model approach general: I do not understand why this model has not been applied to simulate data at the catchment outlet as it seems to be designed to this end. Could not a simpler model been applied to simulate only the data in the topsoil?

We thank the reviewer for this comment. We agree that a lumped and parsimonious modelling approach, such as that developed by Lutz et al. (2017), effectively captures the dynamics of pesticide degradation and transport at the catchment scale. However, this approach was not designed to provide a distributed representation of pesticide concentrations in the topsoil throughout the growing season to assess off-site pesticide transport resulting from runoff and erosion. Such information is critical for informing management decisions and conducting risk assessments. PiBEACH was initially developed for this purpose as a companion model for distributed, event-based pesticide transport models, such as OpenLISEM OLP (Commelin et al., 2024), and cannot be used independently to simulate data at the catchment outlet. We agree that a lumped and parcimonious modelling approach such a as developed by Lutz et al., (2017) was able to capture the dynamic of pesticide degradation and transport at the catchment scale. Therefore, this lumped approach was not designed to provide distributed extent of pesticide amount in topsoil when a rainfall event causes runoff and erosion with associated pesticide transport to support management decisions and risk assessments. PiBEACH was initially developed for this purpose as a companion model for distributed event-based pesticide transport models, such as OpenLISEM OLP (Commelin et al., 2024) and cannot be directly applied to simulate data at the catchment outlet. PiBEACH was not originally designed to simulate discharge and the associated pesticide concentrations in headwater catchments, where surface runoff and off-site transport can play a significant role in pesticide fluxes in rivers, such as in the Alteckendorf catchment with loamy soils (Lefrancq et al., 2018). Additionally, PiBEACH does not account for erosion and the transport of pesticides sorbed in the suspended matter, which is another limitation for

simulating pesticide concentrations in rivers. The particulate fraction of some pesticides can be substantial at the outlet following intense rainfall events (Lefrancq et al., 2018).

As discussed previously, PiBEACH has a dual objective:

- To simulate both degradation and non-degradative processes across the catchment and throughout the crop growing season, providing key inputs for distributed, event-based models such as OpenLISEM OLP, which simulates the transport of both dissolved and particulate pesticides, based on the remaining pesticides in the mixing layer. This contributes significantly to pesticide modelling at the catchment scale.
- To model based-flow and the associated pesticide concentrations, enabling the integration of discharge and pesticide concentrations between rainfall events.

We thank the reviewer for highlighting the importance of considering the complexity or simplicity of a model in providing targeted answers. We have not verified in the section on model description in the Materials and Methods and this issue has been addressed: "the development of the PiBEACH conceptual model relied on field knowledge of hydrological dynamics and associated S-metolachlor flows in the Alteckendorf catchment, translating our experimentalist's understanding (perceptual model) into conceptual model", with an appropriate level of complexity to meet the two objectives.

To clarify the application domain of PiBEACH and prevent any misinterpretation of its capabilities and uses, we have now provided additional details in the title:

*"Constraining topsoil pesticide degradation in a conceptual distributed catchment model with compound-specific isotope analysis (CSIA)"*

In the Abstract (Lines 15-18):

*"A new conceptual distributed hydrological model, PIBEACH, was developed to simulate daily pesticide dissipation in soils and its transport to surface waters. The model integrates changes in the carbon isotopic signatures ($\delta^{13}C$) of S-metolachlor during degradation to constrain key parameters and reduce equifinality."*

and lines 23-25:

*" In addition, PiBEACH, which accounts for spatial and seasonal variations in topsoil pesticide concentrations, enables coupling with distributed, event-based hydrological models such as OpenLISEM OLP to capture intra-event pesticide transport dynamics more accurately."*

As detailed below, this information is also provided in the Materials and Methods section (see the new section on metric calculation, "2.7 Metrics of Pesticide Persistence and Transport Risks," Lines 416-429, page 15 of this document) and in the Results and Discussion section (Lines 544-564):

*"This modelling framework, applied to the same compound and catchment as in Lutz et al. (2017), provides insights into the benefits and limitations of using $\delta^{13}C$ data from soil and river*

*samples to constrain reactive transport models. The lumped model by Lutz et al. (2017) focuses on $\delta^{13}C$ values at the catchment outlet, providing an integrated signature of degradation processes across the catchment. In contrast, PiBEACH allowed evaluation of the added value of topsoil $\delta^{13}C$ data for constraining degradation kinetics. Model predictions for three components of the S-metolachlor mass balance, degraded mass, remaining mass in topsoil, and export at the outlet, were validated against observations. Due to limitations in representing rapid surface off-site transport associated with runoff and erosion, PiBEACH did not incorporate $\delta^{13}C$ data from river into the calibration. However, future coupling of PiBEACH with an event-based model like OpenLISEM-OLP could enable simultaneous integration of both topsoil and river $\delta^{13}C$ datasets, improving constraints on degradation and enhancing simulation of fast-flow transport during high-intensity rainfall-runoff events.*

*In summary, $\delta^{13}C$ data from both soil and river samples can enhance model calibration by constraining degradation processes along different transport pathways. Lumped models (e.g., Lutz et al., 2017) and distributed models like PiBEACH each offer complementary perspectives on pesticide dynamics. However, only a distributed model can account for spatial heterogeneity across the catchment and support management decisions. PiBEACH also provides spatially explicit outputs on pesticide concentrations, hydraulic properties, and topsoil moisture as critical inputs for initializing distributed event-based models. A large portion of the outlet data, i.e. discharge for calibration (as part of the KGE objective function), and discharge and S-metolachlor concentrations for evaluating export, was used to assess model performance. Nevertheless, due to limitations of PiBEACH's conceptual structure and daily resolution, the comparison between observed and simulated reactive transport was limited, unlike previous lumped approaches (Lutz et al., 2017). The forthcoming PiBEACH–OpenLISEM-OLP modelling effort will explore the added value of incorporating observed S-metolachlor concentrations and isotopic signatures at the outlet into the calibration process."*

Lines 205-206: what is meant with the depth of the groundwater layers varying constantly from upstream to downstream?

As described, the Alteckendorf catchment was not equipped with piezometers or wells to monitor the dynamics of the water table. Therefore, we adopted a conceptual approach based on our understanding of soil structure (Lefrancq et al., 2017, 2018), capable of simulating the observed seasonal fluctuations in the water table when monitoring data is available (Molenat et al., 2005). A two-layer shallow aquifer ($z_3$ and $z_4$) was defined, with its thickness decreasing from the upper hillslopes (maximum of 23.2 m) to the riverbanks. The shallow aquifer is encased by unsaturated layers ($Z_0$ to $Z_2$, with a total thickness of 80 cm), which include a mixing layer, the plough layer, and a layer affected by artificial drainage, as determined by our soil structure investigation (Lefrancq et al., 2017, 2018). The shallow aquifer was divided into an upper variably saturated layer ($z_3$), influenced by aquifer recharge, and a lower saturated layer that drains towards the river, functioning as a global linear reservoir (Manfreda et al., 2005). A calibration parameter ($Z_f$) was used to distinguish the upper and lower portions of the shallow aquifer, accounting for the gradual decrease in thickness of both the saturated and variably saturated layers. Consequently, we revised the targeted sentence by replacing 'constantly' with 'gradually' to more accurately reflect our approach.

Lines 218-220:

*"The combined groundwater depths ($z_3$ + $z_4$) varied spatially and seasonally across the catchment, with a maximum observed depth of 23.2 m, consistent with recharge dynamics in monitored hillslopes (Molenat et al., 2005)."*

With a new reference:

Molénat, J., Gascuel-Odoux, C., Davy, P., Durand P.: How to model shallow water-table depth variations: the case of the Kervidy-Naizin catchment, France, Hydrol. Process., 19, 901-920, https://doi.org/10.1002/hyp.5546, 2005.

6. Page 9: as sorption was simulated in the model, why not test / show that sorption isotope fractionation has limited/negligible effects on the simulated isotope ratios in topsoil (and especially at the outlet?)?

As discussed previously in the point 2 raised by the Reviewer, S-metolachlor is not expected to exhibit strong sorption behavior, as strongly sorbing contaminants typically have log $K_{ow}$ > 2 and water solubility < 1 ppm, and S-metolachlor degradation in soil was fast. Molecular diffusion plays a negligible role in contaminant redistribution when S-metolachlor is first introduced into the field, potentially leading to non-equilibrium behavior due to concentration gradients between mobile and immobile phases. Furthermore, unlike highly sorptive contaminants in aquifers, S-metolachlor does not form distinct front and tail effects since it is applied uniformly to corn and sugar beet plots across the catchment and gradually diffuses toward the outlet. Although low but measurable isotope fractionation may occur following multiple sorption–desorption cycles (Kopinke et al., 2017), sorption-induced stable isotope fractionation is generally considered negligible for moderately sorptive pesticides like S-metolachlor in surface agro-ecosystems during seasonal transport. Thus, we did not simulate sorption-induced isotope fractionation in our model.

Lines 268-273:

*"Due to the physicochemical properties of S-metolachlor, abiotic degradation processes such as photolysis, and their associated isotope fractionation, were excluded from the model. While non-destructive processes like dispersion and sorption may induce isotope fractionation under conditions of strong sorption behavior, transient diffusion, or repeated sorption–desorption cycles (Eckert et al., 2013; van Breukelen and Prommer, 2008; van Breukelen and Rolle, 2012; Kopinke et al., 2017), these effects were considered negligible. Given the rapid biodegradation of S-metolachlor in soil and the limited impact of sorption-induced fractionation in surface agro-ecosystems, only biodegradation was included in the model. Since not significant alteration of the isotope composition of pollutants was expected during sorption and ageing (Schmidt et al., 2004; Droz et al., 2021), the ratios of light to heavy isotopologues did not significantly vary during sorption and ageing processes (Eq. 5, first term)."*

Page 10: add bit of explanation to argue why it was chosen for to simulate only the bioavailable fraction? How would it influence model outcomes compared to the more common model formulation that simulates biodegradation of the

total dissolved fraction? Is this in line with earlier catchment degradation model studies?

As noted by the reviewer regarding common model formulations, we also accounted for the biodegradation of the total dissolved fraction. Additionally, we assumed that the reversibly sorbed fraction of pesticides, subject to daily sorption/desorption equilibrium, undergoes biodegradation at a similar rate, expressed by the $DT_{50, reference}$ value. However, as in several pesticide transport models, e.g. PEARL, PELMO, PRZM or MACRO, we included an aged fraction that is more strongly sorbed and undergoes limited degradation.

In summary, consistent with the following models, we considered three pesticide pools: the bioavailable fraction, comprising both the (1) total dissolved fraction and (2) the reversibly sorbed fraction, and (3) a non-extractable fraction.

We have revised the following sentence to clarify the biodegradation (265-268):

*"Biodegradation occurs only in bioavailable fractions of adsorbed (ads) and aqueous (aq) phases (Thullner et al., 2013). A similar stable isotope fractionation associated with degradation was then considered for the bioavailable fractions, including the adsorbed (Eq. 5, second term) and the aqueous (Eq. 6) phases. The bioavailable fraction was controlled kinetically by an ageing rate $k_{age}$ ($d^{-1}$) on the adsorbed fraction (Schwarzenbach, 2003)."*

7. Line 265: "depth-dependent degradation". The argument not to include this in the model seems rather weak (because there are no data available from deeper soil layers; but there are in fact catchment outlet data available). If that reasoning is followed the authors should also only draw conclusions on what happens in the top soil layer and not in the catchment as a whole. Lutz et al. 2017 included "depth-dependent degradation" to enable fitting the concentration and isotope ratios of pesticides at the catchment outlet. But the current model is not constrained with such outlet data and therefore there will be high uncertainty on what happens at the catchment scale. In fact as the model is only calibrated with topsoil data (and discharge at the outlet but that is not that relevant for catchment scale degradation extent), the model is therefore only informative on what happens in the upper 1 cm of the catchment. See also main comment #2.

We fully agree with the Reviewer that the apparent depth-dependent degradation may be a parsimonious approach to represent complexity of depth-dependent microorganism activity as shown in different recent papers. We have chosen in this paper to explicitly link the persistence of pesticide with soil moisture and temperature as key drivers of microorganism activity integrating both temporal, spatial and depth dynamics such as in MACRO or PEARL models. We tested at the lab scale on the Alteckendorf soil, the predictability of this approach (Figure S4) before to implement it in PiBEACH. With this "soil moisture and temperature"-dependent degradation, we can account for periods of slower or faster degradation in the topsoil across the catchment. Figure 3C highlights cold and hot moments, i.e., slower and then faster degradation, primarily driven by temperature. The $DT_{50}$ was close to 100 days during the first application in April at 9.1 ± 2.9°C, before decreasing to 25 days in May. As topsoil temperatures progressively increased (Fig. 3A), the degradation rate stabilized in May, with a

$DT_{50}$ of approximately 25 days—four times faster than in April. The impact of topsoil soil moisture on the predicted $DT_{50}$ was secondary (Fig. 3A and C). Between March and July, the topsoil temperature increased, displaying a more pronounced temperature gradient with depth, which resulted in reduced reactivity at greater depths. For example, in July, the $DT_{50}$ values were 25 days in the topsoil (Fig. 3C) and 34 days at a 2-meter depth (data not shown). The adopted "soil moisture and temperature" dependence as a proxy for microorganism activity allow an uniformized method to depict degradation kinetic across the catchment and throughout the year.

This decrease resulted in a moderate, depth-dependent degradation of PiBEACH, in line with predictions from the calibration of the lumped model developed by Lutz et al. (2017) for this catchment. In both cases, i.e., the depth-dependent degradation (Lutz et al., 2017) and soil moisture and temperature-dependent degradation, the adopted approach can only tackle tendencies of microbial activity which remains largely unknown and complex to monitor along soil depth and at the catchment scale.

The next step, opened with the PiBEACH development, will be the combination with an event-based model such as OpenLISEM OLP which is expected to significantly improve the intra-daily runoff and associated dissolved and particulate transport from the plots to the outlet. This combined approach opens the way to open the grey box of soil reactivity by introducing new dependence and calibrate associated parameters.

We have now clarified the assumptions of soil moisture and temperature dependence in the Materials and Methods section (lines 293-296 / 317-322) and completed the discussion (lines 567-572), as suggested by the Reviewer.

Lines 293-296:

*"Pesticide degradation rates generally decline with soil depth due to reduced microbial activity (Cruz et al., 2008; Lutz et al., 2017) and increased sorption (Arias-Estevez et al., 2008) in deeper layer. However, the absence of concentration and CSIA data for S-metolachlor in subsoil prevented direct quantification of depth-dependence degradation. Following the approach used in advanced pesticide fate models such as MACRO (Garratt et al., 2003), the degradation rate ($k_{deg}$) in PiBEACH was modelled as a function of soil hydro-climatic conditions, such as soil temperature and water content. A dynamic degradation rate ($k_{Dynamic}$, $d^{-1}$) was calculated daily on soil temperature ($F_T$) and water content ($F_\theta$):"*

*Lines 317-322):*

*"Between March and July, topsoil temperatures increased, creating a stronger vertical temperature gradient and leading to reduced degradation activity at greater depths. In July, for instance, $DT_{50}$ values were estimated at 25 days in the topsoil (Fig. 3C) and 34 days at 2 m depth (data not shown), consistent with predictions from the lumped model developed by Lutz et al. (2017) for this catchment. While the model accounts for degradation as a function of depth, soil temperature, and moisture (Lutz et al., 2017), it captures only general trends. Microbial activity remains highly uncertain and difficult to quantify across soil depth and spatial scales."*

*"In PiBEACH, degradation is primarily driven by soil temperature and moisture, which exhibits clear seasonal and vertical gradients. As a result, degradation hotspots were consistently located in the topsoil. In contrast, soil moisture, although more spatial variable than temperature, played a secondary role. No significant spatial differences in $DT_{50}$ were observed among plots treated with S-metolachlor over the growing season. The limited predicted spatial variation in $DT_{50}$ across the catchment does not diminish the value of PiBEACH's distributed nature, particularly when considering its future integration with distributed event-based models such as OpenLISEM-OLP."*

8. The three applications as shown in Fig. 3 are not described in the methods section (or I could not find it): how much pesticide was applied when and where? Exactly this heterogeneity in source zone variation calls for a model like this one but the advantage of using a distributed model has not been illustrated.

The application details were provided in the initial version, in the Supplement (Table S2).

Table S2. Applied mass (Kg) of active ingredient (S-metolachlor) per transect by date and days since 1st application. Ranges indicates uncertainty of exact application date (Alvarez-Zaldivar et al., 2018).

| App. No. | Date | Days | North | Valley | South |
|---|---|---|---|---|---|
| A1 | March 20 - 25th | 0 - 5 | 5.1 | 1.6 | 11.1 |
| A2 | April 13 - 14th | 25 - 26 | 8.0 | 1.8 | 2.9 |
| A3 | May 25 - 31st | 67 – 73 | 7.2 | 2.4 | 0.0 |
| Total | | | 20.2 | 5.9 | 14.0 |

We have now revised the sentence in the Materials and Methods to summarize Table S2, emphasizing the spatial and temporal variability of S-metolachlor applications at the plot scale. In our view, this variability highlights the necessity of a distributed modelling approach such as PiBEACH.

Lines 136-138:

*"Application dates, doses and formulation were obtained for each plot from farmer surveys, providing spatially distributed data for modelling. In total, 20.2 kg, 5.9 kg, and 14.0 kg of S-metolachlor were applied to the North, Valley, and South transects, respectively (Table S2)."*

Results & Discussion:

9. Generally: explain why concentrations and isotope ratios of pesticide at the catchment outlet were not used. I assumed this was done when starting reading the paper but only in the R&D section is became clear that only data from the upper 1cm of the catchment skin were measured and simulated. How then can conclusions be drawn on overall degradation at the catchment body when these simulations are not constrained with outlet data? Thus relatedly,

what is the relevance of the simulations of the upper 1 cm in the soil as presented in the key result Figure 3 on what happens at catchment scale? See also main comment #2.

We acknowledge Reviewer 2's comments regarding the relevance of the catchment results (Figure 5). As suggested by Reviewer 1, we have now reorganized the article's structure by adding a new section to the Materials and Methods, which introduces the six metrics simulated after calibration and derived from soil and river samples. Additionally, we have provided a detailed explanation of how the observed values, i.e., topsoil S-metolachlor concentration and $\delta^{13}$C, but also river discharge, and S-metolachlor concentration at the outlet, were used to derive the observed values for three of the six metrics.

The model was calibrated using topsoil concentration (KGESM > 0.5) and isotopic signature (KGEδ > 0.8), along with the flow rate at the outlet (KGEQ > 0.5), to represent hydrological functioning at the catchment scale. As mentioned in the Results section (lines 438–440)

*"First, PiBEACH operates on a daily time step and is not suited to capture rapid flow dynamics, such as runoff generation and event-scale hydrological dynamics (Sheikh et al., 2009). Consequently, S-metolachlor concentrations and corresponding $\delta^{13}$C values measured at the catchment outlet were not included in the model calibration objective function. This limitation could be addressed by coupling PiBEACH with a distributed event-based model, such as the Limburg Soil Erosion Model (OpenLISEM) (Baartman et al., 2012), which recently integrated a pesticide module (OLP) (Commelin et al., 2024)."*

This result (Figure 3B) confirms the limitations of PiBEACH in accurately representing intra-daily and intra-hourly runoff generation and the associated fast-flow pesticide transport.

The S-metolachlor concentrations measured at the outlet throughout the growing season were also used to derive the observed export metric (Figure 5C). As expected, given PiBEACH's limited ability to capture intra-daily runoff generation, it is unable to accurately simulate S-metolachlor fluxes at the catchment scale representing less than 2% of the applied amount. Therefore, we opted not to include outlet concentration and isotopic signature in the calibration function to avoid a biased calibration process, which could lead to parameter fitting for the wrong reasons in an attempt to compensate for the model's conceptual limitations.

The added value of incorporating observed S-metolachlor concentrations and isotopic signatures at the outlet in the calibration process will be explored in the upcoming PiBEACH-OpenLISEM OLP modelling study.

It is important to emphasize that the primary goal of PiBEACH, which has been now clarified in the manuscript, is to simulate the degradation and persistence of pesticides in topsoil layer for coupling with event-based models such as OpenLISEM OLP. Figures 5A (isotopic dynamics) and 5B (remaining mass in topsoil) demonstrate that PiBEACH performs well in achieving this objective.

In the abstract (Lines 22-25):

*"Overall, integrating CSIA data into PiBEACH model significantly enhances the reliability of pesticide degradation predictions at the catchment scale. In addition, PiBEACH, which accounts for spatial and seasonal variations in topsoil pesticide concentrations, enables coupling with distributed, event-based hydrological models such as OpenLISEM OLP to capture intra-event pesticide transport dynamics more accurately."*

*In the discussion (Lines 549-553):*

*"Due to limitations in representing rapid surface off-site transport associated with runoff and erosion, PiBEACH did not incorporate $\delta^{13}C$ data from river into the calibration. However, future coupling of PiBEACH with an event-based model like OpenLISEM-OLP could enable simultaneous integration of both topsoil and river $\delta^{13}C$ datasets, improving constraints on degradation and enhancing simulation of fast-flow transport during high-intensity rainfall-runoff events."*

*And in lines 562-564:*

*"The forthcoming PiBEACH–OpenLISEM-OLP modelling effort will explore the added value of incorporating observed S-metolachlor concentrations and isotopic signatures at the outlet into the calibration process."*

> 10. The three levels of sampling resolution as presented in Figure 4: composite, transects, plots are unclear, and not clearly described and explained in the method section. This would help also on improving on main comment #1.

Following the suggestion of both reviewers, we have now revised the caption of the Figure 1, and the following sentence, to explain the weekly weighted samples of topsoil for each transect from regular sampling (green dots) along transects (see Alvarez-Zaldivar et al., 2018).

Lines 120-122:

*"Figure 1: The Alteckendorf headwater catchment (Bas-Rhin, France), showing the experimental setup, including transects (weighted samples collected at green dots along red lines) and plot sampling (black dots). Land use for 2016 is also displayed. The "Other" category includes roads, grass strips and orchards."*

*And in lines 142-145:*

*"Thirteen marked plots were sampled before S-metolachlor application and on days 1, 50, and 100 after application. In addition, weekly weighted composite samples were taken along the north, valley, and south transects (green dots in Fig. 1) from March 19 (day 0) to July 12 (day 115). All soil samples were analysed for S-metolachlor concentration and carbon isotope composition ($\delta13C$) using CSIA"*

11. Lines 440-441: in my words: "more detailed spatial soil sampling did not help to better constraint the model". Why then was a spatially distributed model needed?

Mobilizing data of isotopic signatures through higher-resolution soil sampling (plots > transects > transect composite samples) did not significantly improve model calibration in this catchment. Therefore, pooling weekly transect samples into a single composite soil sample for CSIA appears sufficient to constrain the degradation parameter, i.e. $DT_{50, \text{reference}}$, while also reducing sampling and analytical efforts. This composite approach may effectively capture temporal variations in pesticide degradation at the catchment scale.

We hypothesized that the transect composite approach, with a single averaged topsoil $\delta^{13}C$ value per week, could also enhance the calibration of lumped models such as the one developed by Lutz et al. (2017). However, this result does not disqualify distributed modelling approaches such as PiBEACH, which was primarily designed to spatially represent pesticide concentrations, hydraulic properties, and topsoil moisture as initial conditions for distributed event-based models.

We have underlined this point in the conclusion (lines 582-885):

*"Incorporating more detailed physical and biological representations of degradation catchments increases model complexity and uncertainty, making it essential to acquire additional data to constrain parameter ranges and reduce equifinality."*

And 486-490:

*"These results indicated that increasing resolution (plots > transects > transect composite samples) did not significantly improve model calibration in this catchment. Therefore, pooling transect samples into a single composite soil sample was sufficient for CSIA-based calibration, while also reducing sampling and analytical efforts. This approach may reliably capture variations in pesticide degradation at the catchment scale."*

12. Lines 443-447: the difference in fractionation factor -2.7 vs. -1.5 is quite large and needs further explanation. Also compare and discuss the outcome with the fractionation factor calibrated by Lutz et al (2017): - 1.3 permil.

The incorporation of CSIA data into PiBEACH yielded an isotope fractionation factor of $\varepsilon C = -2.7 \pm 0.6$ ‰ for S-metolachlor degradation in topsoil. This value is notably larger than those reported in previous studies, including Alvarez-Zaldivar et al. (2018) ($\varepsilon C = -1.5 \pm 0.5$ ‰) and Lutz et al. (2017) ($\varepsilon C = -1.3 \pm 0.5$ ‰). While this discrepancy suggested differences in degradation dynamics or environmental conditions, it remained within the uncertainty range obtained from the model ensemble in this study.

The observed stronger fractionation effect ($\varepsilon C = -2.7 \pm 0.6$ ‰) may result from variability in environmental conditions, microbial activity, and transformation kinetics, supporting a more refined assessment of S-metolachlor degradation. The fractionation factor reported by Alvarez-Zaldivar et al. (2018) ($\varepsilon C = -1.5$ ‰) was obtained under laboratory conditions, where experimental constraints may have reduced isotope fractionation compared to field-scale

observations. Similarly, the εC = −1.3 ± 0.5 ‰ reported by Lutz et al. (2017) was derived from a different dataset and modelling approach.

The present study incorporates a more extensive dataset and employs model ensembles, capturing a broader range of degradation scenarios and yielding a stronger fractionation effect. This suggests that enzymatic degradation may have been more pronounced, potentially due to higher microbial activity or the presence of a dominant microbial consortia with different metabolic capabilities.

While the previously reported εC values (−1.5 ‰ and −1.3 ± 0.5 ‰) may have slightly overestimated degradation extent, the discrepancy remains within the uncertainty bounds determined in this study. These findings emphasize the need for site-specific calibration when applying CSIA-based degradation assessments and highlight the importance of model ensemble approaches in accounting for uncertainties in isotope fractionation factors under field conditions.

The difference in isotope fractionation in previous and this study is now discussed in the manuscript (lines 492-501):

*"Integration of CSIA data into PiBEACH yielded an isotope fractionation factor of εC = −2.7 ± 0.6 ‰ for S-metolachlor degradation in topsoil, exceeding previously reported values (Alvarez-Zaldivar et al., 2018: −1.5 ± 0.5 ‰; Lutz et al., 2017: −1.3 ± 0.5 ‰). This discrepancy likely reflects differences in environmental conditions, microbial activity, and transformation pathways, though it remains within the model ensemble's uncertainty range. Compared to laboratory-derived estimates, the stronger fractionation observed suggests either alternative degradation pathways or the influence of distinct microbial consortia at the catchment scale, relative to laboratory conditions. These findings underscore the importance of site-specific calibration in CSIA applications and highlight the value of model ensemble approaches in capturing the range of degradation processes in heterogeneous agro-ecosystems. While previously reported εC values may slightly overestimate degradation in some field settings, ensemble modelling with integrated CSIA data provides a more robust, field-relevant assessment of pesticide transformation."*

13. Line 458+459: "measure", change into "measurements"

We have corrected the sentence as suggested.

14. Figure 5: what is the relevance if these results at catchment scale when the model is only calibrated on topsoil data but not on outlet pesticide data?

We acknowledge the Reviewer's comments regarding the relevance of the catchment results (Figure 5). As mentioned in the previous point n°12, this has now been clarified by specifying how the soil and river samples were used in a new section to the Materials and Methods. As discussed previously, it should be mentioned that the primary goal of PiBEACH, is to simulate the degradation and persistence of pesticides in topsoil layer for coupling with event-based models such as OpenLISEM OLP. This has now been clarified in the manuscript. Figures 5A

(isotopic dynamics) and 5B (remaining mass in topsoil) demonstrate that PiBEACH performs well in achieving this objective.

We have integrated these elements in the discussion section (lines 544-564):

*"This modelling framework, applied to the same compound and catchment as in Lutz et al. (2017), provides insights into the benefits and limitations of using $\delta^{13}C$ data from soil and river samples to constrain reactive transport models. The lumped model by Lutz et al. (2017) focuses on $\delta^{13}C$ values at the catchment outlet, providing an integrated signature of degradation processes across the catchment. In contrast, PiBEACH allowed evaluation of the added value of topsoil $\delta^{13}C$ data for constraining degradation kinetics. Model predictions for three components of the S-metolachlor mass balance, degraded mass, remaining mass in topsoil, and export at the outlet, were validated against observations. Due to limitations in representing rapid surface off-site transport associated with runoff and erosion, PiBEACH did not incorporate $\delta^{13}C$ data from river into the calibration. However, future coupling of PiBEACH with an event-based model like OpenLISEM-OLP could enable simultaneous integration of both topsoil and river $\delta^{13}C$ datasets, improving constraints on degradation and enhancing simulation of fast-flow transport during high-intensity rainfall-runoff events.*

*In summary, $\delta^{13}C$ data from both soil and river samples can enhance model calibration by constraining degradation processes along different transport pathways. Lumped models (e.g., Lutz et al., 2017) and distributed models like PiBEACH each offer complementary perspectives on pesticide dynamics. However, only a distributed model can account for spatial heterogeneity across the catchment and support management decisions. PiBEACH also provides spatially explicit outputs on pesticide concentrations, hydraulic properties, and topsoil moisture as critical inputs for initializing distributed event-based models. A large portion of the outlet data, i.e. discharge for calibration (as part of the KGE objective function), and discharge and S-metolachlor concentrations for evaluating export, was used to assess model performance. Nevertheless, due to limitations of PiBEACH's conceptual structure and daily resolution, the comparison between observed and simulated reactive transport was limited, unlike previous lumped approaches (Lutz et al., 2017). The forthcoming PiBEACH–OpenLISEM-OLP modelling effort will explore the added value of incorporating observed S-metolachlor concentrations and isotopic signatures at the outlet into the calibration process."*

4. Discussion is largely lacking. Add a larger discussion section to include for example:

We acknowledge the Reviewer's feedback and we have clarified the discussion by incorporating elements related to the three highlighted points and previously discussed.

1. Comparison with findings Lutz et al. (2017) who applied a similar model to the same catchment but then calibrated to earlier data at the catchment outlet instead of the topsoil. Add some discussion to discuss why this study did find that having isotope data let to better model results, whereas this effect was not clearly found by Lutz et al. (2017). Note that Lutz et al (2027) had to include depth-dependent degradation to enable to simulate the outlet pesticide and isotope ratio data.

We have now added the following sections, in the Introduction (lines 88-9, in Material and Methods (lines 317-322), discussing the main contributions of each modelling study (Lutz et al., 2017 and this one), in the result and discussion section (line 544-564), before the conclusion:

In the introduction (lines 88-94):

*"More recently, CSIA data in a parsimonious have been integrated into a lumped transport model using travel-time distributions, improving the interpretation of pesticide transport at the catchment scale (Lutz et al., 2017). However, lumped models primarily capture aggregate hydrological behaviour and do not account for spatial variability in parameters such as soil moisture or temperature (Fatichi et al., 2016). This limits their ability to represent landscape heterogeneity, including variations in crops and pesticide applications, thereby hindering the identification of contaminant sources and degradation hot spots (Grundmann et al., 2007). In contrast, distributed conceptual and physically-based models explicitly represent spatially distributed hydro-climatic dynamics regulating hydrological processes, such as runoff and infiltration."*

*And in lines 108-109:*

*"This adaptation led to the development of PiBEACH (Pesticide isotope BEACH), extending the lumped isotope modeling approach of Lutz et al. (2017) into a spatially distributed framework."*

In Material and Methods (Lines 317-322):

*"Between March and July, topsoil temperatures increased, creating a stronger vertical temperature gradient and leading to reduced degradation activity at greater depths. In July, for instance, DT50 values were estimated at 25 days in the topsoil (Fig. 3C) and 34 days at 2 m depth (data not shown), consistent with predictions from the lumped model developed by Lutz et al. (2017) for this catchment. While the model accounts for degradation as a function of depth, soil temperature, and moisture (Lutz et al., 2017), it captures only general trends. Microbial activity remains highly uncertain and difficult to quantify across soil depth and spatial scales."*

In the result and discussion section (line 544-564):

*"This modelling framework, applied to the same compound and catchment as in Lutz et al. (2017), provides insights into the benefits and limitations of using $\delta^{13}C$ data from soil and river samples to constrain reactive transport models. The lumped model by Lutz et al. (2017) focuses on $\delta^{13}C$ values at the catchment outlet, providing an integrated signature of degradation processes across the catchment. In contrast, PiBEACH allowed evaluation of the added value of topsoil $\delta^{13}C$ data for constraining degradation kinetics. Model predictions for three components of the S-metolachlor mass balance, degraded mass, remaining mass in topsoil, and export at the outlet, were validated against observations. Due to limitations in representing rapid surface off-site transport associated with runoff and erosion, PiBEACH did not incorporate $\delta^{13}C$ data from river into the calibration. However, future coupling of PiBEACH with an event-based model like OpenLISEM-OLP could enable simultaneous integration of both*

*topsoil and river δ¹³C datasets, improving constraints on degradation and enhancing simulation of fast-flow transport during high-intensity rainfall-runoff events.*

*In summary, δ¹³C data from both soil and river samples can enhance model calibration by constraining degradation processes along different transport pathways. Lumped models (e.g., Lutz et al., 2017) and distributed models like PiBEACH each offer complementary perspectives on pesticide dynamics. However, only a distributed model can account for spatial heterogeneity across the catchment and support management decisions. PiBEACH also provides spatially explicit outputs on pesticide concentrations, hydraulic properties, and topsoil moisture as critical inputs for initializing distributed event-based models. A large portion of the outlet data, i.e. discharge for calibration (as part of the KGE objective function), and discharge and S-metolachlor concentrations for evaluating export, was used to assess model performance. Nevertheless, due to limitations of PiBEACH's conceptual structure and daily resolution, the comparison between observed and simulated reactive transport was limited, unlike previous lumped approaches (Lutz et al., 2017). The forthcoming PiBEACH–OpenLISEM-OLP modelling effort will explore the added value of incorporating observed S-metolachlor concentrations and isotopic signatures at the outlet into the calibration process."*

In the conclusion section (lines 579-588):

*"This study addresses the gap between the increasing complexity of reactive transport models and the limited availability of field data for their calibration and validation. Although the representation of interactions between hydro-climatic conditions, biogeochemical processes, and degradation dynamics has improved significantly in recent decades, it remains oversimplified in many 2D catchment models (Lutz et al., 2013) or is not explicitly defined (DeMars et al., 2018). Incorporating more detailed physical and biological representations of degradation catchments increases model complexity and uncertainty, making it essential to acquire additional data to constrain parameter ranges and reduce equifinality. Our findings suggest that using only topsoil pesticide concentrations and discharge measurements is insufficient to effectively constrain reactive transport models at the catchment scale. We demonstrated that incorporating complementary data, such as pesticide CSIA data (Fenner et al., 2013, Elsner and Imfeld, 2016; Hofstetter et al., 2024), can help differentiate between degradative and non-degradative processes contributing to observed concentration declines."*

2. Pros/cons topsoil vs. catchment outlet data.

We have now added the following section, discussing the strengths and weaknesses of using soil and river δ¹³C data to constrain reactive transport models, in the results section (line 594), before the conclusion:

In the result and discussion section (line 544-564):

*"This modelling framework, applied to the same compound and catchment as in Lutz et al. (2017), provides insights into the benefits and limitations of using δ¹³C data from soil and river samples to constrain reactive transport models. The lumped model by Lutz et al. (2017) focuses on δ¹³C values at the catchment outlet, providing an integrated signature of degradation*

*processes across the catchment. In contrast, PiBEACH allowed evaluation of the added value of topsoil $\delta^{13}C$ data for constraining degradation kinetics. Model predictions for three components of the S-metolachlor mass balance, degraded mass, remaining mass in topsoil, and export at the outlet, were validated against observations. Due to limitations in representing rapid surface off-site transport associated with runoff and erosion, PiBEACH did not incorporate $\delta^{13}C$ data from river into the calibration. However, future coupling of PiBEACH with an event-based model like OpenLISEM-OLP could enable simultaneous integration of both topsoil and river $\delta^{13}C$ datasets, improving constraints on degradation and enhancing simulation of fast-flow transport during high-intensity rainfall-runoff events.*

*In summary, $\delta^{13}C$ data from both soil and river samples can enhance model calibration by constraining degradation processes along different transport pathways. Lumped models (e.g., Lutz et al., 2017) and distributed models like PiBEACH each offer complementary perspectives on pesticide dynamics. However, only a distributed model can account for spatial heterogeneity across the catchment and support management decisions. PiBEACH also provides spatially explicit outputs on pesticide concentrations, hydraulic properties, and topsoil moisture as critical inputs for initializing distributed event-based models. A large portion of the outlet data, i.e. discharge for calibration (as part of the KGE objective function), and discharge and S-metolachlor concentrations for evaluating export, was used to assess model performance. Nevertheless, due to limitations of PiBEACH's conceptual structure and daily resolution, the comparison between observed and simulated reactive transport was limited, unlike previous lumped approaches (Lutz et al., 2017). The forthcoming PiBEACH–OpenLISEM-OLP modelling effort will explore the added value of incorporating observed S-metolachlor concentrations and isotopic signatures at the outlet into the calibration process."*

3. Line 493: I thought the conclusion was that there are no "hot-spots" in this catchment, as there is limited spatial variation among the topsoil samples? Therefore, add some discussion what the added benefit of a spatial distributed model was in this case.

5. Hot-spot and hot-moments. The paper does not discuss hot-spots in the catchment. The advantage of this model is that it enables to account for spatial heterogeneity but it seems that all model parameters were taken spatially homogeneously except with depth. Therefore, the added benefit to the model applied by Lutz et al. 2017 remains unexplored.

We have now added the following section discussing the points 3 and 5 as suggested by the reviewer 2 (lines 565-573):

*"This study demonstrates that even a moderate sampling effort, including weekly CSIA measurements from composite topsoil samples across the catchment, can effectively identify hot spots and hot moments of pesticide degradation at the catchment scale. In PiBEACH, degradation is primarily driven by soil temperature and moisture, which exhibits clear seasonal and vertical gradients. As a result, degradation hotspots were consistently located in the topsoil. In contrast, soil moisture, although more spatial variable than temperature, played a secondary role. No significant spatial differences in $DT_{50}$ were observed among plots treated with S-metolachlor over the growing season. The limited predicted spatial variation in $DT_{50}$ across the catchment does not diminish the value of PiBEACH's distributed nature, particularly*

*when considering its future integration with distributed event-based models such as OpenLISEM-OLP. This coupling remains essential for simulating surface transport during rainfall events and refining predictions of pesticide fate under dynamic hydrological conditions."*

6. Line 510-511: see earlier comments: as pesticide concentrations and isotope ratios were only measured on the topsoil one cannot simply extrapolate to the catchment scale. Degradation rates are likely much higher in the topsoil than in the deeper soil layers leading to overestimation of a catchment to degrade pesticide. The data to constrain the model also at catchment scale are available. Why not use them?

We have now addressed the second major critique by clarifying in the manuscript (new section to the Materials and Methods, introducing the six metrics and in the new element of discussion) how discharge and S-metolachlor concentration in the river at the catchment outlet were used to calibrate the parameters and assess the simulated S-metolachlor export from the catchment. As previously discussed, we opted not to include outlet concentration and isotopic signature, mainly collected during rainfall-runoff events (Alvarez-Zaldivar et al., 2018), in the calibration function to avoid a biased calibration process, which could lead to parameter fitting for the wrong reasons in an attempt to compensate for the model's conceptual limitations.

Lines 544-553:

*"This modelling framework, applied to the same compound and catchment as in Lutz et al. (2017), provides insights into the benefits and limitations of using $\delta^{13}C$ data from soil and river samples to constrain reactive transport models. The lumped model by Lutz et al. (2017) focuses on $\delta^{13}C$ values at the catchment outlet, providing an integrated signature of degradation processes across the catchment. In contrast, PiBEACH allowed evaluation of the added value of topsoil $\delta^{13}C$ data for constraining degradation kinetics. Model predictions for three components of the S-metolachlor mass balance, degraded mass, remaining mass in topsoil, and export at the outlet, were validated against observations. Due to limitations in representing rapid surface off-site transport associated with runoff and erosion, PiBEACH did not incorporate $\delta^{13}C$ data from river into the calibration. However, future coupling of PiBEACH with an event-based model like OpenLISEM-OLP could enable simultaneous integration of both topsoil and river $\delta^{13}C$ datasets, improving constraints on degradation and enhancing simulation of fast-flow transport during high-intensity rainfall-runoff events."*

*Sylvain Payraudeau, on behalf of the co-authors*

*Strasbourg, April 2 2025*

---

## Author Response (AR2)

Anonymous referee #1:

I think the paper improved. However, the explanation about the Morris Scrrening parametrization was not enough. It is necessary to support the parametrization (and decisions) with other studies (citation is lacking).

We thank Referee #1 for the constructive comments, which have been carefully addressed and incorporated into the revised manuscript.

For example how many iterations did you run and why? How you decide that threshold to consider the parameter importance? You need to include citations of each decision that you made with Morris Screening parametrization.

We have now revised the section on the Morris method to clarify the threshold criteria applied for excluding less sensitive parameters, while adding new references.

We have also specified the model outputs considered in the sensitivity analysis and identified the Python library used.

Moreover, we have improved the linkage between the sensitivity results and the corresponding outputs, as illustrated in Figures S5 and S6 of the Supplement.

*L. 375-410: For the sampling method of parameters, PiBEACH required the calibration of 43 parameters (Table S3 in the Supplement). The range, i.e. min-max values, of these parameters were defined based either on literature or field data collected in 2012 and 2016 (Lefrancq et al., 2017 and 2018; Alvarez-Zaldivar et al., 2018). These parameters were assumed to be a priori uniformly distributed within these min and max values (Table S3 in the Supplement). To reduce the number of runs required by the GLUE method, three steps were successively applied. First, a pre-sensitivity global analysis based on the Morris method (Morris, 1991; Herman and Usher, 2017; Campolongo et al., 2007) was conducted with the SALib Python library for sensitivity analysis (section 10 of the Supplement) to select the most sensitive parameters. Although the Morris method yields a qualitative indication of relative parameter importance, it is efficient compared to other sensitivity approaches (Gan et al., 2014) that screen for sensitive parameters (Herman et al., 2013). The mean and standard deviation of the elementary effects (EE) for each parameter were calculated as required by the Morris method. The mean represents the overall effect of a parameter on the model output, while the standard deviation captures the potential for interactions or non-linear effects. Parameters with a mean EE of zero or near zero, indicating negligible impact, were excluded, resulting in the removal of 21 parameters. The sensitivity analysis was conducted for the following outputs: S-metolachlor concentrations at the outlet; concentrations in composite topsoil transects (North, Valley, and South); discharge at the outlet; and isotope signatures both in the river at the outlet and within the topsoil composite transects. The EE statistics for the 25 retained parameters are shown in Figures S5 and S6 of the Supplement for each output variable.*

*The Morris method allowed to reduce the PiBEACH parameter number from 43 to 25 (Table S3 in the Supplement). Second, a Latin-Hypercube sampling (Herman and Usher, 2017) was used to reduce the numbers of runs (n = 2500) to cover the parameter space for the 25 parameters. To further reduce the computation time, the GLUE assessment focused on the growing period (March 19th to July 12th, 2016), where pesticide degradation and exports are of most*

*significance. Initial hydrological state was estimated from a spin-up period of one full hydrological year (Oct. 1st, 2015 - Sept. 30th, 2016) and hydrological parameters calibrated against observed discharge at the catchment outlet (March 19th and July 12th, 2016) using particle swarm optimization (Bratton et al., 2007).*

Page 1 – line 9 (in the initial pdf):
Abstract:
**You have a lot of data, yet the abstract only reports qualitative results:**
We have now enhanced the abstract by incorporating key quantitative results.

***Abstract.*** *Predicting pesticide dissipation at the catchment scale using hydrological models is challenging due to limited field data distinguishing degradative from non-degradative processes. This limitation hampers the calibration of key parameters such as biodegradation and volatilization half-lives ($DT_{50}$), and the carbon-water partition coefficient ($K_{oc}$), often leading to equifinality and reducing confidence in predictions of pesticide persistence in topsoil and transport from agricultural field to catchment outlets. This study examines the use of pesticide Compound-Specific Isotope Analysis (CSIA) data to improve model predictions of pesticide persistence in topsoil and off-site transport at the catchment scale. The study was conducted in a 47-ha crop catchment using the pre-emergence herbicide S-metolachlor. A new conceptual distributed hydrological model, PIBEACH, was developed to simulate daily pesticide dissipation in soils and its transport to surface waters. The model integrates changes in the carbon isotopic signatures ($\delta^{13}C$) of S-metolachlor during degradation to constrain key parameters and reduce equifinality. Model and parameter uncertainties were estimated using the Generalized Likelihood Uncertainty Estimation (GLUE) method. Incorporating $\delta^{13}C$ data and S-metolachlor concentrations from topsoil samples reduced the uncertainty in the estimated degradation half-life $DT_{50}$ by more than half, yielding a value of 18 ± 4 days. This approach also significantly decreased uncertainty in six key metrics of pesticide persistence and transport. Between the day of application (day 0) and day 115, the modelled mass balance components, ranked by relative contribution, were as follows: degradation accounted for the majority at 82 ± 21%, followed by the remaining bioavailable mass in the topsoil at 12 ± 8%. Leaching contributed 4 ± 17%, while export to the river outlet accounted for 2 ± 6%. The irreversibly sorbed mass represented 1.1 ± 2.0%, and volatilization was minimal (<1%). The results highlighted that moderate, targeted sampling effort can identify degradation hot-spots and hot-moments in agricultural soil when stable isotope fractionation is integrated into the model. Overall, integrating CSIA data into PiBEACH model significantly enhances the reliability of pesticide degradation predictions at the catchment scale. In addition, PiBEACH, which accounts for spatial and seasonal variations in topsoil pesticide concentrations, enables coupling with distributed, event-based hydrological models such as OpenLISEM OLP to capture intra-event pesticide transport dynamics more accurately.*

Page 1 – line 31:
Why plural? / You mean pesticide degradation?

We have now modified the sentence as suggested:
*While pressure on aquatic ecosystems continues to increase, accurately quantifying and predicting the contribution of individual pesticide dissipation processes in soil, through degradation and off-site transport to the catchment outlet, remains a major challenge.*

Page 3 – line 88:
There is a noun missing here.

Indeed, we have now revised the sentence:
*L. 92. More recently, CSIA data have been integrated into a lumped transport model using travel-time distributions, improving the interpretation of pesticide transport at the catchment scale (Lutz et al., 2017)*

Page 3 – lines 90-92:
Most of this is captured in the travel time distribution. Your reasoning applies only to another type of lumped model. There are models that work with a convolution of application time distributions and travel time distributions, but these, to my knowledge, have not been applied to pesticides. I think your point is still valid, but the reasoning behind it can be improved.
We agree and we have now emphasised that travel time distributions can capture aspects of hydrological behaviour, as demonstrated in the study by Lutz et al. (2017), which to our knowledge represents one of the few applications in this context.

Additionally, we now have incorporated a recent reference addressing nitrate trends (Broers et al., 2024) (Lines 88–90). We have also highlighted that lumped models are limited in their ability to represent spatial variability of key parameters—such as soil moisture and temperature—that are critical for linking land use, pesticide application, and degradation processes.
Consequently, we have revised the sentence to clarify this limitation of lumped modelling approaches.

With a new reference:
*Broers, H.P., van Vliet, M., Kivits, T., Vernes, R., Brussée, T., Sültenfuß, J., and Fraters, D.: Nitrate trend reversal in Dutch dual-permeability chalk springs, evaluated by tritium-based groundwater travel time distributions, Sci Total Environ., 15;951:175250. https://doi.org/10.1016/j.scitotenv.2024.175250, 2024.*

*L 92: More recently, CSIA data have been integrated into a lumped transport model using travel time distributions, enhancing the interpretation of pesticide transport and transformation processes at the catchment scale (Lutz et al., 2017). However, lumped models primarily capture aggregate hydrological behaviour, with some applications in water quality such as nitrate trend analysis (Broers et al., 2024), but they do not account for spatial variability in land use or topsoil parameters, including soil moisture and soil temperature (Fatichi et al., 2016). This limitation restricts their capacity to represent landscape heterogeneity—such as variations in crop distribution and pesticide application—thereby impeding the accurate identification of contaminant sources and degradation hotspots (Grundmann et al., 2007).*

Page 4 - lines 108-109:
Isn't this an integral part of GLUE?
The sentence has been simplified to acknowledge that the Monte Carlo method is an integral component of the GLUE approach.

Page 5 – line 119 – figure 1:
I do not always understand the catchment boundary in relation to the ditches.

The landscape is predominantly flat, exhibiting slopes of less than 5.7% ± 2.9% throughout the catchment. Catchment delineation, derived from LiDAR data and validated by direct field observations during significant rainfall events, is strongly influenced by the topography of roads and tracks.

Page 5 – line 130
Most of the attributes are more chemical properties than composition

Indeed, and we have now modified the sentence as suggested:
*The soil texture is predominantly composed of silt (61.0 ± 4.5%), followed by clay (30.8 ± 3.9%) and sand (8.5 ± 4.2%). The soil also contains calcium carbonate ($CaCO_3$: 1.1 ± 1.6%), organic matter (2.2 ± 0.3%), and total soluble phosphorus (0.11 ± 0.04 g $kg^{-1}$), and exhibits a cation exchange capacity (CEC) of 15.5 ± 1.3 cmol $kg^{-1}$.*

Page 5 – line137:
That's from top to bottom in Fig. 1, I presume.

Indeed, we have modified the caption of Figure 1 to explicitly include the names of the three transects: North, Valley, and South.
***Figure 1:*** *The Alteckendorf headwater catchment (Bas-Rhin, France), showing the experimental setup, including three transects, i.e. North, Valley and South (weighted samples collected at green dots along red lines) and plot sampling (black dots). Land use for 2016 is also displayed. The "Other" category includes roads, grass strips and orchards.*

Page 5 – line 140:
I presume you mean 'Samples from the topsoil

Indeed, we have now revised the sentence.
*Samples from the topsoils (0-1 cm) were collected from individual plots and upstream-downstream transects across the catchment (Fig. 1 and 140 S1; Alvarez-Zaldivar et al., 2018).*

Page 5 – line 142:
Do you mean you retrieved the same mass of soil at x locations along a transect, then mixed all these samples, and then obtained a single subsmample of the mixture, which was then analyzed for S-metolachlor and isotopes?

Indeed, we have not modified this sentence.
A single mixed sample was collected weekly, combining 30, 25, and 27 subsamples (green dots in the figure) for the North, Valley, and South transects, respectively.

Page 5 – line 142:
Were the masses of the samples taken in the field determined at field soil water content?)

The subsamples were collected in the field using a fixed volume rather than a fixed mass, as the latter is not feasible at the field scale.

Page 5 – line 144:
This was not done for the composite samples, or was it?

We have modified the sentence to clarify that these calculations were performed on both plot-scale and composite samples, as detailed below:
*The volumetric topsoil water content (m³ water m⁻³ soil) was calculated from gravimetric measurements obtained after drying samples at 110°C following NF ISO 1146 (Lefrancq et al., 2018). This procedure was applied to all samples, including field samples collected on days 1, 50 and 100 after application, as well as the weekly mixed samples from the three transects. It* incorporated *seasonal variations* in *topsoil bulk density, as modelled by PiBEACH and* detailed *in the Supplement, Section 7.2.*

Page 6 – line 151.
In the previous sentence you stated the sampling was flow-proportional, but here the sampling was done weekly. I don't follow.

Indeed, flow-proportional sampling was conducted using an ISCO Avalanche autosampler. However, the collected samples were retrieved only on a weekly basis for subsequent analyses. This has now been clarified

I also do not understand what a fixed weekly discharge volume is. The \(cumulative\) discharge volume between samplings can either be fixed, or you can sample at fixed time intervals, but not both.

To optimize our sampling strategy—balancing the need to collect sufficient water for quantifying S-metolachlor concentrations, particularly for $\delta^{13}C$ analysis, while limiting the number of collection bottles to twelve 330 mL units per week—we implemented a variable threshold based on cumulative discharge volume to trigger sampling. This threshold was progressively increased from March to June to reflect the seasonal rise in baseflow discharge. Specifically, one bottle was filled after every 50 m³ of cumulative discharge in March, whereas by June, the threshold had increased to 150 m³ per bottle.

L. 176-184. This has now been clarified:
*Runoff discharge at the catchment outlet was measured using a Doppler flowmeter (2150 Isco) with 3% accuracy and a 2 min resolution. Continuous, refrigerated, flow-proportional sampling was carried out using an Isco Avalanche autosampler equipped with twelve 330 mL bottles. Samples were collected based on fixed weekly discharge volumes from 50 to 150 m³, in order to capture progressive increase in baseflow discharges from April to June 2016 (Alvarez-Zaldivar et al., 2018). To obtain sufficient S-metolachlor for quantification and CSIA, weekly*

*composite samples were prepared by pooling bottles according to hydrograph phase (base-flow, rising limb, and falling limb), yielding one to four samples per week with volumes ≥ 990 mL (Alvarez-Zaldivar et al., 2018). Piezometric monitoring of the shallow aquifer was not possible due to the absence of observation wells at the study site.*

Page 6 – line 158:
river water:

We have modified the sentence accordingly:
*To separate dissolved and particulate phases of S-metolachlor, river water samples were filtered through 0.7 µm glass fiber filters.*

*Page 7 – line 169:*
This is still incomplete. The first paragraph is too general to give the reader enough background to understand the later paragraphs. For instance, the way in which the model represents the landscape and topography is not explained at all, and neither is the way the soil is represented. In later paragraphs you refer to cells, the plow layer and a deeper soil layer, without the reader knowing how these fit into the model.

I do not see this part of the acronym in the full name. Is something missing?
Indeed, BEACH is the acronym for *Bridge Event And Continuous Hydrological* modelling; accordingly, the sentence has been corrected accordingly:
*The Pesticide-isotopes BEACH model (PiBEACH) was developed in Python based on the conceptual Bridge Event And Continuous Hydrological (BEACH) model (Sheikh et al., 2009).*

You have not described the model cells yet. Because you have not explained how the cells are configured, and how the interactions between neighbouring cells are implemented, this text, and much of the rest of this modified text, is unclear.

We have clarified in the "PiBEACH development" section how the model cell is defined in this article and revised Figure 2 to schematically illustrate these cells

*L 84: Similar to BEACH, the PiBEACH model employs square cells (x, y) with variable depths (z) corresponding to the soil layers considered (Fig. 2), in order to represent water and pesticide movement within the catchment, as detailed below.*

Page 7 – line 197:

This is strange. The dispersion has dimensions, the export coefficient is dimensionless. How can you can compare their numerical values, given that you can change the value of one of them by simply changing the units?

Indeed, we acknowledge that our original sentence could be interpreted in this way. In Gatel et al. (2020), the impact of numerical dispersion was assessed as a dimensionless fraction of

the applied pesticide mass within the overall mass balance. We have now revised the sentence to clarify this comparison.

*L 205: This challenge arises because numerical dispersion can affect the mass balance at the catchment scale (Gatel et al., 2020) to an extent comparable to the pesticide export coefficient—defined as the ratio of the mass transported at the outlet to the total mass applied within the catchment—which typically ranges from 0.1‰ to 1% of the applied pesticide load (Lefrancq et al., 2018).*

Page 8 – line 222:

Should Z not be a depth interval? If so, what are the boundaries of this interval? The text above suggests that the interval comprises layers z0 - z2, and that the interval therefore is 0-80 cm. But you need to explain that better.

Equation 1 has also been modified to clearly indicate its application across the depth intervals from $z_0$ to $z_3$, using the generalised notation $z_j$ (with j=0 to 3).

Figure 2 has been revised to use italic font for variables, in accordance with HESS formatting guidelines. In the updated version, the raster-based structure of PiBEACH is highlighted by depicting square cells with a $2 \times 2$ m plan view and variable depths corresponding to the different soil layers described earlier in the document and in the manuscript.

Page 8 – line 226:

This, and other variables like this, does not conform to the HESS guidelines. The notation is also inconsistent with that of other variables. Fonts that are italic in the text are regular here. Please make this consistent, and keep in mind HESS guidelines. Notation not in line with the rest of the paper, and with HESS guidelines. This happens too often to keep flagging it. Please go over the entire paper carefully to fix all occurrences.

We have thoroughly reviewed and revised all variables to appear in italic font, and have updated Figure 2 accordingly to ensure consistency with the HESS guidelines.

Page 9 – line 251:
We have corrected the sentence as suggested

Page 10 – line 266-271:
This limits the applicability of the new approach, does it not? Do you imply that this makes it acceptable to ignore isotope fractionation? If so, state this explicitly.

Indeed, this implies that isotope fractionation induced by sorption can be ignored.
This has now been specified in the manuscript.
L. 323-326. *In our case, since isotope fractionation associated with sorption and ageing processes is expected to be negligible, the observed isotope fractionation can be attributed*

*primarily to biodegradation. This justifies the exclusion of non-destructive processes from the isotope mass balance and supports the use of the model to distinguish between destructive, i.e., biodegradation, and non-destructive processes, thereby enabling a quantitative evaluation of the contribution of biodegradation to pesticide dissipation.*

Page 11 – line 296:
If these variables denote functions of temperature and water content, then say so.

We have modified the sentence as suggested:
*A dynamic degradation rate ($k_{Dynamic}$, $d^{-1}$) was calculated daily as a function of soil temperature ($F_T$) and of water content ($F_\theta$):*

Page 11 – line 300:
What is this?

As introduced in line 296, a dynamic degradation rate ($k_{Dynamic}$, $d^{-1}$) was calculated daily from a $K_{ref}$ and a function of soil temperature and of water content. The $k_{dynamic}$ provides a half time $DT_{dynamic}$ (day) calculated as $DT_{50, Dynamic} = \ln(2) / k_{Dynamic}$.

We have revised the sentence to clarify this step.
*A dynamic half-time $DT_{50,Dynamic} = \ln(2) / k_{Dynamic}$ was derived to be compared to $DT_{50,Ref}$.*

Page 11 – line 296:
The units of F sub T are unclear.

We have now modified the sentence to clarify that the dependence equation of soil temperature ($F_T$) and soil moisture ($F_\theta$) are unitless.
*A dynamic degradation rate ($k_{Dynamic}$, $d^{-1}$) was calculated daily as a function of soil temperature ($F_T$) and of water content ($F_\theta$):*

*Page 11 – line 315 to Page 12 – line 320:*
This text seems out of place here. It contains some observation that should be in the Results section, and does not have a clear connection to the preceding text, even though it is part of the same paragraph. Because the line of thought is broken, I do not understand what point is being made here.

We acknowledge that this part of the text may be out of place. Therefore, we have moved this section, providing intermediate results, in the section results (3.1 Topsoil hydro-climatic dynamics and effect on S-metolachlor degradation rates).

Page 12 – line 329:

We have corrected the font size of the word "pesticide"

Page 12 – line 335:
Is this adjective necessary in this context? and plural for "macropore"

We have now modified the sentence by removing "explicit" and correction on macropores.
*However, the integration of macropores at the catchment scale necessitates advanced in situ measurements (Weiler, 2017), and a combination of geostatistical methods, pedotransfer functions or meta-models, i.e., simplified statistical models built with 1D soil reactive transport models such as MACRO (Lindahl et al. 2008).*

Page 12 – line 339:
Why is this sentence here?

We have removed this sentence and instead referenced previous work highlighting the negligible contribution of S-metolachlor plant uptake to the S-metolachlor mass balance at the plot scale (Lefrancq et al., 2018).
*However, plant uptake of S-metolachlor was not included, as it is likely negligible (Lefrancq et al., 2018).*

Page 13 – section 2.6:
Do I understand correctly that you calibrated the model on the full data set, and that, therefore, a validation was not carried out?
In that case, the graphs in the R&D section show how well the model reproduced the measurements, but tells us nothing about its predictive capabilities. this is correct, the discussion needs to reflect this.

We acknowledge the Referee's comments regarding the distinction between calibration and validation. The mention of model validation has been removed, and the following sentence has been added at the end of Section 3.2.
*L 526: However, these findings necessitate further confirmation with a validation dataset, which was not available for the targeted catchment.*

And the following in the conclusion:
*L 620: The next step should involve confirming these findings with a validation dataset, which was not available for the targeted catchment.*

Page 13 – line 359:
"Parameter values" / "with"
The sentence has been modified as suggested:
*The GLUE method involved a sampling method of PiBEACH parameters values, an objective function incorporating observed dataset (i.e., topsoil S-metolachlor concentration only, then combined with S-metolachlor $\delta^{13}C$), a threshold of this objective function to select behavioural parameter sets, and the calculation of posterior probability distributions for parameters and uncertainties associated to the outputs of PiBEACH.*

Page 14 – line 386:

Why past tense? / Between simulated and observed values? / Should S not be sigma?
The sentence has now been revised to correct the verb tense and to clarify the variables used in the calculation of the correlation coefficient, replacing the letter $S$ with the Greek symbol $\sigma$ to ensure consistency in notation.

where r is the linear correlation coefficient between simulated and observed values, $\alpha_{KGE} = \sigma_i / \sigma_0$, and $\beta_{KGE} = \mu_i / \mu_0$, where $\sigma$ and $\mu$ denoting the standard deviation and mean of simulated and observed values, respectively.

Page 14 – line 391:
I don't understand what this means.

We have now revised the sentence to enhance clarity and improve reader comprehension:

L. 447-451. The Kling–Gupta Efficiency (KGE) provides a more balanced assessment of model performance than traditional metrics such as the Mean Squared Error (MSE) or Nash–Sutcliffe Efficiency (NSE), which often favor parameter sets that underestimate output variability.

Page 14 – line 395:

I think this should be a separate sentence, it is grammatically disconnected from the first part of the sentence, making the meaning of the full sentence (and its grammar) dubious.
Thank you. The sentence has been repositioned to enhance clarity and improve reader comprehension.

These three approaches to aggregating topsoil data were developed to determine the minimum sampling effort for S-metolachlor concentration and $\delta^{13}C$ required to minimise uncertainties in PiBEACH model outputs.

Page 14 – line 401:

How did you arrive at these thresholds?
According to the classification proposed by Kling et al. (2012) and subsequently adopted by Towner et al. (2019), the goodness-of-fit between simulated and observed variables is categorised as "very poor" for KGE ≤ 0, "poor" for KGE ≤ 0.5, "intermediate" for KGE ≤ 0.75, and "good" for KGE > 0.75. In this study, the intermediate threshold of KGE > 0.5 was retained for both S-metolachlor concentration in topsoil ($KGE_{SM}$ >0.5) and discharge ($KGE_Q$ >0.5) at the outlet. A more stringent threshold was applied to the weekly topsoil $\delta^{13}C$ ($KGE_\delta$ > 0.8) in order to better leverage this information as an indicator of degradation. These thresholds were ultimately retained as a compromise to ensure the selection of simulations of at least intermediate quality, while maintaining a sufficient number of parameter sets to derive outputs with robust confidence intervals.

The sentence has been modified to clarify the rationale underlying the selection of specific KGE thresholds, thereby providing a more transparent justification for their application in this study, supported by the inclusion of two additional references.

Kling, H., Fuchs, M., and Paulin, M.: Runoff conditions in the upper Danube basin under an ensemble of climate change scenarios, J. Hydrol., 424, 264–277, https://doi.org/10.1016/j.jhydrol.2012.01.011, 2012.

Towner, J., Cloke, H. L., Zsoter, E., Flamig, Z., Hoch, J. M., Bazo, J., Coughlan de Perez, E., and Stephens, E. M.: Assessing the performance of global hydrological models for capturing peak river flows in the Amazon basin, Hydrol. Earth Syst. Sci., 23, 3057–3080, https://doi.org/10.5194/hess-23-3057-2019, 2019.

*According to the classification proposed by Kling et al. (2012) and subsequently adopted by Towner et al. (2019), the goodness-of-fit between simulated and observed variables is categorised as "very poor" for KGE ≤ 0, "poor" for KGE ≤ 0.5, "intermediate" for KGE ≤ 0.75, and "good" for KGE > 0.75, the threshold to retain acceptable model results runs (out of 2500 simulation runs) for topsoil S-metolachlor degradation and transport was set to $KGE_{SM}$ >0.5 and $KGE_Q$ >0.5. An additional and more stringent criterion ($KGE_\delta$ > 0.8) was applied to weekly topsoil $\delta^{13}C$ data to maximise its value as an indicator of degradation processes. These thresholds were ultimately selected as a compromise, enabling the retention of simulations with at least intermediate accuracy while preserving a sufficient number of parameter sets to support the derivation of outputs with robust confidence intervals.*

Page 14 – line 402:

Hypotheses generally belong in the Introduction.
This sentence has been removed, as the underlying hypothesis is already stated in Introduction.

Page 14 – line 406:
 This comes out of the blue. The last step mentioned is step 2.
The sentence has been revised to accurately indicate that this step corresponds to the final stage of the GLUE framework.
*L. 433: In the final step of the GLUE procedure, the distributions of the 25 most sensitive parameters were extracted from the subset of acceptable parameter sets, i.e. $KGE_{SM}$ >0.5 and $KGE_Q$ >0.5 and $KGE_\delta$ >0.8. PiBEACH outputs were then expressed as the mean considering the 95 % confidence intervals based on these parameter sets, excluding the lower 2.5% and the upper 2.5% of acceptable simulations.*

Page 14 – line 407:
The population of acceptable parameter sets gives a range of acceptable values for each of these parameters, so how come they have a confidence interval? Did you determine the joint distribution of all parameters? Then I can see how you can arrive at confidence intervals, but

that step is missing. In any case, how reliable are the parameter statistics, bases as they are on a sparse sampling (Latin Hypercube) of a 25-dimension parameter space? I just saw that you explain some of this at the end of the paragraph, which makes the line of thought in the paragraph hard to follow, so please rewrite it. I see from there that you did not pursue the joint distribution (which is not trivial for 25 parameters). It would be nice to know what kind of distributions you found, or did you assume normal distributions for the lot?)

The methodology for deriving the 95% ensemble confidence interval for the six PiBEACH outputs, as detailed from lines 405 to 413, has been refined to improve clarity and remove redundancy. Specifically, this involved the systematic exclusion of the lower and upper 2.5% of acceptable simulations to determine the 2.5th and 97.5th percentiles, which are illustrated in Figures 3 and 5. Additionally, the posterior distributions of the 25 parameters retained were provided in Table S3 of the Supplement, delineated by the 2.5$^{th}$ and 97.5$^{th}$ percentiles. As the distributions were not normal for all parameters, as illustrated with the $K_{OC}$ distribution, (Figure S8 in the Supplement), the range (2.5th and 97.5th percentiles) were derived from exclusion of the lower and upper 2.5% of each parameter.

We have now revised the sentence to enhance clarity and improve reader comprehension:
L. 433: In the final step of the GLUE procedure, the distributions of the 25 most sensitive parameters were extracted from the subset of acceptable parameter sets, i.e. $KGE_{SM}$ >0.5 and $KGE_Q$ >0.5 and $KGE_\delta$ >0.8. PiBEACH outputs were then expressed as the mean considering the 95 % confidence intervals based on these parameter sets, excluding the lower 2.5% and the upper 2.5% of acceptable simulations.

Page 14 – line 407:
This is a valid point, but it needs to be made elsewhere. It does not have anything to do with the purpose of section 2.6)

We have now moved this statement in the section 2.4 dedicated to PiBEACH model description:
L 332: Calibrating the PiBEACH parameters, as detailed in Section 2.6, was challenging and warranted the collection of an extensive dataset across the catchment throughout the growing season. This unique dataset incorporated isotopic signatures and comprised 103 topsoil samples analysed for S-metolachlor concentration and $\delta^{13}C$, 115 daily discharge measurements, and 51 river outlet samples with corresponding S-metolachlor concentrations.

Page 15 – line 414:
This is useful.

Page 15 – line 430:
Simulated topsoil...

The sentence has been modified:

*Simulated topsoil ($z_0$) water contents showed substantial variability, consistent with weekly field measurements (Fig. 3A) and previous application of the BEACH model in catchments with similar soils, crops and conditions (Sheikh et al., 2009).*

Page 15 – line 432:
Use this term when you introduce KGE for the first time, not here.

The sentence has been modified:
*Simulated discharges at the catchment outlet closely matched observations (Fig. 3B), with a maximum $KGE_Q$ of 0.75, demonstrating the model's ability to capture prevailing hydrological dynamics.*

The full term was introduced at its first occurrence.
*L. 403: For the second step of the GLUE method, the Kling-Gupta Efficiency (KGE) (Gupta et al., 2009) metric was adopted as the objective function to maximize during calibration.*

Page 15 – line 432:
With a KGE of 0.75, that's a bit optimistic, as Fig 3B shows.

The sentence has been revised to moderate the assessment of the PiBEACH model's performance in simulating daily discharge, as it was not primarily designed for this purpose, while still highlighting its ability to capture the prevailing hydrological dynamics.

*Simulated discharges at the catchment outlet showed reasonable agreement with observations (Fig. 3B), with a maximum $KGE_Q$ of 0.75, demonstrating the model's ability to capture prevailing hydrological dynamics.*

Page 17 – line 452:
"repetitive"

The sentence has been revised to eliminate redundancy and avoid repetition of information.

*L. 479: Out of 2500 simulation runs , 672 were deemed acceptable based on hydrological and concentration performance criteria ($KGE_Q$ >0.5 and $KGE_{SM}$ >0.5).*

Page 17 – line 455:
... ensemble of acceptable simulations to 244.

We have modified the sentence to clarify the term "ensemble":

*L. 481: Applying an additional constraint based on isotope data ($KGE_\delta$ >0.8) further reduced the ensemble of acceptable simulations to 244 simulations.*

Page 17 – line 461:

This belongs in methodology, not here.

This introductory sentence has been revised to minimise redundancy regarding the methodological objectives, while retaining a concise reminder of the two calibration strategies and their respective designations.

*L. 487: This section underlines the benefit of incorporating topsoil CSIA data during model calibration (WIC: with isotope constraint) compared to no isotope constraint (NIC).*

Page X – line Y:
This paragraph suggest that you can make more accurate predictions, but I suspect you are only reporting improved data fitting, because the methodology suggest that you calibrated but not validated your model. See my comment above about the need to have the discussion correctly reflect what exactly you did with your model. If you have indeed calibrated your model on the full data set, the text here would be misleading.

This paragraph illustrates how equifinality in parameter estimation—particularly for $DT_{50}$ related to pesticide dissipation—can be reduced by incorporating both the isotopic signature and concentration of S-metolachlor (WIC), rather than relying solely on topsoil concentration data (NIC). In both cases, the datasets were used during model calibration. Calibration with WIC proved more effective, resulting in a narrower acceptable range for $DT_{50}$ compared to NIC. Notably, the full 2016 dataset was used for both model development and calibration, without a separate validation phase. We have revised the sentence to emphasize the reduction of equifinality during the calibration process.

*L. 499: Reducing uncertainty in estimates of pesticide degradation in soil during the calibration of reactive transport models during calibration is crucial, as degradation half-lives can vary by one order of magnitude depending on the compound (Wang et al., 2018), largely affected by hydro-climatic and soil conditions. In this study, the WIC calibration yielded mean $DT_{50,Ref}$ below 20 days, with low standard deviations (SD <7 days; Fig. 4), indicating that aerobic degradation of S-metolachlor, typically reported between 14 and 21 days (Lewis et al., 2016), was the dominant process in Alteckendorf topsoil. In contrast, anaerobic degradation, characterized by longer half-lives ($DT_{50}$ = 23 - 62 days; Seybold et al., 2001; Long et al., 2014), appeared to play a limited role.*

Page 19 – line 598:
You are probably correct, but you cannot claim this so strongly based on an unvalidated model, which is what I believe you have here.

We have revised the sentence to highlight that the integration of compound-specific isotope analysis (CSIA) enhances the calibration step of the PiBEACH model.

*L. 521: These findings underscore the importance of site-specific calibration in CSIA applications and highlight the value of model ensemble approaches in capturing the range of degradation processes in heterogeneous agro-ecosystems. While previously reported εC values may slightly overestimate degradation in some field settings, the calibration of ensemble modelling with integrated CSIA data provides a more robust and field-relevant assessment of*

*pesticide transformation. However, these findings necessitate further confirmation with a validation dataset, which was not available for the targeted catchment.*

Page 19 – line 514:
This suggests prediction, but it really is just fitting, isn't it?

Observed S-metolachlor exports at the catchment outlet were not used during model calibration and therefore serve as a form of partial validation. Only discharge at the outlet was calibrated, with a Kling–Gupta Efficiency ($KGE_Q$) exceeding 0.5. Consequently, our classification of the results as a 'slight overestimate' is supported by the comparison between simulated and observed S-metolachlor exports, which was performed independently of direct calibration to those data.
We have revised this sentence to clarify this point:
*L. 541: The model slightly overestimated the export of S-metolachlor to the outlet (2 ± 6%) in comparison to the observed values (0.5 ± 0.1%), which were not mobilised during calibration. However, this difference remains within the model's uncertainty range.*

Page 19 – lines 515 and 516:
river water ? fits ?

We have revised the sentence as follows:
*L. 543: It is important to note that observed export**s** were based solely on the dissolved phase, as particulate-bound S-metolachlor (> 0.7 μm) remained below quantification limits in all river* water *samples. The S-metolachlor export metric depends on PiBEACH ability to simulate daily discharge*

Page 19 – line 519:
I think you are overstating the model performance here.

We have revised the sentence to moderate this statement:
*L. 546: Although PiBEACH was originally designed to initialize sub-hourly, event-based distributed models like Openlisem-OLP (Commelin et al., 2024), it also demonstrated a reasonable ability to reproduce daily discharge dynamics (Fig. 3B), consistent with prior applications in similar catchments (Sheikh et al., 2009).*

Page 21 – line 528:
Do the massive confidence intervals not suggest that the criteria for acceptance of a set of parameter values should perhaps have been set more strictly? I am not suggesting you should redo the entire GLUE procedure with new KGE thresholds, but it would be worthwhile taking this up in the discussion. The problem with GLUE is that it will be very hard to estimate a priori the best thresholds for the objective function, while the massive computational demand of a single GLUE run makes it prohibitive to narrow down the range of the thresholds by trial and error. This is not something you can or need to resolve, but you can write a not too long paragraph about it.

We acknowledge the Reviewer's comment and have added a new sentence addressing the question of the optimal thresholds for the objective function.

*L. 569: The wide 95% confidence intervals observed for the six metrics suggest that the thresholds used for the objective function, particularly $KGE_{SM} > 0.5$ and $KGE_Q > 0.5$, may need to be increased to reduce uncertainty. As previously noted in the application of the GLUE method (Jin et al., 2010), selecting an optimal threshold is inherently challenging, as it involves a trade-off between the computational effort required to retain a sufficient number of acceptable simulations and the resulting width of the confidence intervals.*

Page 23 – 568:
Fitted?

We have revised the targeted sentence:
*L. 600: The limited spatial variation in simulated $DT_{50}$ across the catchment does not diminish the value of PiBEACH's distributed nature, particularly when considering its future integration with distributed event-based models such as OpenLISEM-OLP.*

Page 23 – line 578:
But you did not address that in this paper, did you?

We have removed the validation term to prevent any potential misinterpretation, as our work primarily focuses on improving the model calibration process and addressing the issue of parameter equifinality.

*L. 609: This study addresses the gap between the increasing complexity of reactive transport models and the limited availability of field data for their calibration.*

Page 23 – line 587:
I am confused. I saw no evidence of model validation in the paper. Did I overlook something?
As for the previous sentence, we have removed the validation term to prevent any potential misinterpretation.

*L. 618: In many cases, carbon isotope data ($\delta^{13}C$) alone may be adequate to provide evidence of in situ degradation, thereby supporting pesticide mass balance closure and improving model calibration at the catchment scale.*

Page 24 – line 595:
There is no need to introduce an abbreviation at the end of the paper.

The abbreviation has been removed from the targeted sentence